# No-regret learning in games with noisy feedback: Faster rates and adaptivity via learning rate separation

**Yu-Guan Hsieh**[*]
yu-guan.hsieh@univ-grenoble-alpes.fr

**Kimon Antonakopoulos**[†]
kimon.antonakopoulos@epfl.ch

**Volkan Cevher**[†]
volkan.cevher@epfl.ch

**Panayotis Mertikopoulos**[*‡]
panayotis.mertikopoulos@imag.fr

## Abstract

We examine the problem of regret minimization when the learner is involved in a continuous game with other optimizing agents: in this case, if all players follow a no-regret algorithm, it is possible to achieve significantly lower regret relative to fully adversarial environments. We study this problem in the context of variationally stable games (a class of continuous games which includes all convex-concave and monotone games), and when the players only have access to noisy estimates of their individual payoff gradients. If the noise is additive, the game-theoretic and purely adversarial settings enjoy similar regret guarantees; however, if the noise is *multiplicative*, we show that the learners can, in fact, achieve *constant* regret. We achieve this faster rate via an optimistic gradient scheme with *learning rate separation* – that is, the method's extrapolation and update steps are tuned to different schedules, depending on the noise profile. Subsequently, to eliminate the need for delicate hyperparameter tuning, we propose a fully adaptive method that attains nearly the same guarantees as its non-adapted counterpart, while operating without knowledge of either the game or of the noise profile.

## 1 Introduction

Owing to its simplicity and versatility, the notion of regret has been the mainstay of online learning ever since the field's first steps [6, 17]. Stated abstractly, it concerns processes of the following form:

1. At each stage $t = 1, 2, \ldots$, the learner selects an action $x_t$ from some $d$-dimensional real space.
2. The environment determines a convex loss function $\ell_t$ and the learner incurs a loss of $\ell_t(x_t)$.
3. Based on this loss (and any other piece of information revealed), the learner updates their action $x_t \leftarrow x_{t+1}$ and the process repeats.

In this general setting, the agent's regret $\mathrm{Reg}_T$ is defined as the difference between the cumulative loss incurred by the sequence $x_t$, $t = 1, 2, \ldots, T$, versus that of the best fixed action over the horizon of play $T$. Accordingly, the learner's objective is to minimize the growth rate of $\mathrm{Reg}_T$, guaranteeing in this way that the chosen sequence of actions becomes asymptotically efficient over time.

Without further assumptions on the learner's environment or the type of loss functions encountered, it is not possible to go beyond the well-known minimax regret bound of $\Omega(\sqrt{T})$ [18, 37], which is achieved by the online gradient descent (OGD) policy of Zinkevich [39]. However, this lower bound concerns environments that are "adversarial" and loss functions that may vary arbitrarily from one stage to the next: if the environment is "smoother" – and not actively seeking to sabotage the learner's efforts – one could plausibly expect faster regret minimization rates.

---

[*]Univ. Grenoble Alpes
[†]EPFL
[‡]CNRS, Inria, LIG & Criteo AI Lab

36th Conference on Neural Information Processing Systems (NeurIPS 2022).

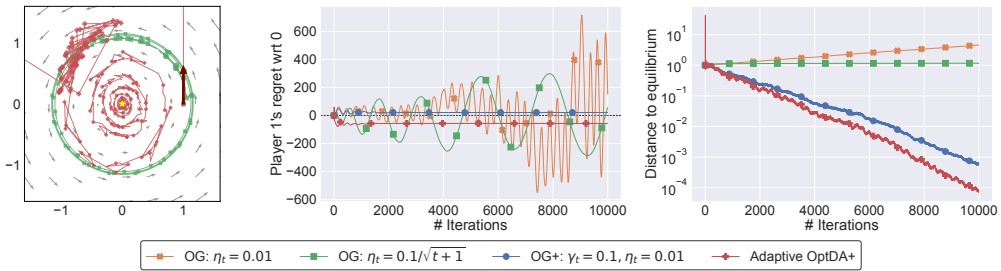

**Figure 1:** The behavior of different algorithms on the game $\min_{\theta \in \mathbb{R}} \max_{\phi \in \mathbb{R}} \theta\phi$ when the feedback is corrupted by noise. Left: trajectories of play. Center: regret of Player 1. Right: distance to equilibrium. Adaptive OptDA+ is run with $q = 1/4$. See Example 1 for the details of the model and Appendix C for additional figures.

This question is particularly relevant – and has received significant attention – in the backdrop of *multi-agent* learning in games. Here, the learners' environment is no longer arbitrary: instead, each player interacts with other regret minimizing players, and every player's individual loss function is determined by the actions chosen by all players via a fixed underlying mechanism – that of a *non-cooperative game*. Because of this mechanism – and the fact that players are changing their actions incrementally from one round to the next – the learners are facing a much more "predictable" sequence of events. As a result, there has been a number of research threads in the literature showing that it is possible to attain *near-constant regret* (i.e., at most polylogarithmic) in different classes of games, from the work of [10, 23] on finite two-player zero-sum games, to more recent works on general-sum finite games [1, 2, 12], extensive form games [16], and even continuous games [20].

**Our contributions in the context of related work.** The enabling technology for this range of near-constant regret guarantees is the *optimistic gradient* (OG) algorithmic template, itself a variant of the extra-gradient (EG) algorithm of Korpelevich [25]. The salient feature of this method – first examined by Popov [34] in a game-theoretic setting and subsequently popularized by Rakhlin and Sridharan [35] in the context of online learning – is that players use past gradient information to take a more informed "look-ahead" gradient step that stabilizes the method and leads to lower regret. This, however, comes with an important caveat: all of the above works crucially rely on the players' having access to exact payoff gradients, an assumption which is often violated in practice. When the players' feedback is corrupted by noise (or other uncertainty factors), the very same algorithms discussed above may incur *superlinear regret* (cf. Figure 1). We are thus led to the following natural question:

*Is it possible to achieve constant regret in the presence of noise and uncertainty?*

Our paper seeks to address this question in a class of continuous games that satisfy a variational stability condition in the spirit of [20, 27]. This class contains all bilinear min-max games (the unconstrained analogue of two-player, zero-sum finite games), cocoercive and monotone games, and it is one of the settings of choice when considering applications to generative models and robust reinforcement learning [7, 19, 22, 26]. As for the noise contaminating the players' gradient feedback, we consider two standard models that build on a classical distinction by Polyak [33]: (*a*) *additive*; and (*b*) *multiplicative* gradient noise. The first model is more common when dealing with problem-agnostic first-order oracles [30]; the latter arises naturally in the study of randomized coordinate descent [30], asynchronous player updating schemes [4], signal processing and control [36], etc.

In this general context, our contributions can be summarized as follows:

1. We introduce a learning rate separation mechanism that effectively disjoins the extrapolation and update steps of the OG algorithm. The resulting method, which we call OG+, guarantees $\mathcal{O}(\sqrt{T})$ regret in the presence of additive gradient noise; however, if the noise is multiplicative and the method is tuned appropriately, it achieves *constant* $\mathcal{O}(1)$ regret.

2. On the downside, OG+ may fail to achieve sublinear regret in an adversarial environment. To counter this, we propose a "primal-dual" variant of OG+, which we call OptDA+, and which retains the above properties of OG+, while achieving $\mathcal{O}(\sqrt{T})$ regret in the adversarial case.

3. Subsequently, to obviate the need for delicate hyperparameter tuning, we propose a fully adaptive method that enjoys nearly the same regret guarantees as mentioned above, without any prior knowledge of the game or of the uncertainties involved. Interestingly, our method features a

|  | Adversarial | All players run the same algorithm | | | |
|---|---|---|---|---|---|
|  | Bounded feedback | Additive noise | | Multiplicative noise | |
|  | Regret | Regret | Convergence | Regret | Convergence |
| OG | ✗ | ✗ | ✗ | ✗ | ✗ |
| OG+ | ✗ | $\sqrt{t}\log t$ | ✓ | constant | ✓ |
| OptDA+ | $\sqrt{t}$ | $\sqrt{t}$ | – | constant | ✓ |
| AdaOptDA+ ($q = 1/4$) | $t^{3/4}$ | $\sqrt{t}$ | – | constant | ✓ |

**Table 1:** Summary of the results obtained in the paper. The cross ✗ indicates a negative result (divergence of trajectory or potentially unbounded regret with decreasing stepsize) while a dash "–" means that the behavior of the algorithm is unknown. Our methods improve upon vanilla OG by separating the two step-size schedules.

    trade-off between achieving small regret when facing adversarial opponents and achieving small regret when facing opponents that adopt the same prescribed strategy, which prevents us from obtaining the optimal $\mathcal{O}(\sqrt{T})$ regret bound in the former situation.

4. Finally, we complement our analysis with a series of equilibrium convergence results for the range of algorithms presented above under both additive and multiplicative noise.

To the best of our knowledge, our work is the first in the literature to point out that constant regret may still be achievable in the presence of stochasticity (even in the simplest case where the noise profile is known in advance). In this regard, it can be seen as a first estimation of the degree of uncertainty that can enter the process before the aspiration of constant (or polylogarithmic) regret becomes an impossible proposition.[4]

A summary of our results is presented in Table 1. In the paper's appendix, we discuss some further related works that are relevant but not directly related to our work. We also mention here that our paper focuses on the unconstrained setting, as this simplifies considerably the presentation and treatment of multiplicative noise models. We defer the constrained case (where players must project their actions to a convex subset of $\mathbb{R}^d$), to future work.

## 2   Problem Setup

Throughout this paper, we focus on deriving optimal regret minimization guarantees for multi-agent game-theoretic settings with noisy feedback. Starting with the single-agent case, given a sequence of actions $x_t \in \mathcal{X} = \mathbb{R}^d$ and a sequence of loss functions $f_t \colon \mathcal{X} \to \mathbb{R}$, we define the associated *regret* induced by $x_t$ relative to a benchmark action $p \in \mathcal{X}$ as

$$\text{Reg}_T(p) = \sum_{t=1}^{T}[f_t(x_t) - f_t(p)]. \tag{1}$$

We then say that learner has *no regret* if $\text{Reg}_T(p) = o(T)$ for all $p \in \mathcal{X}$. In the sequel, we extend this basic framework to the multi-agent, game-theoretic case, and we discuss the various feedback model available to the optimizer(s).

**No-regret learning in games.**   The game-theoretic analogue of the above framework is defined as follows. We consider a finite set of players indexed by $i \in \mathcal{N} = \{1, \ldots, N\}$, each with their individual action space $\mathcal{X}^i = \mathbb{R}^{d^i}$ and their associated loss function $\ell^i \colon \mathcal{X} \to \mathbb{R}$, where $\mathcal{X} = \Pi_{i\in\mathcal{N}}\mathcal{X}^i$ denotes the game's joint action space. For clarity, any ensemble of actions or functions whose definition involves multiple players will be typeset in bold. In particular, we will write $\mathbf{x} = (x^i, \mathbf{x}^{-i}) \in \mathcal{X}$ for the action profile of all players, where $x^i$ and $\mathbf{x}^{-i}$ respectively denote the action of player $i$ and the joint action of all players other than $i$. In this way, each player $i \in \mathcal{N}$ incurs at round $t$ a loss $\ell^i(\mathbf{x}_t)$ which is determined not only by their individual action $x_t^i$, but also by the

---

    [4]We also note that under the additional assumption of *cocoercivity*, a constant regret bound can be derived from [26, Th. 4.4]. That being said, extending this result to the broader family of variationally stable games that we address here requires non-trivial modifications (to both the algorithm and the analysis).

actions $\mathbf{x}_t^{-i}$ of all other players. Thus, by drawing a direct link with (1), given a sequence of play $x_t^i$, the *individual regret* of each player $i \in \mathcal{N}$ is defined as

$$\operatorname{Reg}_T^i(p^i) = \sum_{t=1}^{T} \ell(x_t^i, \mathbf{x}_t^{-i}) - \ell(p^i, \mathbf{x}_t^{-i}), \tag{2}$$

From a static viewpoint, the most widely spread solution concept in game theory is that of a *Nash equilibrium*, i.e., a state from which no player has incentive to deviate unilaterally. Formally, a point $\mathbf{x}_\star \in \mathcal{X}$ is a Nash equilibrium if for all $i \in \mathcal{N}$ and all $x^i \in \mathcal{X}^i$, we have $\ell^i(x_\star^i, \mathbf{x}_\star^{-i}) \leq \ell^i(x^i, \mathbf{x}_\star^{-i})$. In particular, if the players' loss functions are assumed individually convex (see below), Nash equilibria coincide precisely with the zeros of the players' individual gradient field, denoted by $V^i = \nabla_{x^i} \ell^i$. That is, $\mathbf{x}_\star$ is a Nash equilibrium if and only if $\mathbf{V}(\mathbf{x}_\star) = 0$. We will make the following blanket assumptions for all this:

**Assumption 1** (Convexity and Smoothness). For all $i \in \mathcal{N}$, $\ell^i(\cdot, \mathbf{x}^{-i})$ is convex at all $\mathbf{x}^{-i}$ and the individual gradient of each player $\nabla_{x^i} \ell^i$ is $L$-Lipschitz continuous.

**Assumption 2** (Variational Stability). The solution set $\mathcal{X}_\star = \{\mathbf{x} \in \mathcal{X} : \mathbf{V}(\mathbf{x}) = 0\}$ of the game is nonempty, and for all $\mathbf{x} \in \mathcal{X}$, $\mathbf{x}_\star \in \mathcal{X}_\star$, we have $\langle \mathbf{V}(\mathbf{x}), \mathbf{x} - \mathbf{x}_\star \rangle = \sum_{i \in \mathcal{N}} \langle V^i(\mathbf{x}), x^i - x_\star^i \rangle \geq 0$.

The convexity requirement in Assumption 1 is crucial in the literature of online learning; otherwise, it is not possible to transform iterative gradient bounds to bona fide regret guarantees. In a similar vein, variational stability can be seen as a variant of the convexity assumption for multi-agent environments, where unilateral convexity assumptions do not suffice to give rise to a learnable game – for example, finite games are unilaterally linear, but finding a Nash equilibrium of a finite game is a PPAD-complete problem [9]. Our work thus focuses on games that satisfy the variational stability condition. Some important families of games that are covered by this criterion are monotone games (i.e., $\mathbf{V}$ is monotone), which in their turn include convex-concave zero-sum games, zero-sum polymatrix games, Cournot oligopolies, etc.

It is also worth noting that several recent works [1, 12, 15] have managed to bypass Assumption 2 when the players have access to perfect feedback; whether these techniques are applicable in the stochastic setup is an open question. In any case, Assumption 2 seems crucial for the last-iterate convergence presented in Section 6.

**Oracle feedback and noise models.** In terms of feedback, we will assume that players have access to noisy estimates of their individual payoff gradients, and we will consider two noise models, *additive noise* and *multiplicative noise*. To illustrate the difference between these two models, suppose we wish to estimate the value of some quantity $v \in \mathbb{R}$. Then, an estimate of $v$ with additive noise is a random variable $\hat{v}_{\text{add}}$ of the form $\hat{v}_{\text{add}} = v + \xi_{\text{add}}$ for some zero-mean noise variable $\xi_{\text{add}}$; analogously, a multiplicative noise model for $v$ is a random variable of the form $\hat{v}_{\text{mult}} = v(1 + \xi_{\text{mult}})$ for some zero-mean noise variable $\xi_{\text{mult}}$. The two models can be compared directly via the additive representation of the multiplicative noise model as $\hat{v}_{\text{mult}} = v + \xi_{\text{mult}} v$, which gives $\operatorname{Var}[\xi_{\text{add}}] = v^2 \operatorname{Var}[\xi_{\text{mult}}]$.

With all this in mind, we will consider the following oracle feedback model: let $g_t^i = V^i(\mathbf{x}_t) + \xi_t^i$ denote the gradient feedback to player $i$ at round $t$, where $\xi_t^i$ represents the aggregate measurement error relative to $V^i(\mathbf{x}_t)$. Then, with $(\mathcal{F}_t)_{t \in \mathbb{N}}$ denoting the natural filtration associated to $(\mathbf{x}_t)_{t \in \mathbb{N}}$ and $\mathbb{E}_t[\cdot] = \mathbb{E}[\cdot \mid \mathcal{F}_t]$ representing the corresponding conditional expectation, we make the following standard assumption for the measurement error vector $\boldsymbol{\xi}_t = (\xi_t^i)_{i \in \mathcal{N}}$.

**Assumption 3.** The noise vector $(\boldsymbol{\xi}_t)_{t \in \mathbb{N}}$ satisfies the following requirements for some $\sigma_A, \sigma_M \geq 0$.

(a) *Zero-mean:* For all $i \in \mathcal{N}$ and $t \in \mathbb{N}$, $\mathbb{E}_t[\xi_t^i] = 0$.

(b) *Finite variance:* For all $i \in \mathcal{N}$ and $t \in \mathbb{N}$, $\mathbb{E}_t[\|\xi_t^i\|^2] \leq \sigma_A^2 + \sigma_M^2 \|V^i(\mathbf{x}_t)\|^2$.

As an example of the above, the case $\sigma_A, \sigma_M = 0$ corresponds to "perfect information", i.e., when players have full access to their payoff gradients. The case $\sigma_A > 0$, $\sigma_M = 0$, is often referred to as "absolute noise", and it is a popular context-agnostic model for stochastic first-order methods, cf. [21, 29] and references therein. Conversely, the case $\sigma_A = 0$, $\sigma_M > 0$, is sometimes called "relative noise" [33], and it is widely used as a model for randomized coordinate descent methods [30], randomized player updates in game theory [4], physical measurements in signal processing and control [36], etc. In the sequel, we will treat both models concurrently, and we will use the term "noise" to tacitly refer to the presence of both additive and multiplicative components.

# 3 Optimistic gradient methods: Definitions, difficulties, and a test case

To illustrate some of the difficulties faced by first-order methods in a game-theoretic setting, consider the standard bilinear problem $\min_{\theta\in\mathbb{R}}\max_{\phi\in\mathbb{R}}\theta\phi$, i.e., $\ell^1(\theta,\phi)=\theta\phi=-\ell^2(\theta,\phi)$. This simple game has a unique Nash equilibrium at $(0,0)$ but, despite this uniqueness, it is well known that standard gradient descent/ascent methods diverge on this simple problem [11, 28]. To remedy this failure, one popular solution consists of incorporating an additional *extrapolation step* at each iteration of the algorithm, leading to the *optimistic gradient* (OG) method

$$x^i_{t+1} = x^i_t - 2\eta^i_{t+1}g^i_t + \eta^i_t g^i_{t-1},$$

where $\eta^i_t$ is player $i$'s learning rate at round $t$. For posterity, it will be convenient to introduce the auxiliary iterate $X^i_t$ and write $X^i_{t+\frac{1}{2}} = x^i_t$ for the actual sequence of actions. The above update rule then becomes

$$X^i_{t+\frac{1}{2}} = X^i_t - \eta^i_t g^i_{t-1}, \qquad X^i_{t+1} = X^i_t - \eta^i_{t+1}g^i_t. \tag{OG}$$

This form of the algorithm effectively decouples the learner's *extrapolation step* (performed with $g^i_{t-1}$, which acts here as an *optimistic* guess for the upcoming feedback), and the bona fide *update step*, which exploits the received feedback $g^i_t$ to update the player's action state from $X^i_t$ to $X^i_{t+1}$. This mechanism helps the players attain *a*) *lower regret* when their utilities vary slowly (from an online learning viewpoint) [8, 35]; and *b*) *near-constant regret* when all players employ the said algorithm in certain classes of games [1, 2, 12, 16, 20].

However, the above guarantees concern only the case of *perfect gradient feedback*, and may fail completely when the feedback is contaminated by noise, as illustrated in the following example.

**Example 1.** Suppose that the game's objective is an expectation over $\mathcal{L}_1(\theta,\phi) = 3\theta\phi$ and $\mathcal{L}_2(\theta,\phi) = -\theta\phi$ so that $\ell^1 = -\ell^2 = (\mathcal{L}_1+\mathcal{L}_2)/2$. At each round, we randomly draw $\mathcal{L}_1$ or $\mathcal{L}_2$ with probability $1/2$ and return the gradient of the sampled function as feedback. Assumption 3 is clearly satisfied here with $\sigma_A = 0$ and $\sigma_M = 2$, i.e., the noise is multiplicative; however, as shown in Figure 1, running (OG) with either constant or decreasing learning rate leads to *i*) divergent trajectories of play; and *ii*) regret oscillations that grow linearly or even superlinearly in magnitude over time.[5]

In view of the above negative results, we propose in the next section a simple fix of the algorithm that allows us to retain its constant regret guarantees in the presence of multiplicative noise.

# 4 Regret minimization with noisy feedback

In this section, we introduce OG+ and OptDA+, our backbone algorithms for learning under uncertainty, and we present their guarantees in different settings. All proofs are deferred to the appendix.

**Learning rate separation and the role of averaging.** Viewed abstractly, the failure of OG in the face of uncertainty should be attributed to its inability of separating noise from the expected variation of utilities. In fact, in a noisy environment, the two consecutive pieces of feedback are only close *in expectation*, so a player can only exploit this similarity when the noise is mitigated appropriately.

To overcome this difficulty, we adopt a learning rate separation strategy originally proposed for the EG algorithm by Hsieh et al. [19]. The key observation here is that by taking a larger extrapolation step, the noise effectively becomes an order of magnitude smaller relative to the expected variation of utilities. We refer to this generalization of OG as OG+, and we define it formally as

$$X^i_{t+\frac{1}{2}} = X^i_t - \gamma^i_t g^i_{t-1}, \qquad X^i_{t+1} = X^i_t - \eta^i_{t+1}g^i_t, \tag{OG+}$$

where $\gamma^i_t \geq \eta^i_t > 0$ are the player's learning rates (assumed $\mathcal{F}_{t-1}$-measurable throughout the sequel). Nonetheless, the design of OG+ is somehow counter-intuitive because the players' feedback enters the algorithm with decreasing weights. This feature opens up the algorithm to adversarial attacks that can drive it to a suboptimal regime in early iterations, as formally shown in [32, Thm. 3].

To circumvent this issue, we also consider a *dual averaging* variant of OG+ that we refer to as OptDA+, and which treats the gradient feedback used to update the players' chosen actions with the

---

[5]By superlinear we mean that the regret grows faster than $\Theta(T)$, and this is possible here because neither the action set nor the feedback magnitude is bounded.

same weight. Specifically, OptDA+ combines the mechanisms of optimism [11, 35], dual averaging [20, 31, 38], and learning rate separation [19] as follows

$$X_{t+\frac{1}{2}}^i = X_t^i - \gamma_t^i g_{t-1}^i, \quad X_{t+1}^i = X_1^i - \eta_{t+1}^i \sum_{s=1}^t g_s^i. \quad \text{(OptDA+)}$$

As we shall see below, these mechanisms dovetail in an efficient manner and allow the algorithm to achieve sublinear regret even in the adversarial regime. [Of course, OG+ and OptDA+ coincide when the update learning rate $\eta_t^i$ is taken constant.]

**Quasi-descent inequality.** Before stating our main results on the regret incurred by OG+ and OptDA+, we present the key quasi-descent inequality that underlies our analysis, as it provides theoretical evidence on how the separation of learning rates can lead to concrete performance benefits.

**Lemma 1.** *Let Assumptions 1 and 3 hold and all players run either* (OG+) *or* (OptDA+) *with non-increasing learning rate sequences $\gamma_t^i$, $\eta_t^i$. Then, for all $i \in \mathcal{N}$, $t \geq 2$, and $p^i \in \mathcal{X}^i$, we have*

$$\mathbb{E}_{t-1}\left[\frac{\|X_{t+1}^i - p^i\|^2}{\eta_{t+1}^i}\right] \leq \mathbb{E}_{t-1}\left[\frac{\|X_t^i - p^i\|^2}{\eta_t^i} + \left(\frac{1}{\eta_{t+1}^i} - \frac{1}{\eta_t^i}\right)\|u_t^i - p^i\|^2\right. \tag{3a}$$

$$-2\langle V^i(\mathbf{X}_{t+\frac{1}{2}}), X_{t+\frac{1}{2}}^i - p^i\rangle \tag{3b}$$

$$-\gamma_t^i(\|V^i(\mathbf{X}_{t+\frac{1}{2}})\|^2 + \|V^i(\mathbf{X}_{t-\frac{1}{2}})\|^2) \tag{3c}$$

$$-\|X_t^i - X_{t+1}^i\|^2/2\eta_t^i + \gamma_t^i\|V^i(\mathbf{X}_{t+\frac{1}{2}}) - V^i(\mathbf{X}_{t-\frac{1}{2}})\|^2 \tag{3d}$$

$$\left.+(\gamma_t^i)^2 L\|\xi_{t-\frac{1}{2}}^i\|^2 + L\|\boldsymbol{\xi}_{t-\frac{1}{2}}\|_{(\boldsymbol{\eta}_t+\boldsymbol{\gamma}_t)^2}^2 + 2\eta_t^i\|g_t^i\|^2\right], \tag{3e}$$

*where i) $\|\boldsymbol{\xi}_{t-\frac{1}{2}}\|_{(\boldsymbol{\eta}_t+\boldsymbol{\gamma}_t)^2}^2 := \sum_{j=1}^N (\eta_t^j + \gamma_t^j)^2\|\xi_{t-\frac{1}{2}}^j\|^2$, and ii) $u_t^i = X_t^i$ if player $i$ runs* (OG+) *and $u_t^i = X_1^i$ if player $i$ runs* (OptDA+).

Lemma 1 indicates how the (weighted) distance between the player's chosen actions and a fixed benchmark action evolves over time. In order to provide some intuition on how this inequality will be used to derive our results, we sketch below the role that each term plays in our analysis.

1. Thanks to the convexity of the players' loss functions, the regret of each player can be bounded by the sum of the pairing terms in (3b). On the other hand, taking $\mathbf{x}_\star \in \mathcal{X}_\star$, $p^i = x_\star^i$, and summing from $i = 1$ to $N$, we obtain $-2\langle \mathbf{V}(\mathbf{X}_{t+\frac{1}{2}}), \mathbf{X}_{t+\frac{1}{2}} - \mathbf{x}_\star\rangle$, which is non-positive by Assumption 2, and can thus be dropped from the inequality.

2. The weighted squared distance to $p^i$, i.e., $\|X_t^i - p^i\|^2/\eta_t^i$, telescopes when controlling the regret (Section 4) and serves as a Lyapunov function for equilibrium convergence (Section 6).

3. The negative term in (3c) provides a consistent negative drift that partially cancels out the noise.

4. The difference in (3d) can be bounded using the smoothness assumption and leaves out terms that are in the order of $\gamma_t^i(\gamma_t^j)^2$.

5. Line (3e) contains a range of positive terms of the order $(\gamma_t^j)^2 + \eta_t^i$. To ensure that they are sufficiently small with respect to the decrease of (3c), both $(\gamma_t^j)_{j\in\mathcal{N}}$ and $\eta_t^i/\gamma_t^i$ should be small. Applying Assumption 3 gives $\mathbb{E}[\|g_t^i\|^2] \leq \mathbb{E}[(1 + \sigma_M^2)\|V^i(\mathbf{X}_{t+\frac{1}{2}})\| + \sigma_A^2]$, revealing that $\gamma_t^i/\eta_t^i$ needs to be at least in the order of $(1 + \sigma_M^2)$.

6. Last but not least, $(1/\eta_{t+1}^i - 1/\eta_t^i)\|u_t^i - p^i\|^2$ simply telescopes for OptDA+ (in which case $u_t^i = X_1^i$) but is otherwise difficult to control for OG+ when $\eta_{t+1}^i$ differs from $\eta_t^i$. This additional difficulty forces us to use a global learning rate common across all players when analysing OG+.

To summarize, OG+ and OptDA+ are more suitable for learning in games with noisy feedback because the scale separation between the extrapolation and the update steps delivers a consistent negative drift (3c) that is an order of magnitude greater relative to the deleterious effects of the noise. We will exploit this property to derive our main results for OG+ and OptDA+ below.

**Constant regret under uncertainty.** We are now in a position to state our regret guarantees:

**Theorem 1.** *Suppose that Assumptions 1–3 hold and all players run* (OG+) *with non-increasing learning rate sequences $\gamma_t$ and $\eta_t$ such that*

$$\gamma_t \leq \min\left(\frac{1}{3L\sqrt{2N(1+\sigma_M^2)}}, \frac{1}{2(4N+1)L\sigma_M^2}\right) \quad and \quad \eta_t \leq \frac{\gamma_t}{2(1+\sigma_M^2)} \quad for\ all\ t \in \mathbb{N}. \quad (4)$$

*Then, for all $i \in \mathcal{N}$ and all $p^i \in \mathcal{X}^i$, we have*

(a) *If $\gamma_t = \mathcal{O}(1/(t^{\frac{1}{4}}\sqrt{\log t}))$ and $\eta_t = \Theta(1/(\sqrt{t}\log t))$, then $\mathbb{E}\left[\operatorname{Reg}_T^i(p^i)\right] = \tilde{\mathcal{O}}(\sqrt{T})$.*

(b) *If the noise is multiplicative and the learning rates are constant, then $\mathbb{E}\left[\operatorname{Reg}_T^i(p^i)\right] = \mathcal{O}(1)$.*

The first part of Theorem 1 guarantees the standard $\tilde{\mathcal{O}}(\sqrt{T})$ regret in the presence of additive noise, in accordance with existing results in the literature. What is far more surprising is the second part of Theorem 1 which shows that when the noise is multiplicative (i.e., when $\sigma_A = 0$), it is *still* possible to achieve constant regret. This represents a dramatic improvement in performance, which we illustrate in Figure 1: by simply taking the extrapolation step to be 10 times larger, the player's regret becomes completely stabilized. In this regard, Theorem 1 provides fairly conclusive evidence that having access to exact gradient payoffs *is not* an absolute requisite for achieving constant regret in a game-theoretic context.

On the downside, the above result requires all players to use the same learning rate sequences, a technical difficulty that we overcome below by means of the dual averaging mechanism of OptDA+.

**Theorem 2.** *Suppose that Assumptions 1–3 hold and all players run* (OptDA+) *with non-increasing learning rate sequences $\gamma_t^i$ and $\eta_t^i$ such that*

$$\gamma_t^i \leq \frac{1}{2L}\min\left(\frac{1}{\sqrt{3N(1+\sigma_M^2)}}, \frac{1}{(4N+1)\sigma_M^2}\right) \quad and \quad \eta_t^i \leq \frac{\gamma_t^i}{4(1+\sigma_M^2)} \quad for\ all\ t \in \mathbb{N},\ i \in \mathcal{N}. \tag{5}$$

*Then, for any $i \in \mathcal{N}$ and $p^i \in \mathcal{X}^i$, we have:*

(a) *If $\gamma_t^j = \mathcal{O}(1/t^{\frac{1}{4}})$ and $\eta_t^j = \Theta(1/\sqrt{t})$ for all $j \in \mathcal{N}$, then $\mathbb{E}\left[\operatorname{Reg}_T^i(p^i)\right] = \mathcal{O}(\sqrt{T})$.*

(b) *If the noise is multiplicative and the learning rates are constant, then $\mathbb{E}\left[\operatorname{Reg}_T^i(p^i)\right] = \mathcal{O}(1)$.*

The similarity between Theorems 1 and 2 suggests that OptDA+ enjoys nearly the same regret guarantee as OG+ while allowing for the use of *player-specific* learning rates. As OptDA+ and OG+ coincide when run with constant learning rates, Theorem 1(b) is in fact a special case of Theorem 2(b). However, when the algorithms are run with decreasing learning rates, they actually lead to different trajectories. In particular, when the feedback is corrupted by additive noise, this difference translates into the removal of logarithmic factors in the regret bound. More importantly, as we show below, it also helps to achieve sublinear regret when the opponents do not follow the same learning strategy, i.e., in the fully arbitrary, adversarial case.

**Proposition 1.** *Suppose that Assumption 3 holds and player $i$ runs* (OptDA+) *with non-increasing learning rates $\gamma_t^i = \Theta(1/t^{\frac{1}{2}-q})$ and $\eta_t^i = \Theta(1/\sqrt{t})$ for some $q \in [0, 1/4]$. If $\sup_{x^i \in \mathcal{X}^i}\|V^i(x^i)\| < +\infty$, we have $\mathbb{E}[\operatorname{Reg}_T^i(p^i)] = \mathcal{O}(T^{\frac{1}{2}+q})$ for every benchmark action $p^i \in \mathcal{X}^i$.*

We introduce the exponent $q$ in Proposition 1 because, as suggested by Theorem 2(a), the whole range of $q \in [0, 1/4]$ leads to the optimal $\mathcal{O}(\sqrt{T})$ regret bound for additive noise when all the players adhere to the use of OptDA+. However, it turns out that taking smaller $q$ (i.e., smaller extrapolation step) is more favorable in the adversarial regime. This is because arbitrarily different successive feedback may make the extrapolation step harmful rather than helpful. On the other hand, our previous discussion also suggests that taking larger $q$ (i.e., larger extrapolation steps), should be more beneficial when all the players use OptDA+. We will quantify this effect in Section 6; however, before doing so, we proceed in the next section to show how the learning rates of Proposition 1 can lead to the design of a fully adaptive, parameter-agnostic algorithm.

## 5 Adaptive learning rates

So far, we have focused exclusively on algorithms run with *predetermined* learning rates, whose tuning requires knowledge of the various parameters of the model. Nonetheless, even though a player

might be aware of their own loss function, there is little hope that the noise-related parameters are also known by the player. Our goal in this section will be to address precisely this issue through the design of *adaptive* methods enjoying the following desirable properties:

- The method should be implementable by every individual player using only local information and without any prior knowledge of the setting's parameters (for the noise profile and the game alike).
- The method should guarantee sublinear individual regret against any bounded feedback sequence.
- When employed by all players, the method should guarantee $\mathcal{O}(\sqrt{T})$ regret under additive noise and $\mathcal{O}(1)$ regret under multiplicative noise.

In order to achieve the above, inspired by the learning rate requirements of Theorem 2 and Proposition 1, we fix $q \in (0, 1/4]$ and consider the following Adagrad-style [13] learning rate schedule.

$$\gamma_t^i = \frac{1}{\left(1 + \sum_{s=1}^{t-2}\|g_s^i\|^2\right)^{\frac{1}{2}-q}}, \quad \eta_t^i = \frac{1}{\sqrt{1 + \sum_{s=1}^{t-2}\left(\|g_s^i\|^2 + \|X_s^i - X_{s+1}^i\|^2\right)}}. \tag{Adapt}$$

As in Adagrad, the sum of the squared norm of the feedback appears in the denominator. This helps controlling the various positive terms appearing in Lemma 1, such as $L\|\boldsymbol{\xi}_{t-\frac{1}{2}}\|_{(\boldsymbol{\eta}_t + \boldsymbol{\gamma}_t)^2}^2$ and $2\eta_t^i\|g_t^i\|^2$. Nonetheless, this sum is not taken to the same exponent in the definition of the two learning rates. This scale separation ensures that the contribution of the term $-\gamma_t^i\|V^i(\mathbf{X}_{t+\frac{1}{2}})\|^2$ appearing in (3c) remains negative, and it is the key for deriving constant regret under multiplicative noise. As a technical detail, the term $\|X_s^i - X_{s+1}^i\|^2$ is involved in the definition of $\eta_t^i$ for controlling the difference of (3d). Finally, we do not include the previous received feedback $g_{t-1}^i$ in the definition of $\gamma_t^i$ and $\eta_t^i$. This makes these learning rates $\mathcal{F}_{t-1}$-measurable, which in turn implies $\mathbb{E}[\gamma_t^i\eta_t^i\xi_{t-\frac{1}{2}}^i] = 0$.

From a high-level perspective, the goal with (Adapt) is to recover automatically the learning rate schedules of Theorem 2. This in particular means that $\gamma_t^i$ and $\eta_t^i$ should at least be in the order of $\Omega(1/t^{\frac{1}{2}-q})$ and $\Omega(1/\sqrt{t})$, suggesting the following boundedness assumptions on the feedback.

**Assumption 4.** There exists $G, \bar{\sigma} \geq 0$ such that *i)* $\|V^i(x^i)\| \leq G$ for all $i \in \mathcal{N}$, $x^i \in \mathcal{X}^i$; and *ii)* $\|\xi_t^i\| \leq \bar{\sigma}$ for all $i \in \mathcal{N}$, $t \in \mathbb{N}$ with probability 1.

These assumptions are standard in the literature on adaptive methods, cf. [3, 5, 14, 24].

**Regret.** We begin with the method's fallback guarantees, deferring all proofs to the appendix.

**Proposition 2.** *Suppose that Assumption 4 holds and a player $i \in \mathcal{N}$ follows (OptDA+) with learning rates given by (Adapt). Then, for any benchmark action $p^i \in \mathcal{X}^i$, we have $\mathbb{E}[\mathrm{Reg}_T^i(p^i)] = \mathcal{O}(T^{\frac{1}{2}+q})$.*

Proposition 2 provides exactly the same rate as Proposition 1, illustrating in this way the benefit of taking a smaller $q$ for achieving smaller regret against adversarial opponents. Nonetheless, as we see below, taking smaller $q$ may incur higher regret when adaptive OptDA+ is employed by all players. In particular, we require $q > 0$ in order to obtain constant regret under multiplicative noise, and this prevents us from obtaining the optimal $\mathcal{O}(\sqrt{T})$ regret in fully adversarial environments.

**Theorem 3.** *Suppose that Assumptions 1–4 hold and all players run (OptDA+) with learning rates given by (Adapt). Then, for any $i \in \mathcal{N}$ and point $p^i \in \mathcal{X}^i$, we have $\mathbb{E}[\mathrm{Reg}_T^i(p^i)] = \mathcal{O}(\sqrt{T})$. Moreover, if the noise is multiplicative ($\sigma_A = 0$), we have $\mathbb{E}[\mathrm{Reg}_T^i(p^i)] = \mathcal{O}(\exp(1/(2q)))$.*

The proof of Theorem 3 is based on Lemma 1; we also note that the $\mathcal{O}(\sqrt{T})$ regret guarantee can in fact be derived for any $q \leq 1/4$ (even negative ones). The main difficulty here consists in bounding (3d), which does not directly cancel out since $\gamma_t^i\eta_t^i$ might not be small enough. To overcome this challenge, we have involved the squared difference $\|X_s^i - X_{s+1}^i\|^2$ in the definition of $\eta_t^i$ so that the sum of these terms cannot be too large when $\eta_t^i$ is not small enough. More details on this aspect can be found in the proof of Lemma 18 in the appendix.

Importantly, the $\mathcal{O}(\sqrt{T})$ guarantee above does not depend on the choice of $q$. This comes in sharp contrast to the constant regret bounds (in $T$) that we obtain for multiplicative noise. In fact, a key step for proving this is to show that for some (environment-dependent) constant $C$, we have

$$\sum_{i=1}^N \mathbb{E}\left[\left(1 + \sum_{s=1}^t\|g_s^i\|^2\right)^{\frac{1}{2}+q}\right] \leq C\sum_{i=1}^N \mathbb{E}\left[\sqrt{1 + \sum_{s=1}^t\|g_s^i\|^2}\right] \qquad \text{for all } t \in \mathbb{N} \tag{6}$$

This inequality is derived from Lemma 1 by carefully bounding (3a), (3d), (3e) from above and bounding (3b), (3c) from below. Applying Jensen's inequality, we then further deduce that the right-hand side of inequality (6) is bounded by some constant. This constant, however, is exponential in $1/q$. This leads to an inherent trade-off in the choice of $q$: larger values of $q$ favor the situation where all players adopt adaptive OptDA+ under multiplicative noise, while smaller values of $q$ provide better fallback guarantees in adversarial environments.

## 6 Trajectory analysis

In this section, we shift our focus to the analysis of the joint trajectory of play when all players follow the same learning strategy. We derive the convergence of the trajectory of play induced by the algorithms (cf. Figure 1) and provide bounds on the sum of the players' payoff gradient norms $\sum_{t=1}^{T} \|\mathbf{V}(\mathbf{X}_{t+\frac{1}{2}})\|^2$. This may be regarded as a relaxed convergence criterion, and by the design of the algorithms, a feedback sequence of smaller magnitude also suggests a more stable trajectory.

**Convergence of trajectories under multiplicative noise.** When the noise is multiplicative, its effect is in expectation absorbed by the progress brought by the extrapolation step. We thus expect convergence results that are similar to the noiseless case. This is confirmed by the following theorem.

**Theorem 4.** *Suppose that Assumptions 1–3 hold with $\sigma_A = 0$ and all players run (OG+) / (OptDA+) with learning rates given in Theorem 2(b).[6] Then, $\mathbf{X}_{t+\frac{1}{2}}$ converges almost surely to a Nash equilibrium and enjoys the stabilization guarantee $\sum_{t=1}^{+\infty} \mathbb{E}[\|\mathbf{V}(\mathbf{X}_{t+\frac{1}{2}})\|^2] < +\infty$.*

*Idea of proof.* The proof of Theorem 4 follows the following steps.

1. We first show $\sum_{t=1}^{+\infty} \mathbb{E}[\|\mathbf{V}(\mathbf{X}_{t+\frac{1}{2}})\|^2] < +\infty$ using Lemma 1. This implies $\sum_{t=1}^{\infty} \|\mathbf{V}(\mathbf{X}_{t+\frac{1}{2}})\|^2$ is finite almost surely, and thus with probability 1, $\|\mathbf{V}(\mathbf{X}_{t+\frac{1}{2}})\|$ converges to 0 and all cluster point of $(\mathbf{X}_{t+\frac{1}{2}})_{t \in \mathbb{N}}$ is a solution.

2. Applying the Robbins–Siegmund theorem to a suitable quasi-descent inequality then gives the almost sure convergence of $\mathbb{E}_{t-1}[\sum_{i \in \mathcal{N}} \|X_t^i - x_\star^i\|^2 / \eta^i]$ to finite value for any $\mathbf{x}_\star \in \mathcal{X}_\star$.

3. The conditioning on $\mathcal{F}_{t-1}$ makes the above quantity not directly amenable to analysis. This difficulty is specific to the optimistic algorithms that we consider here as they make use of past feedback in each iteration. We overcome this issue by introducing a virtual iterate $\tilde{\mathbf{X}}_t = (\tilde{X}_t^i)_{i \in \mathcal{N}}$ with $\tilde{X}_t^i = X_t^i + \eta^i \xi_{t-\frac{1}{2}}^i$ that serves as a $\mathcal{F}_{t-1}$-measurable surrogate for $X_t^i$. We then derive the almost sure convergence of $\sum_{i \in \mathcal{N}} \|\tilde{X}_t^i - x_\star^i\|^2 / \eta^i$.

4. To conclude, along with the almost sure convergence of $\|\mathbf{X}_{t+\frac{1}{2}} - \tilde{\mathbf{X}}_t\|$ and $\|\mathbf{V}(\mathbf{X}_{t+\frac{1}{2}})\|$ to 0 we derive the almost sure convergence of $\mathbf{X}_{t+\frac{1}{2}}$ to a Nash equilibrium. $\square$

In case where the players run the adaptive variant of OptDA+, we expect the learning rates to behave as constants asymptotically and thus similar reasoning can still apply. Formally, we show in the appendix that under multiplicative noise the learning rates of the players converge almost surely to positive constants, and prove the following results concerning the induced trajectory.

**Theorem 5.** *Suppose that Assumptions 1–4 hold with $\sigma_A = 0$ and all players run (OptDA+) with learning rates (Adapt). Then, i) $\sum_{t=1}^{+\infty} \|\mathbf{V}(\mathbf{X}_{t+\frac{1}{2}})\|^2 < +\infty$ with probability 1, and ii) $\mathbf{X}_{t+\frac{1}{2}}$ converges almost surely to a Nash equilibrium.*

Compared to Theorem 4, we can now only bound $\sum_{t=1}^{\infty} \|\mathbf{V}(\mathbf{X}_{t+\frac{1}{2}})\|^2$ in an almost sure sense. This is because in the case of adaptive learning rates, our proof relies on inequality (6), and deriving a bound on $\sum_{t=1}^{\infty} \mathbb{E}[\|\mathbf{V}(\mathbf{X}_{t+\frac{1}{2}})\|^2]$ from this inequality does not seem possible. Nonetheless, with the almost sure convergence of the learning rates to positive constants, we still manage to prove almost sure last-iterate convergence of the trajectory of play towards a Nash equilibrium.

Such last-iterate convergence results for adaptive methods are relatively rare in the literature, and most of them assume perfect oracle feedback. To the best of our knowledge, the closest antecedents

---

[6]Recall that OG+ and OptDA+ are equivalent when run with constant learning rates.

to our result are [2, 26], but both works make the more stringent cocoercive assumptions and consider adaptive learning rate that is the same for all the players. In particular, their learning rates are computed with global feedback and are thus less suitable for the learning-in-game setup.

**Convergence of trajectories under additive noise.** To ensure small regret under additive noise, we take vanishing learning rates. This makes the analysis much more difficult as the term $(1/\eta_{t+1}^i - 1/\eta_t^i)\|u_t^i - p^i\|^2$ appearing on the right-hand side of inequality (3a) is no longer summable. Nonetheless, it is still possible to provide bound on the sum of the squared operator norms.

**Theorem 6.** *Suppose that Assumptions 1–3 hold and either i) all players run (OG+) with learning rates described in Theorem 1(a) and $\gamma_t = \Omega(1/t^{\frac{1}{2}-q})$ for some $q \in [0, 1/4]$; ii) all players run (OptDA+) with learning rates described in Theorem 2(a) and $\gamma_t^i = \Omega(1/t^{\frac{1}{2}-q})$ for all $i \in \mathcal{N}$ for some $q \in [0, 1/4]$; or iii) all players run (OptDA+) with learning rates (Adapt) and Assumption 4 holds. Then, $\sum_{t=1}^T \mathbb{E}[\|\mathbf{V}(\mathbf{X}_{t+\frac{1}{2}})\|^2] = \tilde{\mathcal{O}}(T^{1-q})$.*

Theorem 6 suggests that the convergence speed of $\|\mathbf{V}(\mathbf{X}_{t+\frac{1}{2}})\|^2$ under additive noise actually depends on $q$. Therefore, though the entire range of $q \in [0, 1/4]$ leads to $\mathcal{O}(\sqrt{T})$ regret, taking larger $q$ may result in a more stabilized trajectory. This again goes against Propositions 1 and 2, which suggests smaller $q$ leads to smaller regret in the face of adversarial opponents.

Finally, we also show last-iterate convergence of the trajectory of OG+ under additive noise.

**Theorem 7.** *Suppose that Assumptions 1–3 hold and all players run (OG+) with non-increasing learning rate sequences $\gamma_t$ and $\eta_t$ satisfying (4) and $\gamma_t = \Theta(1/(t^{\frac{1}{2}-q}\sqrt{\log t}))$, $\eta_t = \Theta(1/(\sqrt{t}\log t))$ for some $q \in (0, 1/4]$. Then, $\mathbf{X}_t$ converges almost surely to a Nash equilibrium. Moreover, if $\sup_{t \in \mathbb{N}} \mathbb{E}[\|\boldsymbol{\xi}_t\|^4] < +\infty$, then $\mathbf{X}_{t+\frac{1}{2}}$ converges almost surely to a Nash equilibrium.*

Theorem 7, in showing that the sequence $\mathbf{X}_t$ generated by OG+ converges under suitable learning rates, resolves an open question of [19]. By contrast, the analysis of OG+ is much more involved due to the use of past feedback, as explained in the proof of Theorem 4. Going further, in the second part of statement, we show that $\mathbf{X}_{t+\frac{1}{2}}$ also converges to a Nash equilibrium as long as the 4-th moment of the noise is bounded. Compared to OptDA+, it is possible to show last-iterate convergence for OG+ under additive noise because we can use $\mathbb{E}_{t-1}[\|\mathbf{X}_t - \mathbf{x}_\star\|^2]$ (with $\mathbf{x}_\star \in \mathcal{X}_\star$) as a Lyapunov function. The same strategy does not apply to OptDA+ due to summability issues. This is a common challenge shared by trajectory convergence analysis of the dual averaging template under additive noise.

## 7 Concluding remarks

In this paper, we look into the fundamental problem of no-regret learning in games under uncertainty. We exhibited algorithms that enjoy constant regret under multiplicative noise. Building upon this encouraging result, we further studied an adaptive variant and proved trajectory convergence of the considered algorithms. A central element that is ubiquitous in our work is the trade-off between robustness in the fully adversarial setting and faster convergence in the game-theoretic case, as encoded by the exponent $q$. Whether this trade-off is inherent to the problem or an artifact of the algorithm design warrants further investigation.

Moving forward, there are many important problems that remain to be addressed. On the technical side, a first goal would be to deepen our understanding on the convergence behavior of OptDA+ under additive noise. Extension of our results to learning in other type of games and/or under different types of uncertainty – such as learning in finite games with sampling- or payoff-based feedback – would likewise be a valuable contribution. Going one step further, analyzing the situation where only a fraction of players deviate is practically relevant (the cases studied in this paper represent the extreme of this spectrum). Taking into account other type of regret that may be more suitable for game-theoretic settings is yet another fruitful research direction to pursue.

## Acknowledgments

This work received financial support from MIAI@Grenoble Alpes (ANR-19-P3IA-0003), the European Research Council (ERC) under the European Union's Horizon 2020 research and innovation program (grant agreement n° 725594 - time-data), and the Swiss National Science Foundation (SNSF) under grant number 200021_205011. P. Mertikopoulos was also supported by the grant ALIAS (ANR-19-CE48-0018-01).

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
