# No-regret learning in games with noisy feedback: Faster rates and adaptivity via learning rate separation

**Yu-Guan Hsieh**
Univ. of Grenoble Alpes
yu-guan.hsieh@univ-grenoble-alpes.fr

**Kimon Antonakopoulos**
EPFL
kimon.antonakopoulos@epfl.ch

**Volkan Cevher**
EPFL
volkan.cevher@epfl.ch

**Panayotis Mertikopoulos**
Univ. Grenoble Alpes, CNRS, LIG & Criteo AI Lab

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

As we shall see below, these mechanisms dovetail in an efficient manner and allow the algorithm to achieve sublinear regret even in the adversarial regime. [Of course, OG+ and OptDA+ coincide when the update learning rate $\eta^i_t$ is taken constant.]

**Quasi-descent inequality.** Before stating our main results on the regret incurred by OG+ and OptDA+, we present the key quasi-descent inequality that underlies our analysis, as it provides theoretical evidence on how the separation of learning rates can lead to concrete performance benefits.

**Lemma 1.** *Let Assumptions 1 and 3 hold and all players run either (OG+) or (OptDA+) with non-increasing learning rate sequences $\gamma^i_t, \eta^i_t$. Then, for all $i \in \mathcal{N}$, $t \geq 2$, and $p^i \in \mathcal{X}^i$, we have*

$$\mathbb{E}_{t-1}\left[\frac{\|X^i_{t+1} - p^i\|^2}{\eta^i_{t+1}}\right] \leq \mathbb{E}_{t-1}\left[\frac{\|X^i_t - p^i\|^2}{\eta^i_t} + \left(\frac{1}{\eta^i_{t+1}} - \frac{1}{\eta^i_t}\right)\|u^i_t - p^i\|^2\right. \tag{3a}$$

$$-2\langle V^i(\mathbf{X}_{t+\frac{1}{2}}), X^i_{t+\frac{1}{2}} - p^i\rangle \tag{3b}$$

$$-\gamma^i_t(\|V^i(\mathbf{X}_{t+\frac{1}{2}})\|^2 + \|V^i(\mathbf{X}_{t-\frac{1}{2}})\|^2) \tag{3c}$$

$$-\|X^i_t - X^i_{t+1}\|^2/2\eta^i_t + \gamma^i_t\|V^i(\mathbf{X}_{t+\frac{1}{2}}) - V^i(\mathbf{X}_{t-\frac{1}{2}})\|^2 \tag{3d}$$

$$\left. + (\gamma^i_t)^2 L\|\xi^i_{t-\frac{1}{2}}\|^2 + L\|\boldsymbol{\xi}_{t-\frac{1}{2}}\|^2_{(\boldsymbol{\eta}_t + \boldsymbol{\gamma}_t)^2} + 2\eta^i_t\|g^i_t\|^2\right], \tag{3e}$$

*where i) $\|\boldsymbol{\xi}_{t-\frac{1}{2}}\|^2_{(\boldsymbol{\eta}_t + \boldsymbol{\gamma}_t)^2} := \sum_{j=1}^{N}(\eta^j_t + \gamma^j_t)^2\|\xi^j_{t-\frac{1}{2}}\|^2$, and ii) $u^i_t = X^i_t$ if player $i$ runs (OG+) and $u^i_t =

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

# Appendix

**Table of Contents**

# A  Prelude

The appendix is organized as follows. In Appendix B we complement our introduction with an overview on other related works. In Appendix C we expand on our plots for better visibility. We also provide some additional figures there. Subsequently, we build toward the proofs of our main results in Appendices D–H. Appendix D introduces the notations used in the proofs. Some technical details concerning the measurability of the noises and learning rates are discussed as well. Appendix E contains elementary energy inequalities that are repeatedly used through out our analysis. Appendices F and G are dedicated to the regret analysis of the non-adaptive and the adaptive variants. Bounds on the expectation of the sum of the squared operator norms $\sum_{t=1}^{T}\|\mathbf{V}(\mathbf{X}_{t+\frac{1}{2}})\|^2$ are also established in these two sections, as bounding this quantity often consists in an important step for bounding the regret. Finally, proofs on the trajectory convergence are presented in Appendix H.

Importantly, in the appendix we present our results in a way that fits better the analysis. Hence, both the organization and the ordering of the these results differ from those in the main paper. For the ease of the reader, we summarize below how the results in the appendix correspond to those in the main paper.

| Results of main paper | Results of appendix |
| --- | --- |
| Lemma 1 | Lemma 4; Lemma 6 |
| Theorem 1 | Theorem 9; Lemma 8 |
| Theorem 2 | Theorem 11; Lemma 8 |
| Theorem 3 | Theorem 13; Theorem 14; Lemma 8 |
| Theorem 4 | Theorem 10 (b); Theorem 17 |
| Theorem 5 | Theorem 18 |
| Theorem 6 | Theorem 8 (a); Theorem 10 (a); Theorem 12 |
| Theorem 7 | Theorem 15; Theorem 16 |
| Proposition 1 | Proposition 7; Lemma 8 |
| Proposition 2 | Proposition 8; Lemma 8 |

**Table 2:** Correspondence between results presented in the appendix and results presented in the main paper.

# B  Further Related Work

On the algorithmic side, both OG and EG have been extensively studied over the past decades in the contexts of, among others, variational inequalities [44, 47], online optimization [13], and learning in games [17, 53]. While the original design of these methods considered the use of the same learning rate for both the extrapolation and the update step, several recent works have shown the benefit of scale separation between the two steps. Our method is directly inspired by [28], which proposed a double step-size variant of EG for achieving last-iterate convergence in stochastic variationally stable games. Among the other uses of learning rate separation of optimistic gradient methods, we should mention here [23, 58] for faster convergence in bilinear games, [18, 40, 50] for performance guarantees under weaker assumptions, and [24, 31] for robustness against delays.

Concerning the last-iterate convergence of no-regret learning dynamics in games with noisy feedback, most existing results rely on the use of vanishing learning rates and are established under more restrictive assumptions such as strong monotonicity [7, 28, 36] or strict variational stability [42, 43]. Our work, in contrast, studies learning with potentially non-vanishing learning rates in variationally stable games. This is made possible thanks to a clear distinction between additive and multiplicative noise; the latter has only been formerly explored in the game-theoretic context by [4, 41] for the class of cocoercive games.[4] Relaxing the cocoercivity assumption is a nontrivial challenge, as testified by the few number of works that establish last-iterate convergence results of stochastic algorithms for

---

[4]In the said works they use the term absolute random noise and relative random noise for additive noise and multiplicative noise.

monotone games. Except for [29] mentioned above, this was achieved either through mini-batching [10, 32], Tikhonov regularization / Halpen iteration [39], or both [11].

## C Additional Figures

In this section we provide the complete version of Figure 1. In additional to the algorithms already considered in the said figure, we also present results for the case where the two players follow the vanilla gradient descent methods, which we mark as GDA (gradient descent/ascent).

To begin, we complement the leftmost plot of Figure 1 by Figure 2, where we present individual plots of the trajectories induced by different algorithms for better visibility. For optimistic algorithm, we present the trajectory both of the sequence of play $\mathbf{x}_t = \mathbf{X}_{t+\frac{1}{2}}$ and of the auxiliary iterate $\mathbf{X}_t$. The two algorithms GDA and OG have their iterates spiral out, indicating a divergence behavior, conformed to our previous discussions. For OG+ run with constant learning rate and adaptive OptDA+, we observe that the trajectory of $\mathbf{X}_t$ is much "smoother" than that of $\mathbf{x}_t = \mathbf{X}_{t+\frac{1}{2}}$. This is because the extrapolation step is taken with a larger learning rate. Finally, adaptive OptDA+ has its iterates go far away from the equilibrium in the first few iterations due to the initialization with large learning rates, but eventually finds the right learning rates itself and ends up with a convergence speed and regret that is competitive with carefully tuned OG+.

Next, In Figure 3, we expand on the right two plots of Figure 1 with additional curves for GDA. GDA and OG run with the same decreasing learning rate sequences $\eta_t = 0.1/\sqrt{t+1}$ turn out to have similar performance. This suggests that without learning rate separation, the benefit of the extrapolation step may be completely lost in the presence of noise.

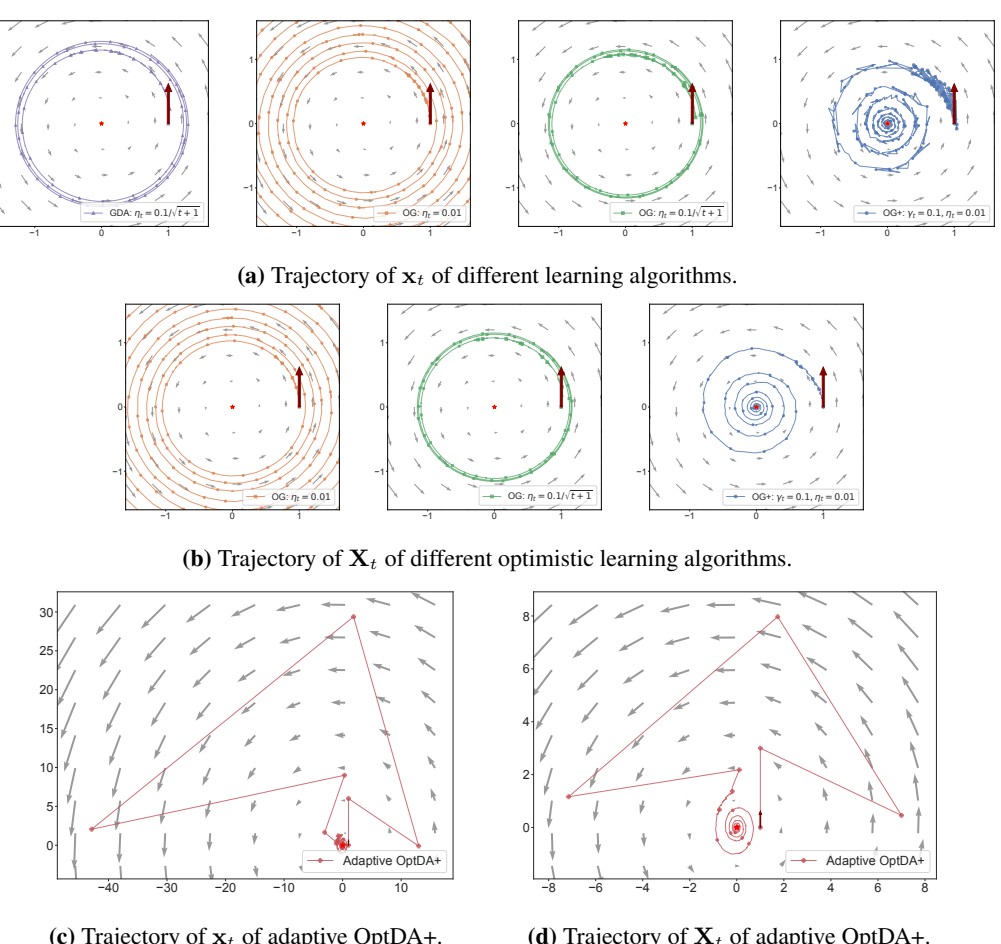

(a) Trajectory of $\mathbf{x}_t$ of different learning algorithms.

(b) Trajectory of $\mathbf{X}_t$ of different optimistic learning algorithms.

(c) Trajectory of $\mathbf{x}_t$ of adaptive OptDA+.    (d) Trajectory of $\mathbf{X}_t$ of adaptive OptDA+.

**Figure 2:** Trajectories induced by different learning algorithms on the model described in Example 1. We recall that for optimistic learning algorithms, the played point is $\mathbf{x}_t = \mathbf{X}_{t+\frac{1}{2}}$. We take $q = 1/4$ for adaptive OptDA+.

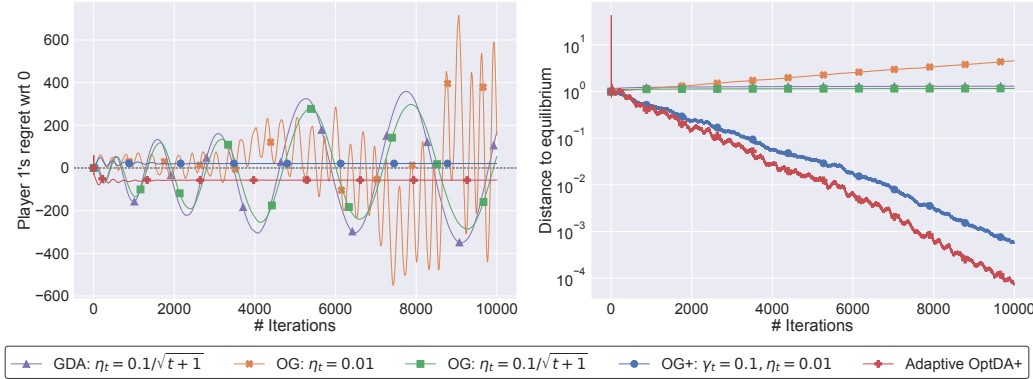

**Figure 3:** Player 1's regret and distance to equilibrium when both players follow a certain learning strategy in the model described in Example 1. We take $q = 1/4$ for adaptive OptDA+.

# D   Technical Details and Notations

In this section we introduce the necessary notations for our analysis and discuss some technical details omitted in the main text.

**Noise, initialization, and measurability.**    Throughout our proof, to emphasize that $g_t^i = V^i(x_t^i) + \xi_t^i$ is a stochastic estimate of $V^i(X_{t+\frac{1}{2}}^i)$ in our algorithms, we use the notations $\hat{V}_{t+\frac{1}{2}}^i = g_t^i$ and $\xi_{t+\frac{1}{2}}^i = \xi_t^i$. For the update of $X_{3/2}^i$, we systematically take $g_0^i = \hat{V}_{1/2}^i = 0$. We also write $\xi_{1/2}^i = 0$.

A part of our analysis will be built on the fact that $\xi_{t-\frac{1}{2}}^i$ is $\mathcal{F}_t$-measurable. There is however no a priori reason for this to be true – as $(\mathcal{F}_t)_{t \in \mathbb{N}}$ is the natural filtration associated to $(\mathbf{x}_t)_{t \in \mathbb{N}}$, a sequence that can for example be taken constant independent of the feedback. To address this, we establish here that $\xi_{t-\frac{1}{2}}^i$ is indeed $\mathcal{F}_t$-measurable when player $i$ uses OG+ or OptDA+ with learning rates satisfying a certain measurability assumption. To state it, we define $\mathcal{F}_t^i$ as the $\sigma$-algebra generated by $\{(\mathbf{x}_s)_{s=1}^t, (\xi_s^i)_{s=1}^{t-1}\}$.

**Assumption 5.**   For all $t \in \mathbb{N}$, the learning rates $\gamma_{t+1}^i$ and $\eta_{t+1}^i$ are $\mathcal{F}_t^i$-measurable.

The following lemma shows that whenever Assumption 5 holds, one can directly work with $(\mathcal{F}_t)_{t \in \mathbb{N}}$.

**Lemma 2.**   *Let player $i$ run (OG+) or (OptDA+) with learning rates satisfying Assumption 5. Then, for every $t \in \mathbb{N}$, it holds $\mathcal{F}_t^i = \mathcal{F}_t$. In other words, $\xi_{t-\frac{1}{2}}^i$ is $\mathcal{F}_t$-measurable.*

*Proof.*  We prove the lemma by induction. For $t = 1$, this is true by definition. Now, fix $t \geq 2$ and assume that we have proven the statements for all $s \leq t - 1$. To show that the statement is also true for $t$, we note that for both OG+ and OptDA+, $x_t^i = X_{t+\frac{1}{2}}^i$ is a linear combination of the vectors in $\{V^i(\mathbf{x}_s)\}_{s=1}^{t-1} \cup \{\xi_{s+\frac{1}{2}}^i\}_{s=1}^{t-1}$ with coefficients in $\{\eta_s^i\}_{s=1}^t \cup \{\gamma_t^i\}$. All the involved quantities except for $\xi_{t-\frac{1}{2}}^i$ is $\mathcal{F}_{t-1}$-measurable by the induction hypothesis. They are thus $\mathcal{F}_t$-measurable, and as $X_t^i$ is $\mathcal{F}_t$-measurable by the definition of $\mathcal{F}_t$ we concludes that $\xi_{t-\frac{1}{2}}^i$ is also $\mathcal{F}_t$-measurable, which along with the induction hypothesis implies immediately $\mathcal{F}_t^i = \mathcal{F}_t$. $\qquad\square$

An immediate consequence of Lemma 2 is the following.

**Corollary 1.**   *Let player $i$ run (OG+) or (OptDA+) with learning rates satisfying Assumption 5. Then for every $t \in \mathbb{N}$, $\gamma_{t+1}^i$ and $\eta_{t+1}^i$ are $\mathcal{F}_t$-measurable.*

Throughout the sequel, both Lemma 2 and Corollary 1 will be used implicitly. Our adaptive learning rates (Adapt) apparently satisfy Assumption 5. As for the non-adaptive case, for simplicity, we assume all their learning rates are predetermined, that is, they are $\mathcal{F}_1$-measurable; for more details on this point see Remark 1. $\mathcal{F}_0$ denotes the trivial $\sigma$-algebra.

As another technical detail, in our proofs we assume deterministic $\mathbf{X}_1$, but the entire analysis still goes through for random $\mathbf{X}_1$ under the following conditions

1. For non-adaptive algorithms, we require $\mathbb{E}[\|\mathbf{X}_1\|^2] < +\infty$.
2. For adaptive OptDA+, we require existence of $R \in \mathbb{R}_+$ such that $\|\mathbf{X}_1\| \leq R$ holds almost surely.

**Notations related to the learning rates.** For any $\mathbf{x} = (x^i)_{i \in \mathcal{N}} \in \mathcal{X} = \mathbb{R}^d$ and $\boldsymbol{\alpha} = (\alpha^i)_{i \in \mathcal{N}} \in \mathbb{R}_+^N$, we write the weighted norm as $\|\mathbf{x}\|_{\boldsymbol{\alpha}} = \sqrt{\sum_{i=1}^N \alpha^i \|x^i\|^2}$. The weights $\boldsymbol{\alpha}$ will be taken as a function of the learning rates. It is thus convenient to write $\boldsymbol{\eta}_t = (\eta_t^i)_{i \in \mathcal{N}}$ and $\boldsymbol{\gamma}_t = (\gamma_t^i)_{i \in \mathcal{N}}$ for the joint learning rates. The arithmetic manipulation and the comparisons of these vectors should be taken elementwisely. For example, the element-wise division is $1/\boldsymbol{\eta}_t = (1/\eta_t^i)_{i \in \mathcal{N}}$. For ease of notation, we also write $\|\boldsymbol{\alpha}\|_1 = \sum_{i=1}^N \alpha^i$ and $\|\boldsymbol{\alpha}\|_\infty = \max_{i \in \mathcal{N}} \alpha^i$ respectively for the L1 norm and the L-infinity norm of an $N$-dimensional vector $\boldsymbol{\alpha}$.

# E   Preliminary Analysis for OG+ and OptDA+

In this section, we lay out the basis for the analysis of OG+ and OptDA+.

## E.1   Generalized Schemes with Arbitrary Input Sequences

As a starting point, we derive elementary energy inequalities for the following two generalized schemes run with arbitrary vector sequences $(g_t)_{t \in \mathbb{N}}$ and $(g_{t+\frac{1}{2}})_{t \in \mathbb{N}}$.

- Generalized OG+ $\qquad\qquad X_{t+\frac{1}{2}} = X_t - \gamma_t g_t, \quad X_{t+1} = X_t - \eta_{t+1} g_{t+\frac{1}{2}}$
- Generalized OptDA+ $\qquad X_{t+\frac{1}{2}} = X_t - \gamma_t g_t, \quad X_{t+1} = X_1 - \eta_{t+1} \sum_{s=1}^t g_{s+\frac{1}{2}}$

In fact, Generalized OG+ with $g_{t+\frac{1}{2}} = \nabla f_t(X_{t+\frac{1}{2}})$ is nothing but the unconstrained, double step-size variant of the optimistic mirror descent method proposed in [53]. On the other hand, Generalized OptDA+ with single learning rate was introduced in [5] under the name of generalized extra-gradient. These two methods coincide when the learning rates are taken constant. In practice, $g_{t+\frac{1}{2}}$ is almost always an estimate of $\nabla f_t(X_{t+\frac{1}{2}})$ while $g_t$ is an approximation of $g_{t+\frac{1}{2}}$. As a matter of fact, as we show in the following propositions, the dot product $\langle g_{t+\frac{1}{2}}, g_t \rangle$ appears with a negative sign in the energy inequalities, which results in a negative contribution when the two vectors are close.

We start with the energy inequality for Generalized OG+.

**Proposition 3** (Energy inequality for Generalized OG+). *Let $(X_t)_{t \in \mathbb{N}}$ and $(X_{t+\frac{1}{2}})_{t \in \mathbb{N}}$ be generated by Generalized OG+. It holds for any $p \in \mathcal{X}$ and $t \in \mathbb{N}$ that*

$$\|X_{t+1} - p\|^2 = \|X_t - p\|^2 - 2\eta_{t+1}\langle g_{t+\frac{1}{2}}, X_{t+\frac{1}{2}} - p \rangle - 2\gamma_t \eta_{t+1}\langle g_{t+\frac{1}{2}}, g_t \rangle + (\eta_{t+1})^2 \|g_{t+\frac{1}{2}}\|^2.$$

*Proof.* We develop directly

$$
\begin{aligned}
\|X_{t+1} - p\|^2 &= \|X_t - \eta_{t+1} g_{t+\frac{1}{2}} - p\|^2 \\
&= \|X_t - p\|^2 - 2\langle g_{t+\frac{1}{2}}, X_t - p \rangle + (\eta_{t+1})^2 \|g_{t+\frac{1}{2}}\|^2 \\
&= \|X_t - p\|^2 - 2\eta_{t+1}\langle g_{t+\frac{1}{2}}, X_{t+\frac{1}{2}} - p \rangle - 2\gamma_t \eta_{t+1}\langle g_{t+\frac{1}{2}}, g_t \rangle + (\eta_{t+1})^2 \|g_{t+\frac{1}{2}}\|^2,
\end{aligned}
$$

where in the last equality we use the fact that $X_t = X_{t+\frac{1}{2}} + \gamma_t g_t$. $\qquad\square$

For Generalized OptDA+ we have almost the same inequality but for squared distance weighted by $1/\eta_t$, with the notation $\eta_1 = \eta_2$.

**Proposition 4** (Energy inequality for Generalized OptDA+). *Let $(X_t)_{t \in \mathbb{N}}$ and $(X_{t+\frac{1}{2}})_{t \in \mathbb{N}}$ be generated by Generalized OptDA+. It holds for any $p \in \mathcal{X}$ and $t \in \mathbb{N}$ that*

$$
\begin{aligned}
\frac{\|X_{t+1} - p\|^2}{\eta_{t+1}} &= \frac{\|X_t - p\|^2}{\eta_t} - \frac{\|X_t - X_{t+1}\|^2}{\eta_t} \\
&\quad + \left(\frac{1}{\eta_{t+1}} - \frac{1}{\eta_t}\right)\|X_1 - p\|^2 - \left(\frac{1}{\eta_{t+1}} - \frac{1}{\eta_t}\right)\|X_1 - X_{t+1}\|^2 \\
&\quad - 2\langle g_{t+\frac{1}{2}}, X_{t+\frac{1}{2}} - p \rangle - 2\gamma_t\langle g_{t+\frac{1}{2}}, g_t \rangle + \langle g_{t+\frac{1}{2}}, X_t - X_{t+1} \rangle.
\end{aligned}
$$

*Proof.* Using $g_{t+\frac{1}{2}} = (X_t - X_1)/\eta_t - (X_{t+1} - X_1)/\eta_{t+1}$, we can write

$$\langle g_{t+\frac{1}{2}}, X_{t+1} - p \rangle = \left\langle \frac{X_t - X_1}{\eta_t} - \frac{X_{t+1} - X_1}{\eta_{t+1}}, X_{t+1} - p \right\rangle$$

$$= \frac{1}{\eta_t}\langle X_t - X_{t+1}, X_{t+1} - p \rangle + \left(\frac{1}{\eta_{t+1}} - \frac{1}{\eta_t}\right)\langle X_1 - X_{t+1}, X_{t+1} - p \rangle$$

$$= \frac{1}{2\eta_t}(\|X_t - p\|^2 - \|X_{t+1} - p\|^2 - \|X_t - X_{t+1}\|^2)$$

$$+ \left(\frac{1}{2\eta_{t+1}} - \frac{1}{2\eta_t}\right)(\|X_1 - p\|^2 - \|X_{t+1} - p\|^2 - \|X_1 - X_{t+1}\|^2).$$

Multiplying the equality by 2 and rearranging, we get

$$\frac{\|X_{t+1} - p\|^2}{\eta_{t+1}} = \frac{\|X_t - p\|^2}{\eta_t} - \frac{\|X_t - X_{t+1}\|^2}{\eta_t} + \left(\frac{1}{\eta_{t+1}} - \frac{1}{\eta_t}\right)\|X_1 - p\|^2$$

$$- \left(\frac{1}{\eta_{t+1}} - \frac{1}{\eta_t}\right)\|X_1 - X_{t+1}\|^2 - 2\langle g_{t+\frac{1}{2}}, X_{t+1} - p \rangle.$$

We conclude with the equality

$$\langle g_{t+\frac{1}{2}}, X_{t+1} - p \rangle = \langle g_{t+\frac{1}{2}}, X_{t+1} - X_t \rangle + \langle g_{t+\frac{1}{2}}, X_t - X_{t+\frac{1}{2}} \rangle + \langle g_{t+\frac{1}{2}}, X_{t+\frac{1}{2}} - p \rangle$$

$$= \langle g_{t+\frac{1}{2}}, X_{t+1} - X_t \rangle + \gamma_t\langle g_{t+\frac{1}{2}}, g_t \rangle + \langle g_{t+\frac{1}{2}}, X_{t+\frac{1}{2}} - p \rangle,$$

where we have used $X_t = X_{t+\frac{1}{2}} + \gamma_t g_t$. $\qquad\square$

Throughout our work, we assume the learning rate sequences to be non-increasing. This is essential for OptDA+, as it guarantees the following corollary.

**Corollary 2.** *Let $(X_t)_{t\in\mathbb{N}}$ and $(X_{t+\frac{1}{2}})_{t\in\mathbb{N}}$ be generated by Generalized OptDA+. For any $p \in \mathcal{X}$ and $t \in \mathbb{N}$, if $\eta_{t+1} \leq \eta_t$, it holds that*

$$\frac{\|X_{t+1} - p\|^2}{\eta_{t+1}} \leq \frac{\|X_t - p\|^2}{\eta_t} + \left(\frac{1}{\eta_{t+1}} - \frac{1}{\eta_t}\right)\|X_1 - p\|^2 - 2\langle g_{t+\frac{1}{2}}, X_{t+\frac{1}{2}} - p \rangle$$

$$- 2\gamma_t\langle g_{t+\frac{1}{2}}, g_t \rangle + \eta_t^2\|g_{t+\frac{1}{2}}\|^2 + \min\left(\eta_t^2\|g_{t+\frac{1}{2}}\|^2 - \frac{\|X_t - X_{t+1}\|^2}{2\eta_t}, 0\right).$$

*Proof.* This is immediate from Proposition 4 by applying Young's inequality. More precisely, we use $(1/\eta_{t+1} - 1/\eta_t)\|X_1 - X_{t+1}\|^2 \geq 0$ and

$$2\langle g_{t+\frac{1}{2}}, X_{t+\frac{1}{2}} - p \rangle \leq \min\left(\eta_t^2\|g_{t+\frac{1}{2}}\|^2 + \frac{\|X_t - X_{t+1}\|^2}{\eta_t}, 2\eta_t^2\|g_{t+\frac{1}{2}}\|^2 + \frac{\|X_t - X_{t+1}\|^2}{2\eta_t}\right). \quad\square$$

### E.2 Quasi-Descent Inequalities for OG+ and OptDA+

We now turn back to (OG+) and (OptDA+) introduced in Section 4. These are special cases of Generalized OG+ and Generalized OptDA+ with $g_t = g_{t-\frac{1}{2}} = \hat{V}_{t-\frac{1}{2}}^i$. The following lemma provides an upper bound on the conditional expectation of $\langle \hat{V}_{t+\frac{1}{2}}^i, \hat{V}_{t-\frac{1}{2}}^i \rangle$ when all the players follow one of the two strategies, and is essential for establishing our quasi-descent inequities.

**Lemma 3.** *Let Assumptions 1 and 3 hold and all players run either (OG+) or (OptDA+) with learning rates satisfying Assumption 5. Then, for all $i \in \mathcal{N}$ and $t \geq 2$, it holds*

$$-2\,\mathbb{E}_{t-1}[\langle \hat{V}_{t+\frac{1}{2}}^i, \hat{V}_{t-\frac{1}{2}}^i \rangle] \leq \mathbb{E}_{t-1}\Bigg[ -\|V^i(\mathbf{X}_{t+\frac{1}{2}})\|^2 - \|V^i(\mathbf{X}_{t-\frac{1}{2}})\|^2$$

$$+ \|V^i(\mathbf{X}_{t+\frac{1}{2}}) - V^i(\mathbf{X}_{t-\frac{1}{2}})\|^2$$

$$+ L\left(\gamma_t^i\|\xi_{t-\frac{1}{2}}^i\|^2 + \sum_{j=1}^N \frac{(\eta_t^j + \gamma_t^j)^2\|\xi_{t-\frac{1}{2}}^j\|^2}{\gamma_t^i}\right)\Bigg]$$

*Proof.* Thanks to Lemma 2, we can apply the law of total expectation of the expectation to get

$$\mathbb{E}_{t-1}[\langle \hat{V}^i_{t+\frac{1}{2}}, \hat{V}^i_{t-\frac{1}{2}} \rangle] = \mathbb{E}_{t-1}[\langle \mathbb{E}_t[\hat{V}^i_{t+\frac{1}{2}}], \hat{V}^i_{t-\frac{1}{2}} \rangle]$$
$$= \mathbb{E}_{t-1}[\langle V^i(\mathbf{X}_{t+\frac{1}{2}}), \hat{V}^i_{t-\frac{1}{2}} \rangle]$$
$$= \mathbb{E}_{t-1}[\langle V^i(\mathbf{X}_{t+\frac{1}{2}}), V^i(\mathbf{X}_{t-\frac{1}{2}}) \rangle + \langle V^i(\mathbf{X}_{t+\frac{1}{2}}), \xi^i_{t-\frac{1}{2}} \rangle]. \tag{7}$$

We rewrite the first term as

$$2\langle V^i(\mathbf{X}_{t+\frac{1}{2}}), V^i(\mathbf{X}_{t-\frac{1}{2}}) \rangle = \|V^i(\mathbf{X}_{t+\frac{1}{2}})\|^2 + \|V^i(\mathbf{X}_{t-\frac{1}{2}})\|^2 - \|V^i(\mathbf{X}_{t+\frac{1}{2}}) - V^i(\mathbf{X}_{t-\frac{1}{2}})\|^2. \tag{8}$$

As for the second term, for all $j \in \mathcal{N}$, we define $\tilde{X}^j_{t+\frac{1}{2}} = X^j_{t+\frac{1}{2}} + (\eta^j_t + \gamma^j_t)\xi^j_{t-\frac{1}{2}}$ and as a surrogate for $X^j_{t+\frac{1}{2}}$ obtained by removing the noise of round $t-1$. For OG+ and OptDA+ we have respectively

$$\tilde{X}^j_{t+\frac{1}{2}} = X^j_{t-1} - (\eta^j_t + \gamma^j_t)V^j(\mathbf{X}_{t-\frac{1}{2}})$$
$$\tilde{X}^j_{t+\frac{1}{2}} = X^i_1 - \eta^j_t \sum_{s=1}^{t-2} \hat{V}^j_{s+\frac{1}{2}} - (\eta^j_t + \gamma^j_t)V^j(\mathbf{X}_{t-\frac{1}{2}}).$$

With Assumption 5 we then deduce that $\tilde{\mathbf{X}}_{t+\frac{1}{2}}$ is $\mathcal{F}_{t-1}$-measurable and hence

$$\mathbb{E}_{t-1}[\langle V^i(\tilde{\mathbf{X}}_{t+\frac{1}{2}}), \xi^i_{t-\frac{1}{2}} \rangle] = \langle V^i(\tilde{\mathbf{X}}_{t+\frac{1}{2}}), \mathbb{E}_{t-1}[\xi^i_{t-\frac{1}{2}}] \rangle = 0.$$

Moreover, by definition of $\tilde{\mathbf{X}}_{t+\frac{1}{2}}$ we have

$$\|\mathbf{X}_{t+\frac{1}{2}} - \tilde{\mathbf{X}}_{t+\frac{1}{2}}\|^2 = \sum_{j=1}^{N} \|X^j_{t+\frac{1}{2}} - \tilde{X}^j_{t+\frac{1}{2}}\|^2 = \sum_{j=1}^{N} (\eta^j_t + \gamma^j_t)^2 \|\xi^j_{t-\frac{1}{2}}\|^2$$

It then follows from the Lipschitz continuity of $V^i$ that

$$\mathbb{E}_{t-1}[-\langle V^i(\mathbf{X}_{t+\frac{1}{2}}), \xi^i_{t-\frac{1}{2}} \rangle] = \mathbb{E}_{t-1}[-\langle V^i(\mathbf{X}_{t+\frac{1}{2}}) - V^i(\tilde{\mathbf{X}}_{t+\frac{1}{2}}), \xi^i_{t-\frac{1}{2}} \rangle]$$
$$- \mathbb{E}_{t-1}[\langle V^i(\tilde{\mathbf{X}}_{t+\frac{1}{2}}), \xi^i_{t-\frac{1}{2}} \rangle]$$
$$\leq \mathbb{E}_{t-1}[L\|\mathbf{X}_{t+\frac{1}{2}} - \tilde{\mathbf{X}}_{t+\frac{1}{2}}\| \|\xi^i_{t-\frac{1}{2}}\|]$$
$$\leq \mathbb{E}_{t-1}\left[L\left(\frac{\|\mathbf{X}_{t+\frac{1}{2}} - \tilde{\mathbf{X}}_{t+\frac{1}{2}}\|^2}{2\gamma^i_t} + \frac{\gamma^i_t\|\xi^i_{t-\frac{1}{2}}\|^2}{2}\right)\right]$$
$$= \mathbb{E}_{t-1}\left[L\left(\frac{\gamma^i_t\|\xi^i_{t-\frac{1}{2}}\|^2}{2} + \sum_{j=1}^{N} \frac{(\eta^j_t + \gamma^j_t)^2\|\xi^j_{t-\frac{1}{2}}\|^2}{2\gamma^i_t}\right)\right]. \tag{9}$$

Putting (7), (8), and (9) together gives the desired inequality. $\qquad\square$

**Quasi-Descent Inequalities for OG+.** Below we establish respectively the individual and the global quasi-descent inequalities for OG+. In this part, all the players use the same learning rate sequences and we can thus drop the player index in the learning rates.

**Lemma 4** (Individual quasi-descent inequality for OG+). *Let Assumptions 1 and 3 hold and all players run* (OG+) *with the same predetermined learning rate sequences. Then, for all $i \in \mathcal{N}$, $t \geq 2$, and $p^i \in \mathcal{X}^i$, it holds*

$$\mathbb{E}_{t-1}[\|X^i_{t+1} - p^i\|^2] \leq \mathbb{E}_{t-1}[\|X^i_t - p^i\|^2 - 2\eta_{t+1}\langle V^i(\mathbf{X}_{t+\frac{1}{2}}), X^i_{t+\frac{1}{2}} - p^i \rangle$$
$$- \gamma_t\eta_{t+1}(\|V^i(\mathbf{X}_{t+\frac{1}{2}})\|^2 + \|V^i(\mathbf{X}_{t-\frac{1}{2}})\|^2)$$
$$+ \gamma_t\eta_{t+1}\|V^i(\mathbf{X}_{t+\frac{1}{2}}) - V^i(\mathbf{X}_{t-\frac{1}{2}})\|^2 + \gamma_t^2\eta_{t+1}L\|\xi^i_{t-\frac{1}{2}}\|^2$$
$$+ \eta_{t+1}(\eta_t + \gamma_t)^2 L\|\boldsymbol{\xi}_{t-\frac{1}{2}}\|^2 + (\eta_{t+1})^2\|\hat{V}^i_{t+\frac{1}{2}}\|^2]. \tag{10}$$

*Proof.* We apply Proposition 3 to player $i$'s update and $p \leftarrow p^i$. Since the inequality holds for any realization we can take expectation with respect to $\mathcal{F}_{t-1}$ to get

$$\mathbb{E}_{t-1}[\|X_{t+1}^i - p^i\|^2] = \mathbb{E}_{t-1}[\|X_t^i - p^i\|^2 - 2\eta_{t+1}\langle \hat{V}_{t+\frac{1}{2}}^i, X_{t+\frac{1}{2}}^i - p^i\rangle$$
$$- 2\gamma_t\eta_{t+1}\langle \hat{V}_{t+\frac{1}{2}}^i, \hat{V}_{t-\frac{1}{2}}^i\rangle + (\eta_{t+1})^2\|\hat{V}_{t+\frac{1}{2}}^i\|^2].$$

The learning rates $\gamma_t$ and $\eta_{t+1}$ being $\mathcal{F}_1$-measurable and in particular $\mathcal{F}_{t-1}$-measurable, we conclude immediately with Lemma 3 and the equality

$$\mathbb{E}_{t-1}[\eta_{t+1}\langle \hat{V}_{t+\frac{1}{2}}^i, X_{t+\frac{1}{2}}^i - p^i\rangle] = \eta_{t+1}\,\mathbb{E}_{t-1}[\langle V^i(\mathbf{X}_{t+\frac{1}{2}}), X_{t+\frac{1}{2}}^i - p^i\rangle]. \qquad \square$$

**Remark 1.** From the proof of Lemma 4 we see that the exact requirement concerning the measurability of the learning rates here is that both $\gamma_t$ and $\eta_{t+1}$ should be $\mathcal{F}_{t-1}$-measurable. For simplicity throughout our analysis for OG+ we simply say that all the learning rates are predetermined, i.e., $\mathcal{F}_1$-measurable. In contrast, for OptDA+ Assumption 5 is indeed sufficient. This is a technical detail that we have omitted in the main text.

**Lemma 5** (Global quasi-descent inequality for OG+). *Let Assumptions 1–3 hold and all players run* (OG+) *with the same predetermined learning rate sequences. Then, for all $t \geq 2$ and $\mathbf{x}_\star \in \mathcal{X}_\star$, we have*

$$\mathbb{E}_{t-1}[\|\mathbf{X}_{t+1} - \mathbf{x}_\star\|^2] \leq \mathbb{E}_{t-1}[\|\mathbf{X}_t - \mathbf{x}_\star\|^2 - \gamma_t\eta_{t+1}(\|\mathbf{V}(\mathbf{X}_{t+\frac{1}{2}})\|^2 + \|\mathbf{V}(\mathbf{X}_{t-\frac{1}{2}})\|^2)$$
$$+ 3\gamma_t\eta_{t+1}NL^2((\eta_t^2 + \gamma_t^2)\|\hat{\mathbf{V}}_{t-\frac{1}{2}}\|^2 + (\gamma_{t-1})^2\|\hat{\mathbf{V}}_{t-\frac{3}{2}}\|^2)$$
$$+ (\gamma_t^2\eta_{t+1} + N\eta_{t+1}(\eta_t + \gamma_t)^2)L\|\boldsymbol{\xi}_{t-\frac{1}{2}}\|^2 + (\eta_{t+1})^2\|\hat{\mathbf{V}}_{t+\frac{1}{2}}\|^2].$$

*Proof.* We will apply Lemma 4 to $x_\star^i$. We first bound the variation $\|V^i(\mathbf{X}_{t+\frac{1}{2}}) - V^i(\mathbf{X}_{t-\frac{1}{2}})\|^2$ by

$$\|V^i(\mathbf{X}_{t+\frac{1}{2}}) - V^i(\mathbf{X}_{t-\frac{1}{2}})\|^2 \leq 3\|V^i(\mathbf{X}_{t+\frac{1}{2}}) - V^i(\mathbf{X}_t)\|^2 + 3\|V^i(\mathbf{X}_t) - V^i(\mathbf{X}_{t-1})\|^2$$
$$+ 3\|V^i(\mathbf{X}_{t-1}) - V^i(\mathbf{X}_{t-\frac{1}{2}})\|^2$$
$$\leq 3\gamma_t^2L^2\|\hat{\mathbf{V}}_{t-\frac{1}{2}}\|^2 + 3\eta_t^2L^2\|\hat{\mathbf{V}}_{t-\frac{1}{2}}\|^2 + 3(\gamma_{t-1})^2L^2\|\hat{\mathbf{V}}_{t-\frac{3}{2}}\|^2. \quad (11)$$

In the second inequality, we have used the Lipschitz continuity of $V^i$ and $\mathbf{X}_{t+\frac{1}{2}} = \mathbf{X}_t - \gamma_t\hat{\mathbf{V}}_{t-\frac{1}{2}}$ to obtain

$$\|V^i(\mathbf{X}_{t+\frac{1}{2}}) - V^i(\mathbf{X}_t)\|^2 \leq L^2\|\mathbf{X}_{t+\frac{1}{2}} - \mathbf{X}_t\|^2 = 3\gamma_t^2L^2\|\hat{\mathbf{V}}_{t-\frac{1}{2}}\|^2.$$

The terms $\|V^i(\mathbf{X}_t) - V^i(\mathbf{X}_{t-1})\|^2$ and $\|V^i(\mathbf{X}_{t-1}) - V^i(\mathbf{X}_{t-\frac{1}{2}})\|^2$ were bounded in the same way. Applying Lemma 4 with $p^i \leftarrow x_\star^i$, plugging (11) into (10), and summing from $i = 1$ to $N$ then yields

$$\mathbb{E}_{t-1}[\|\mathbf{X}_{t+1} - \mathbf{x}_\star\|^2] \leq \mathbb{E}_{t-1}[\|\mathbf{X}_t - \mathbf{x}_\star\|^2 - \eta_{t+1}\langle \mathbf{V}(\mathbf{X}_{t+\frac{1}{2}}), \mathbf{X}_{t+\frac{1}{2}} - \mathbf{x}_\star\rangle$$
$$- \gamma_t\eta_{t+1}(\|\mathbf{V}(\mathbf{X}_{t+\frac{1}{2}})\|^2 + \|\mathbf{V}(\mathbf{X}_{t-\frac{1}{2}})\|^2)$$
$$+ 3\gamma_t\eta_{t+1}NL^2((\eta_t^2 + \gamma_t^2)\|\hat{\mathbf{V}}_{t-\frac{1}{2}}\|^2 + (\gamma_{t-1})^2\|\hat{\mathbf{V}}_{t-\frac{3}{2}}\|^2)$$
$$+ (\gamma_t^2\eta_{t+1} + N\eta_{t+1}(\eta_t + \gamma_t)^2)L\|\boldsymbol{\xi}_{t-\frac{1}{2}}\|^2 + (\eta_{t+1})^2\|\hat{\mathbf{V}}_{t+\frac{1}{2}}\|^2].$$

To conclude, we drop $-\eta_{t+1}\langle \mathbf{V}(\mathbf{X}_{t+\frac{1}{2}}), \mathbf{X}_{t+\frac{1}{2}} - \mathbf{x}_\star\rangle$ which is non-positive by Assumption 2. $\quad \square$

**Quasi-Descent Inequalities for OptDA+.** Similarly, we establish quasi-descent inequalities for OptDA+ that will be used for both non-adaptive and adaptive analyses.

**Lemma 6** (Individual quasi-descent inequality for OptDA+). *Let Assumptions 1 and 3 hold and all players run* (OptDA+) *with non-increasing learning rates satisfying Assumption 5. Then, for all*

$i \in \mathcal{N}$, $t \geq 2$, and $p^i \in \mathcal{X}^i$, it holds

$$\mathbb{E}_{t-1}\left[\frac{\|X_{t+1}^i - p^i\|^2}{\eta_{t+1}^i}\right] \leq \mathbb{E}_{t-1}\left[\frac{\|X_t^i - p^i\|^2}{\eta_t^i} + \left(\frac{1}{\eta_{t+1}^i} - \frac{1}{\eta_t^i}\right)\|X_1^i - p^i\|^2\right.$$
$$- 2\langle V^i(\mathbf{X}_{t+\frac{1}{2}}), X_{t+\frac{1}{2}}^i - p^i\rangle$$
$$- \gamma_t^i(\|V^i(\mathbf{X}_{t+\frac{1}{2}})\|^2 + \|V^i(\mathbf{X}_{t-\frac{1}{2}})\|^2)$$
$$+ \gamma_t^i\|V^i(\mathbf{X}_{t+\frac{1}{2}}) - V^i(\mathbf{X}_{t-\frac{1}{2}})\|^2$$
$$+ \min\left(-\frac{\|X_t^i - X_{t+1}^i\|^2}{2\eta_t^i} + \eta_t^i\|\hat{V}_{t+\frac{1}{2}}^i\|^2, 0\right)$$
$$\left. + (\gamma_t^i)^2 L\|\xi_{t-\frac{1}{2}}^i\|^2 + L\|\boldsymbol{\xi}_{t-\frac{1}{2}}\|_{(\boldsymbol{\eta}_t + \boldsymbol{\gamma}_t)^2}^2 + \eta_t^i\|\hat{V}_{t+\frac{1}{2}}^i\|^2\right]. \quad (12)$$

*Proof.* This is an immediate by combining Corollary 2 and Lemma 3. We just notice that as $\gamma_t^i$ is $\mathcal{F}_{t-1}$-measurable, we have $\mathbb{E}_{t-1}[\gamma_t^i\langle\hat{V}_{t+\frac{1}{2}}^i, \hat{V}_{t-\frac{1}{2}}^i\rangle] = \gamma_t^i \mathbb{E}_{t-1}[\langle\hat{V}_{t+\frac{1}{2}}^i, \hat{V}_{t-\frac{1}{2}}^i\rangle]$. □

**Lemma 7** (Global quasi-descent inequality for OptDA+). *Let Assumptions 1–3 hold and all players run (OptDA+) with non-increasing learning rates satisfying Assumption 5. Then, for all $t \geq 2$ and $\mathbf{x}_\star \in \mathcal{X}_\star$, if $\boldsymbol{\eta}_t \leq \boldsymbol{\gamma}_t$, we have*

$$\mathbb{E}_{t-1}[\|\mathbf{X}_{t+1} - \mathbf{x}_\star\|_{1/\boldsymbol{\eta}_{t+1}}^2] \leq \mathbb{E}_{t-1}[\|\mathbf{X}_t - \mathbf{x}_\star\|_{1/\boldsymbol{\eta}_t}^2 + \|\mathbf{X}_1 - \mathbf{x}_\star\|_{1/\boldsymbol{\eta}_{t+1} - 1/\boldsymbol{\eta}_t}^2$$
$$- \|\mathbf{V}(\mathbf{X}_{t+\frac{1}{2}})\|_{\boldsymbol{\gamma}_t}^2 - \|\mathbf{V}(\mathbf{X}_{t-\frac{1}{2}})\|_{\boldsymbol{\gamma}_t}^2$$
$$- \|\mathbf{X}_t - \mathbf{X}_{t+1}\|_{1/(2\boldsymbol{\eta}_t)}^2 + 3\|\mathbf{V}(\mathbf{X}_t) - \mathbf{V}(\mathbf{X}_{t-1})\|_{\boldsymbol{\gamma}_t}^2$$
$$+ 3L^2(\|\boldsymbol{\gamma}_t\|_1\|\hat{\mathbf{V}}_{t-\frac{1}{2}}\|_{\boldsymbol{\gamma}_t^2}^2 + \|\boldsymbol{\gamma}_{t-1}\|_1\|\hat{\mathbf{V}}_{t-\frac{3}{2}}\|_{(\boldsymbol{\gamma}_{t-1})^2}^2)$$
$$+ (4N+1)L\|\boldsymbol{\xi}_{t-\frac{1}{2}}\|_{\boldsymbol{\gamma}_t^2}^2 + 2\|\hat{\mathbf{V}}_{t+\frac{1}{2}}\|_{\boldsymbol{\eta}_t}^2]. \quad (13)$$

*Proof.* The result is proved in the same way as Lemma 5 but instead of Lemma 4 we make use of Lemma 6 with

$$\min\left(-\frac{\|X_t^i - X_{t+1}^i\|^2}{2\eta_t^i} + \eta_t^i\|\hat{V}_{t+\frac{1}{2}}^i\|^2, 0\right) \leq -\frac{\|X_t^i - X_{t+1}^i\|^2}{2\eta_t^i} + \eta_t^i\|\hat{V}_{t+\frac{1}{2}}^i\|^2.$$

Moreover, as there is not a simple expression for $\|\mathbf{X}_t - \mathbf{X}_{t+1}\|$, in the place of (11) we use

$$\|V^i(\mathbf{X}_{t+\frac{1}{2}}) - V^i(\mathbf{X}_{t-\frac{1}{2}})\|^2 \leq 3L^2\|\hat{\mathbf{V}}_{t-\frac{1}{2}}\|_{\boldsymbol{\gamma}_t^2}^2 + 3L^2\|\hat{\mathbf{V}}_{t-\frac{3}{2}}\|_{(\boldsymbol{\gamma}_{t-1})^2}^2 + 3\|V^i(\mathbf{X}_t) - V^i(\mathbf{X}_{t-1})\|^2. \quad (14)$$

To obtain (13), we further use $\boldsymbol{\eta}_t \leq \boldsymbol{\gamma}_t$ and $\|\boldsymbol{\gamma}_t\|_1 \leq \|\boldsymbol{\gamma}_{t-1}\|_1$. □

**Remark 2.** The players can take different learning rates in OptDA+ because in the quasi-descent inequality (12), there is no learning rate in front of $\langle V^i(\mathbf{X}_{t+\frac{1}{2}}), X_{t+\frac{1}{2}}^i - p^i\rangle$. Take $p^i \leftarrow x_\star^i$ and summing from $i = 1$ to $N$ we get directly $\langle\mathbf{V}(\mathbf{X}_{t+\frac{1}{2}}), \mathbf{X}_{t+\frac{1}{2}} - \mathbf{x}_\star\rangle$ which is non-negative according to Assumption 2. While it is also possible to put (10) in the form of Lemma 1, we are not able to control the sum of $(1/\eta_{t+1}^i - 1/\eta_t^i)\|X_t^i - p^i\|^2$ as explained in Section 4.

# F   Regret Analysis with Predetermined Learning Rates

In this section, we tackle the regret analysis of OG+ and OptDA+ run with non-adaptive learning rates. We prove bounds on the pseudo-regret $\max_{p^i \in \mathcal{K}^i} \mathbb{E}[\text{Reg}_T^i(p^i)]$ and on the sum of the expected magnitude of the noiseless feedback $\sum_{t=1}^T \mathbb{E}[\|\mathbf{V}(\mathbf{X}_{t+\frac{1}{2}})\|^2]$. In fact, in our analysis, building bounds on $\sum_{t=1}^T \mathbb{E}[\|\mathbf{V}(\mathbf{X}_{t+\frac{1}{2}})\|^2]$ is a crucial step for deriving bounds on the pseudo-regret.

Moreover, as the loss functions are convex in their respective player's action parameter, a player's regret can be bounded by its linearized counterpart, as stated in the following lemma.

**Lemma 8.** *Let Assumption 1 hold. Then, for all $i \in \mathcal{N}$, any sequence of actions $(\mathbf{x}_t)_{t\in\mathbb{N}}$, and all reference point $p^i \in \mathcal{X}^i$, we have*

$$\mathrm{Reg}_T^i(p^i) \le \sum_{t=1}^{T} \langle V^i(\mathbf{x}_t), x_t^i - p^i \rangle$$

We therefore focus exclusively on bounding the linearized regret in the sequel.

### F.1 Bounds for OG+

In this part we will simply assume the learning rates to be $\mathcal{F}_1$-measurable, a technical detailed that we ignored in the main text. The global quasi-descent inequality of OG+ introduced in Lemma 5 indeed allows us to bound several important quantities, as shown below.

**Proposition 5** (Bound on sum of squared norms). *Let Assumptions 1–3 hold and all players run (OG+) with learning rates described in Theorem 1. Then, for all $T \in \mathbb{N}$ and $\mathbf{x}_\star \in \mathcal{X}_\star$, we have*

$$\mathbb{E}[\|\mathbf{X}_{t+1} - \mathbf{x}_\star\|^2] + \frac{1}{2}\sum_{t=1}^{T}\gamma_t\eta_{t+1}\,\mathbb{E}[\|\mathbf{V}(\mathbf{X}_{t+\frac{1}{2}})\|^2]$$

$$\le \|\mathbf{X}_1 - \mathbf{x}_\star\|^2 + \gamma_1\eta_2\|V(\mathbf{X}_1)\|^2 + \sum_{t=1}^{T}\left(9\gamma_t^3\eta_{t+1}NL^2 + \gamma_t^2\eta_{t+1}(4N+1)L + (\eta_{t+1})^2\right)N\sigma_A^2.$$

*Accordingly, $\sum_{t=1}^{\infty}\gamma_t\eta_{t+1}\,\mathbb{E}[\|\mathbf{V}(\mathbf{X}_{t+\frac{1}{2}})\|^2] < \infty$.*

*Proof.* Since $\hat{\mathbf{V}}_{1/2} = 0$, we have $\mathbf{X}_{3/2} = \mathbf{X}_1$ and with $\mathbf{X}_2 = \mathbf{X}_1 - \eta_2\hat{\mathbf{V}}_{3/2}$ we obtain

$$\|\mathbf{X}_2 - \mathbf{x}_\star\|^2 = \|\mathbf{X}_1 - \mathbf{x}_\star\|^2 - 2\eta_2\langle\hat{\mathbf{V}}_{3/2}, \mathbf{X}_{3/2} - \mathbf{x}_\star\rangle + \eta_2^2\|\hat{\mathbf{V}}_{3/2}\|^2.$$

Taking expectation then gives

$$\mathbb{E}[\|\mathbf{X}_2 - \mathbf{x}_\star\|^2] = \mathbb{E}[\|\mathbf{X}_1 - \mathbf{x}_\star\|^2 - 2\eta_2\langle\mathbf{V}(\mathbf{X}_{3/2}), \mathbf{X}_{3/2} - \mathbf{x}_\star\rangle + \eta_2^2\|\hat{\mathbf{V}}_{3/2}\|^2]$$
$$\le \mathbb{E}[\|\mathbf{X}_1 - \mathbf{x}_\star\|^2 + \eta_2^2\|\hat{\mathbf{V}}_{3/2}\|^2], \tag{15}$$

where we have used Assumption 2 to deduce that $\langle\mathbf{V}(\mathbf{X}_{3/2}), \mathbf{X}_{3/2}-\mathbf{x}_\star\rangle \ge 0$. Taking total expectation of the inequality of Lemma 5, summing from $t = 2$ to $T$, and further adding (15) gives

$$\underbrace{\mathbb{E}\left[\|\mathbf{X}_{T+1} - \mathbf{x}_\star\|^2 + \sum_{t=2}^{T}\gamma_t\eta_{t+1}(\|\mathbf{V}(\mathbf{X}_{t+\frac{1}{2}})\|^2 + \|\mathbf{V}(\mathbf{X}_{t-\frac{1}{2}})\|^2)\right]}_{(A)}$$

$$\le \mathbb{E}\left[\|\mathbf{X}_1 - \mathbf{x}_\star\|^2 + \sum_{t=1}^{T}(\eta_{t+1})^2\|\hat{\mathbf{V}}_{t+\frac{1}{2}}\|^2 + \sum_{t=2}^{T}3\gamma_t\eta_{t+1}(\eta_t^2 + \gamma_t^2)NL^2\|\hat{\mathbf{V}}_{t-\frac{1}{2}}\|^2\right.$$

$$\left. + \sum_{t=2}^{T}3\gamma_t\eta_{t+1}(\gamma_{t-1})^2NL^2\|\hat{\mathbf{V}}_{t-\frac{3}{2}}\|^2 + \sum_{t=2}^{T}\left(\gamma_t^2\eta_{t+1} + N\eta_{t+1}(\eta_t + \gamma_t)^2\right)L\|\boldsymbol{\xi}_{t-\frac{1}{2}}\|^2\right].$$

We use Assumption 3 to bound the noise terms. For example, we have

$$\mathbb{E}[\|\boldsymbol{\xi}_{t+\frac{1}{2}}\|^2] = \sum_{i=1}^{N}\mathbb{E}[\|\xi_{t+\frac{1}{2}}^i\|^2] \le \sum_{i=1}^{N}(\sigma_A^2 + \sigma_M^2\|V^i(\mathbf{x}_t)\|^2) = \sigma_M^2\|\mathbf{V}(\mathbf{X}_{t+\frac{1}{2}})\|^2 + N\sigma_A^2. \tag{16}$$

Subsequently,

$$\mathbb{E}[(\eta_{t+1})^2\|\hat{\mathbf{V}}_{t+\frac{1}{2}}\|^2] = (\eta_{t+1})^2\,\mathbb{E}[\|\mathbf{V}(\mathbf{X}_{t+\frac{1}{2}})\|^2 + \|\boldsymbol{\xi}_{t+\frac{1}{2}}\|^2]$$
$$\le (\eta_{t+1})^2\left(\mathbb{E}[(1 + \sigma_M^2)\|\mathbf{V}(\mathbf{X}_{t+\frac{1}{2}})\|^2] + N\sigma_A^2\right).$$

Along with the fact that the learning rates are non-increasing and $\eta_s \leq \gamma_s$ for all $s \in \mathbb{N}$, we get

$$(A) \leq \mathbb{E}\left[\|\mathbf{X}_1 - \mathbf{x}_\star\|^2 + \sum_{t=1}^{T}(\eta_{t+1})^2\left((1 + \sigma_M^2)\|\mathbf{V}(\mathbf{X}_{t+\frac{1}{2}})\|^2 + N\sigma_A^2\right)\right.$$

$$+ \sum_{t=2}^{T}\left(6\gamma_t^3\eta_{t+1}NL^2(1+\sigma_M^2) + \gamma_t^2\eta_{t+1}(4N+1)L\sigma_M^2\right)\|\mathbf{V}(\mathbf{X}_{t-\frac{1}{2}})\|^2$$

$$+ \sum_{t=2}^{T}\left(6\gamma_t^3\eta_{t+1}NL^2 + \gamma_t^2\eta_{t+1}(4N+1)L\right)N\sigma_A^2$$

$$\left.+ \sum_{t=3}^{T}3(\gamma_{t-1})^3\eta_t NL^2\left((1+\sigma_M^2)\|\mathbf{V}(\mathbf{X}_{t-\frac{3}{2}})\|^2 + N\sigma_A^2\right)\right]. \tag{17}$$

Re-indexing the summations and adding positive terms to the right-hand side (RHS) of the inequality, we deduce

$$(A) \leq \mathbb{E}\left[\|\mathbf{X}_1 - \mathbf{x}_\star\|^2 + \sum_{t=2}^{T}\left(9\gamma_t^3\eta_{t+1}NL^2(1+\sigma_M^2) + \gamma_t^2\eta_{t+1}(4N+1)L\sigma_M^2\right)\|\mathbf{V}(\mathbf{X}_{t-\frac{1}{2}})\|^2\right.$$

$$+ \sum_{t=1}^{T}(\eta_{t+1})^2(1+\sigma_M^2)\|\mathbf{V}(\mathbf{X}_{t+\frac{1}{2}})\|^2$$

$$\left.+ \sum_{t=1}^{T}\left((\eta_{t+1})^2 + 9\gamma_t^3\eta_{t+1}NL^2 + \gamma_t^2\eta_{t+1}(4N+1)L\right)N\sigma_A^2\right].$$

On the other hand, we have

$$(A) = \|\mathbf{X}_{T+1} - \mathbf{x}_\star\|^2 - \gamma_1\eta_2\|\mathbf{V}(\mathbf{X}_{3/2})\|^2$$

$$+ \sum_{t=1}^{T}\gamma_t\eta_{t+1}\,\mathbb{E}[\|\mathbf{V}(\mathbf{X}_{t+\frac{1}{2}})\|^2] + \sum_{t=2}^{T}\gamma_t\eta_{t+1}\,\mathbb{E}[\|\mathbf{V}(\mathbf{X}_{t-\frac{1}{2}})\|^2].$$

Combining the above two (in)equalities, rearranging, and using $\mathbf{X}_{3/2} = \mathbf{X}_1$ leads to

$$\mathbb{E}[\|\mathbf{X}_{T+1} - \mathbf{x}_\star\|^2] + \sum_{t=1}^{T}\gamma_t\eta_{t+1}\left(1 - \frac{(1+\sigma_M^2)\eta_{t+1}}{\gamma_t}\right)\mathbb{E}[\|\mathbf{V}(\mathbf{X}_{t+\frac{1}{2}})\|^2]$$

$$+ \sum_{t=2}^{T}\gamma_t\eta_{t+1}(1 - a_t(1+\sigma_M^2) - b_t\sigma_M^2)\,\mathbb{E}[\|\mathbf{V}(\mathbf{X}_{t-\frac{1}{2}})\|^2]$$

$$\leq \|\mathbf{X}_1 - \mathbf{x}_\star\|^2 + \gamma_1\eta_2\|V(\mathbf{X}_1)\|^2 + \sum_{t=1}^{T}\gamma_t\eta_{t+1}\left(\frac{\eta_{t+1}}{\gamma_t} + a_t + b_t\right)N\sigma_A^2,$$

where $a_t = 9\gamma_t^2 NL^2$ and $b_t = \gamma_t(4N+1)L$. To conclude, we notice that with the learning rate choices of Theorem 1, it always holds $1 - (1+\sigma_M^2)(\eta_{t+1}/\gamma_t) \geq 1/2$, $1 - a_t(1+\sigma_M^2) - b_t\sigma_M^2 \geq 0$, and $\sum_{t=1}^{+\infty}\gamma_t\eta_{t+1}(\eta_{t+1}/\gamma_t + a_t + b_t)N\sigma_A^2 < +\infty$. □

From Proposition 5 we obtain immediately the bounds on $\sum_{t=1}^{T}\mathbb{E}[\|\mathbf{V}(X_{t+\frac{1}{2}})\|^2]$ of OG+ as claimed in Section 6.

**Theorem 8.** *Let Assumptions 1–3 hold and all players run (OG+) with non-increasing learning rate sequences $(\gamma_t)_{t\in\mathbb{N}}$ and $(\eta_t)_{t\in\mathbb{N}}$ satisfying (4). We have*

(a) *If there exists $q \in [0, 1/4]$ such that $\gamma_t = \mathcal{O}(1/(t^{\frac{1}{4}}\sqrt{\log t}))$, $\gamma_t = \Omega(1/t^{\frac{1}{2}-q})$, and $\eta_t = \Theta(1/(\sqrt{t}\log t))$, then*

$$\sum_{t=1}^{T}\mathbb{E}[\|\mathbf{V}(X_{t+\frac{1}{2}})\|^2] = \tilde{\mathcal{O}}\left(T^{1-q}\right)$$

(b) *If the noise is multiplicative (i.e., $\sigma_A = 0$) and the learning rates are constant $\gamma_t \equiv \gamma$, $\eta_t \equiv \eta$, then*

$$\sum_{t=1}^{T} \mathbb{E}[\|\mathbf{V}(\mathbf{X}_{t+\frac{1}{2}})\|^2] \leq \frac{2 \operatorname{dist}(\mathbf{X}_1, \mathcal{X}_\star)^2}{\gamma \eta} + 2\|\mathbf{V}(\mathbf{X}_1)\|^2.$$

*In particular, if the equalities hold in (4), then the above is in $\mathcal{O}(N^3 L^2 (1 + \sigma_M^2)^3)$.*

*Proof.* Let $\mathbf{x}_\star = \Pi_{\mathcal{X}_\star}(\mathbf{X}_1)$. By the choice of our learning rates, the constant

$$C := \|\mathbf{X}_1 - \mathbf{x}_\star\|^2 + \gamma_1 \eta_2 \|V(\mathbf{X}_1)\|^2 + \sum_{t=1}^{+\infty} (9\gamma_t^3 \eta_{t+1} N L^2 + \gamma_t^2 \eta_{t+1}(4N+1)L + (\eta_{t+1})^2)N\sigma_A^2.$$

is finite. In addition, from Proposition 5 we know tat

$$\sum_{t=1}^{T} \gamma_t \eta_{t+1} \mathbb{E}[\|\mathbf{V}(\mathbf{X}_{t+\frac{1}{2}})\|^2] \leq 2C.$$

On the other hand since the learning rates are non-increasing, it holds

$$\sum_{t=1}^{T} \gamma_t \eta_{t+1} \mathbb{E}[\|\mathbf{V}(\mathbf{X}_{t+\frac{1}{2}})\|^2] \geq \gamma_{T+1}\eta_{T+1} \sum_{t=1}^{T} \mathbb{E}[\|\mathbf{V}(\mathbf{X}_{t+\frac{1}{2}})\|^2].$$

As a consequence,

$$\sum_{t=1}^{T} \mathbb{E}[\|\mathbf{V}(\mathbf{X}_{t+\frac{1}{2}})\|^2] \leq \frac{2C}{\gamma_{T+1}\eta_{T+1}}. \tag{18}$$

The results are then immediate from our choice of learning rates. $\square$

**Remark 3.** In the estimation of (b) we use $\operatorname{dist}(\mathbf{X}_1, \mathcal{X}_\star)^2 = \mathcal{O}(N)$ and $1/\gamma = \mathcal{O}(NL(1+\sigma_M^2))$. We can get improved dependence on $N$ if the noises of the players are supposed to be mutually independent conditioned on the past. In fact, in this case we only require $\gamma \leq \min \left( \frac{1}{3L\sqrt{2N(1+\sigma_M^2)}}, \frac{1}{8L\sigma_M^2} \right)$.

**Bounding Linearized Regret.** We proceed to bound the linearized regret. The following lemma is a direct consequence of the individual quasi-descent inequality of Lemma 4.

**Lemma 9** (Bound on linearized regret)**.** *Let Assumptions 1–3 hold and all players run (OG+) with learning rates described in Theorem 1. Then, for all $i \in \mathcal{N}$, $T \in \mathbb{N}$, and $p^i \in \mathcal{X}^i$, we have*

$$\sum_{t=1}^{T} \mathbb{E}[\langle V^i(\mathbf{X}_{t+\frac{1}{2}}), X_{t+\frac{1}{2}}^i - p^i \rangle] \leq \mathbb{E}\left[ \frac{\|X_1^i - p^i\|^2}{2\eta_2} + \sum_{t=2}^{T} \left( \frac{1}{2\eta_{t+1}} - \frac{1}{2\eta_t} \right) \|X_t^i - p^i\|^2 \right.$$

$$\left. + \sum_{t=1}^{T} \left( \frac{3\gamma_t}{4} \|\mathbf{V}(\mathbf{X}_{t+\frac{1}{2}})\|^2 + \frac{a_t \sigma_A^2}{2} \right) \right],$$

*where $a_t = 9\gamma_t^3 L^2 N + \gamma_t^2(4N+1)L + \eta_{t+1}$.*

*Proof.* Applying Lemma 4, dividing both sides of (10) by $\eta_{t+1}$, rearranging, taking total expectation, and using Assumption 3, we get

$$\mathbb{E}[2\langle V^i(\mathbf{X}_{t+\frac{1}{2}}), X_{t+\frac{1}{2}}^i - p^i \rangle]$$

$$\leq \mathbb{E}\left[ \frac{\|X_t^i - p^i\|^2}{\eta_{t+1}} - \frac{\|X_{t+1}^i - p^i\|^2}{\eta_{t+1}} \right.$$

$$- \gamma_t(\|V^i(\mathbf{X}_{t+\frac{1}{2}})\|^2 + \|V^i(\mathbf{X}_{t-\frac{1}{2}})\|^2) + \gamma_t \|V^i(\mathbf{X}_{t+\frac{1}{2}}) - V^i(\mathbf{X}_{t-\frac{1}{2}})\|^2$$

$$+ \gamma_t^2 L\sigma_M^2 \|V^i(\mathbf{X}_{t-\frac{1}{2}})\|^2 + (\eta_t + \gamma_t)^2 L\sigma_M^2 \|\mathbf{V}(\mathbf{X}_{t-\frac{1}{2}})\|^2 + \eta_{t+1}(1 + \sigma_M^2)\|V^i(\mathbf{X}_{t+\frac{1}{2}})\|^2$$

$$\left. + \gamma_t^2 L\sigma_A^2 + (\eta_t + \gamma_t)^2 LN\sigma_A^2 + \eta_{t+1}\sigma_A^2 \right]$$

$$
\leq \mathbb{E}\left[ \frac{\|X_t^i - p^i\|^2}{\eta_{t+1}} - \frac{\|X_{t+1}^i - p^i\|^2}{\eta_{t+1}} \right.
$$

$$
+ 3\gamma_t^3 L^2 \|\hat{\mathbf{V}}_{t-\frac{1}{2}}\|^2 + 3\gamma_t \eta_t^2 L^2 \|\hat{\mathbf{V}}_{t-\frac{1}{2}}\|^2 + 3\gamma_t(\gamma_{t-1})^2 L^2 \|\hat{\mathbf{V}}_{t-\frac{3}{2}}\|^2
$$

$$
\left. + 5\gamma_t^2 L\sigma_M^2 \|\mathbf{V}(\mathbf{X}_{t-\frac{1}{2}})\|^2 + \eta_{t+1}(1+\sigma_M^2)\|\mathbf{V}(\mathbf{X}_{t+\frac{1}{2}})\|^2 + \gamma_t^2(4N+1)L\sigma_A^2 + \eta_{t+1}\sigma_A^2 \right]
$$

$$
\leq \mathbb{E}\left[ \frac{\|X_t^i - p^i\|^2}{\eta_t} + \left( \frac{1}{\eta_{t+1}} - \frac{1}{\eta_t} \right)\|X_t^i - p^i\|^2 - \frac{\|X_{t+1}^i - p^i\|^2}{\eta_{t+1}} \right.
$$

$$
+ \frac{\gamma_t}{2}\|\mathbf{V}(\mathbf{X}_{t+\frac{1}{2}})\|^2 + \frac{5\gamma_t}{6}\|\mathbf{V}(\mathbf{X}_{t-\frac{1}{2}})\|^2 + 3(\gamma_{t-1})^3 L^2\|\hat{\mathbf{V}}_{t-\frac{3}{2}}\|^2
$$

$$
\left. + 6\gamma_t^3 L^2 N\sigma_A^2 + \gamma_t^2(4N+1)L\sigma_A^2 + \eta_{t+1}\sigma_A^2 \right].
$$

In the last inequality we have used $\eta_t^2 \leq \gamma_t^2 \leq 1/(18L^2(1+\sigma_M^2))$, $5\gamma_t L\sigma_M^2 \leq \gamma_1(4N+1)L\sigma_M^2 \leq 1/2$ and $\eta_{t+1}(1+\sigma_M^2) \leq \eta_t(1+\sigma_M^2) \leq \gamma_t/2$. As for the $\|\hat{\mathbf{V}}_{t-\frac{3}{2}}\|^2$ term, we recall that $\hat{\mathbf{V}}_{1/2} = 0$ and otherwise its expectation can again be bounded using Assumption 3. Summing the above inequality from $t = 2$ to $T$ and dividing both sides by 2, we then obtain

$$
\sum_{t=2}^{T} \mathbb{E}[\langle V^i(\mathbf{X}_{t+\frac{1}{2}}), X_{t+\frac{1}{2}}^i - p^i \rangle] \leq \mathbb{E}\left[ \frac{\|X_2^i - p^i\|^2}{2\eta_2} + \sum_{t=2}^{T} \left( \frac{1}{2\eta_{t+1}} - \frac{1}{2\eta_t} \right)\|X_t^i - p^i\|^2 \right.
$$

$$
+ \sum_{t=2}^{T} \frac{1}{4}\left( \gamma_t\|\mathbf{V}(\mathbf{X}_{t+\frac{1}{2}})\|^2 + 2\gamma_t\|\mathbf{V}(\mathbf{X}_{t-\frac{1}{2}})\|^2 \right.
$$

$$
\left.\left. + 18\gamma_t^3 L^2 N\sigma_A^2 + \gamma_t^2(8N+2)L\sigma_A^2 + 2\eta_{t+1}\sigma_A^2 \right) \right].
\tag{19}
$$

For $t = 1$, since $X_{3/2}^i = X_1^i$ and $X_2^i = X_1^i - \eta_2 \hat{V}_{3/2}^i$, we have

$$
\|X_2^i - p^i\|^2 = \|X_t^1 - p^i\|^2 - 2\eta_2\langle \hat{V}_{3/2}^i, X_{3/2}^i - p^i \rangle + \eta_2^2\|\hat{V}_{3/2}^i\|^2.
$$

Taking expectation then gives

$$
\mathbb{E}[\|X_2^i - p^i\|^2] \leq \mathbb{E}[\|X_1^i - p^i\|^2 - 2\eta_2\langle V^i(\mathbf{X}_{3/2}), X_{3/2}^i - p^i \rangle + \eta_2^2(1+\sigma_M^2)\|V^i(\mathbf{X}_{3/2})\|^2 + \eta_2^2\sigma_A^2].
\tag{20}
$$

Combining (19) and (20) and bounding $\eta_2(1+\sigma_M^2)\|V^i(\mathbf{X}_{3/2})\|^2 \leq (\gamma_t/2)\|\mathbf{V}(\mathbf{X}_{3/2})\|^2$, we get the desired inequality. $\qquad\square$

With Lemma 9 and Proposition 5, we are now ready to prove our result concerning the regret of OG+. The main difficulty here consists in controlling the sum of $(1/(2\eta_{t+1}) - 1/(2\eta_t))\mathbb{E}[\|X_t^i - p^i\|^2]$ when the learning rates are not constant.

**Theorem 9.** *Let Assumptions 1–3 hold and all players run (OG+) with non-increasing learning rate sequences $(\gamma_t)_{t\in\mathbb{N}}$ and $(\eta_t)_{t\in\mathbb{N}}$ satisfying (4). For any $i \in \mathcal{N}$ and bounded set $\mathcal{K}^i \subset \mathcal{X}^i$ with $R \geq \sup_{p^i}\|X_1^i - p^i\|$, we have:*

*(a) If $\gamma_t = \mathcal{O}(1/(t^{\frac{1}{4}}\sqrt{\log t}))$ and $\eta_t = \Theta(1/(\sqrt{t}\log t))$, then*

$$
\max_{p^i\in\mathcal{K}^i} \mathbb{E}\left[ \sum_{t=1}^{T}\langle V^i(\mathbf{X}_{t+\frac{1}{2}}), X_{t+\frac{1}{2}}^i - p^i \rangle \right] = \tilde{\mathcal{O}}\left( \sqrt{T} \right).
$$

*(b) If the noise is multiplicative (i.e., $\sigma_A = 0$) and the learning rates are constant $\gamma_t \equiv \gamma$, $\eta_t \equiv \eta$, then*

$$
\max_{p^i\in\mathcal{K}^i} \mathbb{E}\left[ \sum_{t=1}^{T}\langle V^i(\mathbf{X}_{t+\frac{1}{2}}), X_{t+\frac{1}{2}}^i - p^i \rangle \right] \leq \frac{R^2}{2\eta} + \frac{2}{\eta}(\text{dist}(\mathbf{X}_1, \mathcal{X}_\star)^2 + \gamma\eta\|\mathbf{V}(\mathbf{X}_1)\|^2).
$$

*In particular, if the equalities hold in (4), the above is in $\mathcal{O}(N^2 L(1+\sigma_M^2)^2)$.*

*Proof.* Let $\mathbf{x}_\star = \Pi_{\mathcal{X}_\star}(\mathbf{X}_1)$ be the projection of $\mathbf{X}_1$ onto the solution set. For any $p^i \in \mathcal{K}^i$, it holds

$$\sum_{t=2}^{T} \left( \frac{1}{2\eta_{t+1}} - \frac{1}{2\eta_t} \right) \|X_t^i - p^i\|^2$$

$$\leq \sum_{t=2}^{T} \left( \frac{1}{\eta_{t+1}} - \frac{1}{\eta_t} \right) \left( \|X_t^i - x_\star^i\|^2 + \|x_\star^i - p^i\|^2 \right)$$

$$\leq \sum_{t=2}^{T} \left( \frac{1}{\eta_{t+1}} - \frac{1}{\eta_t} \right) \left( \|\mathbf{X}_t - \mathbf{x}_\star\|^2 + \|x_\star^i - X_1^i + X_1^i - p^i\|^2 \right)$$

$$\leq \left( \frac{1}{\eta_{T+1}} - \frac{1}{\eta_2} \right) \left( 2\|X_1^i - x_\star^i\|^2 + 2R^2 \right) + \sum_{t=2}^{T} \left( \frac{1}{\eta_{t+1}} - \frac{1}{\eta_t} \right) \|\mathbf{X}_t - \mathbf{x}_\star\|^2. \tag{21}$$

To proceed, with Proposition 5, we know that for $C$ defined in the proof of Theorem 8, we have for all $t \in \mathbb{N}$

$$\mathbb{E}[\|\mathbf{X}_{t+1} - \mathbf{x}_\star\|^2] + \sum_{s=1}^{t} \frac{\gamma_s \eta_{s+1}}{2} \mathbb{E}[\|\mathbf{V}(\mathbf{X}_{s+\frac{1}{2}})\|^2] \leq C \tag{22}$$

We can therefore write

$$\sum_{t=2}^{T} \left( \frac{1}{\eta_{t+1}} - \frac{1}{\eta_t} \right) \mathbb{E}[\|\mathbf{X}_t - \mathbf{x}_\star\|^2] \leq \sum_{t=2}^{T} \left( \frac{1}{\eta_{t+1}} - \frac{1}{\eta_t} \right) C \leq \frac{C}{\eta_{T+1}}. \tag{23}$$

Since $\eta_{T+1} \leq \eta_{t+1}$ for all $t \leq T$. From (22) we also deduce

$$\sum_{t=1}^{T} \gamma_t \mathbb{E}[\|\mathbf{V}(\mathbf{X}_{t+\frac{1}{2}})\|^2] \leq \frac{1}{\eta_{T+1}} \sum_{t=1}^{T} \gamma_t \eta_{t+1} \mathbb{E}[\|\mathbf{V}(\mathbf{X}_{t+\frac{1}{2}})\|^2] \leq \frac{2C}{\eta_{T+1}}. \tag{24}$$

Plugging (21), (23), and (24) into Lemma 9, we obtain

$$\sum_{t=1}^{T} \mathbb{E}[\langle V^i(\mathbf{X}_{t+\frac{1}{2}}), X_{t+\frac{1}{2}}^i - p^i \rangle] \leq \mathbb{E}\left[ \frac{2\operatorname{dist}(\mathbf{X}_1, \mathcal{X}_\star)^2 + 2R^2}{\eta_{T+1}} + \frac{3C}{\eta_{T+1}} + \sum_{t=1}^{T} \frac{a_t \sigma_A^2}{2} \right].$$

The result is now immediate from $\gamma_t = \mathcal{O}(1/(t^{\frac{1}{4}}\sqrt{\log t}))$ and $\eta_t = \Theta(1/(\sqrt{t}\log t))$.

(b) Let $p^i \in \mathcal{K}^i$. With $\sigma_A^2 = 0$, constant learning rates, and $\|X_1^i - p^i\| \leq R^2$, Lemma 9 gives

$$\sum_{t=1}^{T} \mathbb{E}[\langle V^i(\mathbf{X}_{t+\frac{1}{2}}), X_{t+\frac{1}{2}}^i - p^i \rangle] \leq \mathbb{E}\left[ \frac{R^2}{2\eta} + \sum_{t=1}^{T} \frac{3\gamma}{4} \|\mathbf{V}(\mathbf{X}_{t+\frac{1}{2}})\|^2 \right],$$

We conclude immediately with the help of Theorem 8(b). $\qquad \square$

## F.2 Bounds for OptDA+

For the analysis of OptDA+, we first establish two preliminary bounds respectively for the linearized regret and for the sum of the squared operator norms. These bounds are used later for deriving more refined bounds in the non-adaptive and the adaptive case. We use the notation $\eta_1^i = \eta_2^i$.

**Lemma 10** (Bound on linearized regret). *Let Assumptions 1 and 3 hold and all players run (OptDA+) with non-increasing learning rates satisfying Assumption 5 and $\boldsymbol{\eta}_t \leq \boldsymbol{\gamma}_t$ for all $t \in \mathbb{N}$. Then, for all $i \in \mathcal{N}$, $T \in \mathbb{N}$, and $p^i \in \mathcal{X}^i$, we have*

$$\mathbb{E}\left[ \sum_{t=1}^{T} \langle V^i(\mathbf{X}_{t+\frac{1}{2}}), X_{t+\frac{1}{2}}^i - p^i \rangle \right] \leq \mathbb{E}\left[ \frac{\|X_1^i - p^i\|^2}{2\eta_{T+1}^i} + \frac{1}{2} \sum_{t=1}^{T} \eta_t^i \|\hat{V}_{t+\frac{1}{2}}^i\|^2 \right.$$

$$+ \sum_{t=2}^{T} \gamma_t^i L^2 \left( 3\|\hat{\mathbf{V}}_{t-\frac{1}{2}}\|_{\gamma_t^2}^2 + \frac{3}{2} \|\mathbf{X}_t - \mathbf{X}_{t-1}\|^2 \right)$$

$$\left. + \frac{1}{2} \sum_{t=2}^{T} ((\gamma_t^i)^2 L \|\xi_{t-\frac{1}{2}}^i\|^2 + 4L \|\boldsymbol{\xi}_{t-\frac{1}{2}}\|_{\gamma_t^2}^2) \right]. \tag{25}$$

*Proof.* Applying Lemma 6, dropping non-positive terms on the RHS of (12), using

$$\min\left(-\frac{\|X_t^i - X_{t+1}^i\|^2}{2\eta_t^i} + \eta_t^i \|\hat{V}_{t+\frac{1}{2}}^i\|^2, 0\right) \leq 0$$

and taking total expectation gives

$$\mathbb{E}\left[\frac{\|X_{t+1}^i - p^i\|^2}{\eta_{t+1}^i}\right] \leq \mathbb{E}\left[\frac{\|X_t^i - p^i\|^2}{\eta_t^i} + \left(\frac{1}{\eta_{t+1}^i} - \frac{1}{\eta_t^i}\right)\|X_1^i - p^i\|^2\right.$$
$$- 2\langle V^i(\mathbf{X}_{t+\frac{1}{2}}), X_{t+\frac{1}{2}}^i - p^i\rangle + \gamma_t^i\|V^i(\mathbf{X}_{t+\frac{1}{2}}) - V^i(\mathbf{X}_{t-\frac{1}{2}})\|^2$$
$$\left. + (\gamma_t^i)^2 L\|\xi_{t-\frac{1}{2}}^i\|^2 + L\|\boldsymbol{\xi}_{t-\frac{1}{2}}\|^2_{(\boldsymbol{\eta}_t + \boldsymbol{\gamma}_t)^2} + \eta_t^i\|\hat{V}_{t+\frac{1}{2}}^i\|^2\right]. \qquad (26)$$

The above inequality holds for $t \geq 2$. As for $t = 1$, we notice that with $X_2^i = X_1^i - \eta_2^i\hat{V}_{3/2}^i$, we have in fact

$$\|X_2^i - p^i\|^2 = \|X_1^i - p^i\|^2 - 2\eta_2^i\langle\hat{V}_{3/2}^i, X_1^i - p^i\rangle + (\eta_2^i)^2\|\hat{V}_{3/2}^i\|^2.$$

As $X_{3/2}^i = X_1^i = 0$ and $\eta_1^i = \eta_2^i$, the above implies

$$\mathbb{E}\left[\langle V^i(X_{3/2}^i), X_{3/2}^i - p^i\rangle\right] = \mathbb{E}\left[\frac{\|X_1^i - p^i\|^2}{2\eta_2^i} - \frac{\|X_2^i - p^i\|^2}{2\eta_2^i} + \frac{\eta_1^i\|\hat{V}_{3/2}^i\|^2}{2}\right]. \qquad (27)$$

Summing (26) from $t = 2$ to $T$, dividing by 2, adding (27), and using $\boldsymbol{\eta}_t \leq \boldsymbol{\gamma}_t$ leads to

$$\sum_{t=1}^{T}\mathbb{E}[\langle V^i(\mathbf{X}_{t+\frac{1}{2}}), X_{t+\frac{1}{2}}^i - p^i\rangle] \leq \frac{1}{2}\mathbb{E}\left[\frac{\|X_1^i - p^i\|^2}{\eta_{T+1}^i} + \sum_{t=1}^{T}\eta_t^i\|\hat{V}_{t+\frac{1}{2}}^i\|^2\right.$$
$$+ \sum_{t=2}^{T}\gamma_t^i\|V^i(\mathbf{X}_{t+\frac{1}{2}}) - V^i(\mathbf{X}_{t-\frac{1}{2}})\|^2$$
$$\left. + \sum_{t=2}^{T}((\gamma_t^i)^2 L\|\xi_{t-\frac{1}{2}}^i\|^2 + 4L\|\boldsymbol{\xi}_{t-\frac{1}{2}}\|^2_{\boldsymbol{\gamma}_t^2})\right].$$

Similar to (11), we can bound the difference term by

$$\|V^i(\mathbf{X}_{t+\frac{1}{2}}) - V^i(\mathbf{X}_{t-\frac{1}{2}})\|^2 \leq 3L^2\|\hat{\mathbf{V}}_{t-\frac{1}{2}}\|^2_{\boldsymbol{\gamma}_t^2} + 3L^2\|\hat{\mathbf{V}}_{t-\frac{3}{2}}\|^2_{(\boldsymbol{\gamma}_{t-1})^2} + 3L^2\|\mathbf{X}_t - \mathbf{X}_{t-1}\|^2.$$

Combining the above two inequalities and using $\hat{\mathbf{V}}_{1/2} = 0$ gives the desired inequality. $\qquad \square$

**Lemma 11** (Bound on sum of squared norms). *Let Assumptions 1–3 hold and all players run (OptDA+) with non-increasing learning rates satisfying Assumption 5 and $\boldsymbol{\eta}_t \leq \boldsymbol{\gamma}_t$ for all $t \in \mathbb{N}$. Then, for all $T \in \mathbb{N}$ and $\mathbf{x}_\star \in \mathcal{X}_\star$, we have*

$$\sum_{t=2}^{T}\mathbb{E}[\|\mathbf{V}(\mathbf{X}_{t+\frac{1}{2}})\|^2_{\boldsymbol{\gamma}_t} + \|\mathbf{V}(\mathbf{X}_{t-\frac{1}{2}})\|^2_{\boldsymbol{\gamma}_t}]$$

$$\leq \mathbb{E}\left[\|\mathbf{X}_1 - \mathbf{x}_\star\|^2_{1/\boldsymbol{\eta}_{T+1}} + \sum_{t=1}^{T}\left(3\|\mathbf{V}(\mathbf{X}_t) - \mathbf{V}(\mathbf{X}_{t+1})\|^2_{\boldsymbol{\gamma}_t} - \|\mathbf{X}_t - \mathbf{X}_{t+1}\|^2_{1/(2\boldsymbol{\eta}_t)}\right)\right.$$
$$\left. + \sum_{t=2}^{T}6\|\boldsymbol{\gamma}_t\|_1 L^2\|\hat{\mathbf{V}}_{t-\frac{1}{2}}\|^2_{\boldsymbol{\gamma}_t^2} + \sum_{t=2}^{T}(4N + 1)L\|\boldsymbol{\xi}_{t-\frac{1}{2}}\|^2_{\boldsymbol{\gamma}_t^2} + \sum_{t=1}^{T}2\|\hat{\mathbf{V}}_{t+\frac{1}{2}}\|^2_{\boldsymbol{\eta}_t}\right]. \qquad (28)$$

*Proof.* This is a direct consequence of [Lemma 7]. In fact, taking total expectation of (13) and summing from $t = 2$ to $T$ gives already

$$\sum_{t=2}^{T} \mathbb{E}[\|\mathbf{V}(\mathbf{X}_{t+\frac{1}{2}})\|_{\gamma_t}^2 + \|\mathbf{V}(\mathbf{X}_{t-\frac{1}{2}})\|_{\gamma_t}^2]$$

$$\leq \mathbb{E}\left[\|\mathbf{X}_2 - \mathbf{x}_\star\|_{1/\eta_2}^2 + \|\mathbf{X}_1 - \mathbf{x}_\star\|_{1/\eta_{T+1} - 1/\eta_2}^2\right.$$

$$+ \sum_{t=2}^{T}(3\|\mathbf{V}(\mathbf{X}_t) - \mathbf{V}(\mathbf{X}_{t-1})\|_{\gamma_t}^2 - \|\mathbf{X}_t - \mathbf{X}_{t+1}\|_{1/(2\eta_t)}^2)$$

$$\left. + \sum_{t=2}^{T} 6\|\gamma_t\|_1 L^2 \|\hat{\mathbf{V}}_{t-\frac{1}{2}}\|_{\gamma_t^2}^2 + \sum_{t=2}^{T}(4N+1)L\|\xi_{t-\frac{1}{2}}\|_{\gamma_t^2}^2 + \sum_{t=2}^{T} 2\|\hat{\mathbf{V}}_{t+\frac{1}{2}}\|_{\eta_t}^2\right]. \tag{29}$$

We have in particular used $\hat{\mathbf{V}}_{1/2} = 0$ to bound

$$\sum_{t=2}^{T} 3L^2(\|\gamma_t\|_1\|\hat{\mathbf{V}}_{t-\frac{1}{2}}\|_{\gamma_t^2}^2 + \|\gamma_{t-1}\|_1\|\hat{\mathbf{V}}_{t-\frac{3}{2}}\|_{(\gamma_{t-1})^2}^2)$$

$$= \sum_{t=2}^{T} 3\|\gamma_t\|_1 L^2 \|\hat{\mathbf{V}}_{t-\frac{1}{2}}\|_{\gamma_t^2}^2 + \sum_{t=3}^{T} \|\gamma_t\|_1 3NL^2\|\hat{\mathbf{V}}_{t-\frac{1}{2}}\|_{\gamma_t^2}^2$$

$$\leq \sum_{t=2}^{T} 6\|\gamma_t\|_1 L^2 \|\hat{\mathbf{V}}_{t-\frac{1}{2}}\|_{\gamma_t^2}^2.$$

To obtain (28), we further bound

$$\sum_{t=2}^{T} 3\|\mathbf{V}(\mathbf{X}_t) - \mathbf{V}(\mathbf{X}_{t-1})\|_{\gamma_t}^2 = \sum_{t=1}^{T-1} 3\|\mathbf{V}(\mathbf{X}_t) - \mathbf{V}(\mathbf{X}_{t+1})\|_{\gamma_{t+1}}^2 \leq \sum_{t=1}^{T} 3\|\mathbf{V}(\mathbf{X}_t) - \mathbf{V}(\mathbf{X}_{t+1})\|_{\gamma_t}^2 \tag{30}$$

For $t = 1$, we use (27) with $p^i \leftarrow x_\star^i$; that is

$$\frac{\|X_2^i - x_\star^i\|^2}{\eta_2^i} = \frac{\|X_1^i - x_\star^i\|^2}{\eta_2^i} - 2\langle V^i(\mathbf{X}_{3/2}) + \xi_{3/2}^i, X_1^i - x_\star^i\rangle + \eta_1^i\|\hat{V}_{3/2}^i\|^2.$$

Since $\mathbf{X}_{3/2} = \mathbf{X}_1$, summing the above inequality from $i = 1$ to $N$ leads to

$$\|\mathbf{X}_2 - \mathbf{x}_\star\|_{1/\eta_2}^2 = \|\mathbf{X}_1 - \mathbf{x}_\star\|_{1/\eta_2}^2 - 2\langle \mathbf{V}(\mathbf{X}_{3/2}) + \xi_{3/2}, \mathbf{X}_{3/2} - \mathbf{x}_\star\rangle + \|\hat{\mathbf{V}}_{3/2}\|_{\eta_1}^2. \tag{31}$$

[Assumptions 2] and [3] together ensure

$$\mathbb{E}[\langle \mathbf{V}(\mathbf{X}_{3/2}) + \xi_{3/2}, \mathbf{X}_{3/2} - \mathbf{x}_\star\rangle] = \langle \mathbf{V}(\mathbf{X}_{3/2}), \mathbf{X}_{3/2} - \mathbf{x}_\star\rangle \geq 0.$$

Subsequently,

$$\mathbb{E}[\|\mathbf{X}_2 - \mathbf{x}_\star\|_{1/\eta_2}^2] \leq \mathbb{E}[\|\mathbf{X}_1 - \mathbf{x}_\star\|_{1/\eta_2}^2 + \|\hat{V}_{3/2}^i\|_{\eta_2}^2]$$

$$\leq \mathbb{E}[\|\mathbf{X}_1 - \mathbf{x}_\star\|_{1/\eta_2}^2 + 2\|\hat{V}_{3/2}^i\|_{\eta_1}^2 - \|\mathbf{X}_1 - \mathbf{X}_2\|_{(1/2\eta_1)}^2]. \tag{32}$$

Combining (29), (30), and (32) gives exactly (28). $\qquad\square$

### F.2.1 Dedicated Analysis for Non-Adaptive OptDA+

In this part, we show how the non-adaptive learning rates suggested in [Theorem 2] helps to achieve small regret and lead to fast convergence of the norms of the payoff gradients.

**Proposition 6** (Bound on sum of squared norms)**.** *Let [Assumptions 1–3] hold and all players run* (OptDA+) *with learning rates described in [Theorem 2]. Then, for all $T \in \mathbb{N}$ and $\mathbf{x}_\star \in \mathcal{X}_\star$, we have*

$$\frac{1}{2}\sum_{s=1}^{T} \mathbb{E}[\|\mathbf{V}(\mathbf{X}_{t+\frac{1}{2}})\|_{\gamma_t}^2] + \sum_{t=1}^{T} 21\|\gamma_1\|_\infty NL^2 \, \mathbb{E}[\|\mathbf{X}_t - \mathbf{X}_{t+1}\|^2]$$

$$\leq \|\mathbf{X}_1 - \mathbf{x}_\star\|_{1/\eta_{T+1}}^2 + \|\mathbf{V}(\mathbf{X}_1)\|_{\gamma_1}^2 + \sum_{t=1}^{T}\left(6\|\gamma_t\|_\infty^3 NL^2 + \|\gamma_t\|_\infty^2(4N+1)L + 2\|\eta_t\|_\infty\right)N\sigma_A^2$$

*Proof.* We first apply Lemma 11 to obtain (28). We bound the expectations of the following three terms separately.

$$A_t = 3\|\mathbf{V}(\mathbf{X}_t) - \mathbf{V}(\mathbf{X}_{t+1})\|_{\boldsymbol{\gamma}_t}^2 - \|\mathbf{X}_t - \mathbf{X}_{t+1}\|_{1/(2\boldsymbol{\eta}_t)}^2,$$
$$B_t = 6\|\boldsymbol{\gamma}_t\|_1 L^2 \|\hat{\mathbf{V}}_{t-\frac{1}{2}}\|_{\boldsymbol{\gamma}_t^2}^2 + (4N+1)L\|\boldsymbol{\xi}_{t-\frac{1}{2}}\|_{\boldsymbol{\gamma}_t^2}^2, \quad C_t = 2\|\hat{\mathbf{V}}_{t+\frac{1}{2}}\|_{\boldsymbol{\eta}_t}^2.$$

To bound $A_t$, we first use $\boldsymbol{\eta}_t \leq \boldsymbol{\gamma}_t/(4(1+\sigma_M^2)) \leq \|\boldsymbol{\gamma}_1\|_\infty/(4(1+\sigma_M^2))$ to get

$$\|\mathbf{X}_t - \mathbf{X}_{t+1}\|_{1/(2\boldsymbol{\eta}_t)}^2 \geq \frac{2(1+\sigma_M^2)}{\|\boldsymbol{\gamma}_1\|_\infty}\|\mathbf{X}_t - \mathbf{X}_{t+1}\|^2.$$

Moreover, with $\|\boldsymbol{\gamma}_1\|_\infty^2 \leq 1/(12NL^2(1+\sigma_M^2))$ we indeed have

$$\frac{2(1+\sigma_M^2)}{\|\boldsymbol{\gamma}_1\|_\infty} \geq 24NL^2(1+\sigma_M^2)^2\|\boldsymbol{\gamma}_1\|_\infty \geq 24NL^2\|\boldsymbol{\gamma}_1\|_\infty.$$

On the other hand, with the Lipschitz continuity of $(V^i)_{i\in\mathcal{N}}$ it holds

$$3\|\mathbf{V}(\mathbf{X}_t) - \mathbf{V}(\mathbf{X}_{t+1})\|_{\boldsymbol{\gamma}_t}^2 \leq \sum_{i=1}^N 3\gamma_t^i L^2\|\mathbf{X}_t - \mathbf{X}_{t+1}\|^2 \leq 3\|\boldsymbol{\gamma}_1\|_\infty NL^2\|\mathbf{X}_t - \mathbf{X}_{t+1}\|^2.$$

Combining the above inequalities we deduce that $A_t \leq -21\|\boldsymbol{\gamma}_1\|_\infty NL^2\|\mathbf{X}_t - \mathbf{X}_{t+1}\|^2$ and accordingly

$$\mathbb{E}[A_t] \leq \mathbb{E}[-21\|\boldsymbol{\gamma}_1\|_\infty NL^2\|\mathbf{X}_t - \mathbf{X}_{t+1}\|^2]. \tag{33}$$

We proceed to bound $\mathbb{E}[B_t]$. The exploration learning rates $\boldsymbol{\gamma}_t$ being $\mathcal{F}_{t-1}$-measurable, using Assumption 3 and the law of total expectation, we get

$$\mathbb{E}[B_t] = \mathbb{E}\left[\mathbb{E}_{t-1}[6\|\boldsymbol{\gamma}_t\|_1 L^2\|\hat{\mathbf{V}}_{t-\frac{1}{2}}\|_{\boldsymbol{\gamma}_t^2}^2 + (4N+1)L\|\boldsymbol{\xi}_{t-\frac{1}{2}}\|_{\boldsymbol{\gamma}_t^2}^2]\right]$$

$$= \mathbb{E}\left[\sum_{i=1}^N \left(6\|\boldsymbol{\gamma}_t\|_1(\gamma_t^i)^2 L^2\,\mathbb{E}_{t-1}[\|\hat{V}_{t-\frac{1}{2}}^i\|^2] + (\gamma_t^i)^2(4N+1)L\,\mathbb{E}_{t-1}[\|\xi_{t-\frac{1}{2}}^i\|^2]\right)\right]$$

$$\leq \mathbb{E}\left[6\|\boldsymbol{\gamma}_t\|_\infty^2 NL^2(1+\sigma_M^2)\|\mathbf{V}(\mathbf{X}_{t-\frac{1}{2}})\|_{\boldsymbol{\gamma}_t}^2 + \|\boldsymbol{\gamma}_t\|_\infty(4N+1)L\sigma_M^2\|\mathbf{V}(\mathbf{X}_{t-\frac{1}{2}})\|_{\boldsymbol{\gamma}_t}^2\right.$$

$$\left. + (6\|\boldsymbol{\gamma}_t\|_\infty^3 NL^2 + \|\boldsymbol{\gamma}_t\|_\infty^2(4N+1)L)N\sigma_A^2\right]. \tag{34}$$

Similarly, $\boldsymbol{\eta}_{t+1}$ being deterministic and in particular $\mathcal{F}_t$-measurable, we have

$$\mathbb{E}[C_t] = \mathbb{E}[\mathbb{E}_t[2\|\hat{\mathbf{V}}_{t+\frac{1}{2}}\|_{\boldsymbol{\eta}_t}^2]] \leq \mathbb{E}\left[2(1+\sigma_M^2)\|\mathbf{V}(\mathbf{X}_{t+\frac{1}{2}})\|_{\boldsymbol{\eta}_t}^2 + 2\|\boldsymbol{\eta}_t\|_\infty N\sigma_A^2\right]. \tag{35}$$

Putting together (28), (33), (34), and (35), we get

$$\sum_{t=2}^T \mathbb{E}[\|\mathbf{V}(\mathbf{X}_{t+\frac{1}{2}})\|_{\boldsymbol{\gamma}_t - 2(1+\sigma_M^2)\boldsymbol{\eta}_t}^2 + (1 - a_t(1+\sigma_M^2) - b_t\sigma_M^2)\|\mathbf{V}(\mathbf{X}_{t-\frac{1}{2}})\|_{\boldsymbol{\gamma}_t}^2]$$

$$\leq \mathbb{E}\left[\|\mathbf{X}_1 - \mathbf{x}_\star\|_{1/\boldsymbol{\eta}_{T+1}}^2 + 2(1+\sigma_M^2)\|\mathbf{V}(\mathbf{X}_{3/2})\|_{\boldsymbol{\eta}_1}^2 - \sum_{t=1}^T 21\|\boldsymbol{\gamma}_1\|_\infty NL^2\|\mathbf{X}_t - \mathbf{X}_{t+1}\|^2\right.$$

$$\left. + \sum_{t=1}^T (a_t + b_t + 2\|\boldsymbol{\eta}_t\|_\infty)N\sigma_A^2\right],$$

where $a_t = 6\|\boldsymbol{\gamma}_t\|_\infty^3 NL^2$ and $b_t = \|\boldsymbol{\gamma}_t\|_\infty^2(4N+1)L$. We conclude by using $\mathbf{X}_{3/2} = \mathbf{X}_1$ and noticing that under our learning rate requirement it is always true that $1 - 6\|\boldsymbol{\gamma}_t\|_\infty^2 NL^2(1+\sigma_M^2) - \|\boldsymbol{\gamma}_t\|_\infty(4N+1)L\sigma_M^2 \geq 0$ and $\boldsymbol{\gamma}_t - 2(1+\sigma_M^2)\boldsymbol{\eta}_t \geq \boldsymbol{\gamma}_t/2$. $\qquad\square$

**Remark 4.** We notice that in the analysis, we can replace the common Lipschitz constant by the ones that are proper to each player (i.e., $V^i$ is $L^i$-Lipschitz continuous) when bounding $B_t$. This is however hot the case for our bound on $A_t$, unless we bound directly $\gamma_t^i (L^i)^2$ by a constant.

Again, from Proposition 6 we obtain immediately the bounds on $\sum_{t=1}^T \mathbb{E}[\|\mathbf{V}(X_{t+\frac{1}{2}})\|^2]$ of non-adaptive OptDA+ as claimed in Section 6.

**Theorem 10.** *Let Assumptions 1–3 hold and all players run* (OptDA+) *with non-increasing learning rate sequences* $(\gamma_t^i)_{t\in\mathbb{N}}$ *and* $(\eta_t^i)_{t\in\mathbb{N}}$ *satisfying* (5). *We have*

(a) *If there exists* $q \in [0, 1/4]$ *such that* $\gamma_t^j = \mathcal{O}(1/t^{\frac{1}{4}})$, $\gamma_t^j = \Omega(1/t^{\frac{1}{2}-q})$, *and* $\eta_t^j = \Theta(1/\sqrt{t})$ *for all* $j \in \mathcal{N}$, *then*

$$\sum_{t=1}^T \mathbb{E}[\|\mathbf{V}(X_{t+\frac{1}{2}})\|^2] = \mathcal{O}\left(T^{1-q}\right)$$

(b) *If the noise is multiplicative (i.e.,* $\sigma_A = 0$*) and the learning rates are constant* $\gamma_t \equiv \gamma$, $\eta_t \equiv \eta$, *then*

$$\sum_{t=1}^T \mathbb{E}[\|\mathbf{V}(\mathbf{X}_{t+\frac{1}{2}})\|^2] \le \frac{2}{\min_{i\in\mathcal{N}} \gamma^i} \left( \mathrm{dist}_{1/\boldsymbol{\eta}}(\mathbf{X}_1, \mathcal{X}_\star)^2 + \|\mathbf{V}(\mathbf{X}_1)\|_{\gamma_t}^2 \right)$$

*In particular, if the equalities hold in* (5), *then the above is in* $\mathcal{O}(N^3 L^2 (1 + \sigma_M^2)^3)$.

*Proof.* Let us define $a_t = 6\|\boldsymbol{\gamma}_t\|_\infty^3 N L^2 + \|\boldsymbol{\gamma}_t\|_\infty^2 (4N+1) L + 2\|\boldsymbol{\eta}_t\|_\infty$. From Proposition 6 we know that for all $\mathbf{x}_\star \in \mathcal{X}_\star$, it holds

$$\sum_{s=1}^T \mathbb{E}[\|\mathbf{V}(\mathbf{X}_{t+\frac{1}{2}})\|_{\gamma_t/2}^2]] \le \|\mathbf{X}_1 - \mathbf{x}_\star\|_{1/\boldsymbol{\eta}_{T+1}}^2 + \|\mathbf{V}(\mathbf{X}_1)\|_{\gamma_1}^2 + \sum_{t=1}^T a_t N \sigma_A^2,$$

Since the learning rates are decreasing, we can lower bound $\boldsymbol{\gamma}_t$ by $\boldsymbol{\gamma}_t \ge \boldsymbol{\gamma}_T \ge \min_{i\in\mathcal{N}} \gamma_T^i$. Accordingly,

$$\sum_{s=1}^T \mathbb{E}[\|\mathbf{V}(\mathbf{X}_{t+\frac{1}{2}})\|^2]] \le \frac{2}{\min_{i\in\mathcal{N}} \gamma_T^i} \left( \|\mathbf{X}_1 - \mathbf{x}_\star\|_{1/\boldsymbol{\eta}_{T+1}}^2 + \|\mathbf{V}(\mathbf{X}_1)\|_{\gamma_1}^2 + \sum_{t=1}^T a_t N \sigma_A^2 \right), \quad (36)$$

The result then follows immediately from our learning rate choices. For (a), we observe that with $\|\boldsymbol{\gamma}_t\|_\infty = \mathcal{O}(1/t^{\frac{1}{4}})$ and $\|\boldsymbol{\eta}_t\|_\infty = \mathcal{O}(1/\sqrt{t})$, we have $\sum_{t=1}^T a_t = \mathcal{O}(\sqrt{T})$, while $\gamma_t^j = \Omega(1/t^{\frac{1}{2}-q})$, and $\eta_t^j = \Omega(1/\sqrt{t})$ guarantees $1/\min_{i\in\mathcal{N}} \gamma_T^i = \mathcal{O}(T^{\frac{1}{2}-q})$ and $1/\min_{i\in\mathcal{N}} \eta_{t+1}^T = \mathcal{O}(\sqrt{T})$. For (b), we take $\mathbf{x}_\star = \arg\min_{\mathbf{x}\in\mathcal{X}_\star} \|\mathbf{X}_1 - \mathbf{x}\|_{1/\boldsymbol{\eta}}$. $\qquad\square$

**Bounding Linearized Regret.** To bound the linearized regret, we refine Lemma 10 as follows.

**Lemma 12** (Bound on linearized regret). *Let Assumptions 1–3 hold and all players run* (OptDA+) *with learning rates described in Theorem 2. Then, for all* $i \in \mathcal{N}$, $T \in \mathbb{N}$, *and* $p^i \in \mathcal{X}^i$, *we have*

$$\mathbb{E}\left[ \sum_{t=1}^T \langle V^i(\mathbf{X}_{t+\frac{1}{2}}), X_{t+\frac{1}{2}}^i - p^i \rangle \right] \le \mathbb{E}\left[ \frac{\|X_1^i - p^i\|^2}{2\eta_{T+1}^i} + \sum_{t=1}^T \frac{5}{8} \|\mathbf{V}(\mathbf{X}_{t+\frac{1}{2}})\|_{\gamma_t}^2 \right.$$

$$+ \sum_{t=1}^{T-1} \frac{3\|\gamma_1\|_\infty L^2}{2} \|\mathbf{X}_t - \mathbf{X}_{t+1}\|^2$$

$$\left. + \frac{1}{2} \sum_{t=1}^T \left( 6\|\gamma_t\|_\infty^3 N L^2 + \|\gamma_t\|_\infty^2 (4N+1) L + \eta_t^i \right) \sigma_A^2 \right].$$

*Proof.* Thanks to Lemma 10 and Assumption 3, we can bound

$$\left[\sum_{t=1}^{T}\langle V^i(\mathbf{X}_{t+\frac{1}{2}}), X^i_{t+\frac{1}{2}} - p^i\rangle\right]$$

$$\leq \mathbb{E}\left[\frac{\|X^i_1 - p^i\|^2}{2\eta^i_{T+1}} + \sum_{t=2}^{T}\gamma^i_t L^2\left(3(1+\sigma_M^2)\|\mathbf{V}(\mathbf{X}_{t-\frac{1}{2}})\|^2_{\gamma_t^2} + 3\|\gamma_t^2\|_1\sigma_A^2 + \frac{3}{2}\|\mathbf{X}_t - \mathbf{X}_{t-1}\|^2\right)\right.$$

$$+ \frac{1}{2}\sum_{t=2}^{T}\left((\gamma^i_t)^2 L(\sigma_M^2\|V^i(\mathbf{X}_{t-\frac{1}{2}})\|^2 + \sigma_A^2) + 4L(\sigma_M^2\|\mathbf{V}(\mathbf{X}_{t-\frac{1}{2}})\|^2_{\gamma_t^2} + \|\gamma_t^2\|_1\sigma_A^2)\right)$$

$$+ \left. \frac{1}{2}\sum_{t=1}^{T}\eta^i_t\left((1+\sigma_M^2)\|V^i(\mathbf{X}_{t+\frac{1}{2}})\|^2 + \sigma_A^2\right)\right].$$

In the following, we further bound the above inequality using *i)* $\eta^i_t \leq \gamma^i_t/(4(1+\sigma_M^2))$, *ii)* $\gamma_{t+1} \leq \gamma_t$, *iii)* $\alpha^i_t\|V^i(\mathbf{x})\|^2 \leq \|\mathbf{V}(\mathbf{x})\|^2_{\alpha}$ for any $\alpha \in \mathbb{R}^N_+$ and $\mathbf{x} \in \mathcal{X}$, and *iv)* $\|\alpha\|_\infty = \max_{i\in\mathcal{N}}\alpha^i$ and in particular $\|\alpha^2\|_1 \leq N\|\alpha\|^2_\infty$ for $\alpha \in \mathbb{R}^N_+$.

$$\left[\sum_{t=1}^{T}\langle V^i(\mathbf{X}_{t+\frac{1}{2}}), X^i_{t+\frac{1}{2}} - p^i\rangle\right]$$

$$\leq \mathbb{E}\left[\frac{\|X^i_1 - p^i\|^2}{2\eta^i_{T+1}} + \sum_{t=2}^{T}3\|\gamma_t\|^2_\infty L^2\left((1+\sigma_M^2)\|\mathbf{V}(\mathbf{X}_{t-\frac{1}{2}})\|^2_{\gamma_t} + \|\gamma_t\|_\infty N\sigma_A^2\right)\right.$$

$$+ \frac{1}{2}\sum_{t=2}^{T}\left(\|\gamma_t\|_\infty L\sigma_M^2(\gamma^i_t\|V^i(\mathbf{X}_{t-\frac{1}{2}})\|^2 + 4\|\mathbf{V}(\mathbf{X}_{t-\frac{1}{2}})\|^2_{\gamma_t}) + \|\gamma_t\|^2_\infty(4N+1)L\sigma_A^2\right)$$

$$+ \left. \sum_{t=2}^{T}\frac{3\|\gamma_t\|_\infty L^2}{2}\|\mathbf{X}_t - \mathbf{X}_{t-1}\|^2 + \frac{1}{2}\sum_{t=1}^{T}\left(\frac{\gamma^i_t}{4}\|V^i(\mathbf{X}_{t+\frac{1}{2}})\|^2 + \eta^i_t\sigma_A^2\right)\right]$$

$$\leq \mathbb{E}\left[\frac{\|X^i_1 - p^i\|^2}{2\eta^i_{T+1}} + \sum_{t=1}^{T}\left(3\|\gamma_t\|^2_\infty L^2(1+\sigma_M^2) + \frac{5\|\gamma_t\|_\infty L\sigma_M^2}{2} + \frac{1}{8}\right)\|\mathbf{V}(\mathbf{X}_{t+\frac{1}{2}})\|^2_{\gamma_t}\right.$$

$$+ \sum_{t=1}^{T-1}\frac{3\|\gamma_1\|_\infty L^2}{2}\|\mathbf{X}_t - \mathbf{X}_{t+1}\|^2$$

$$+ \left. \frac{1}{2}\sum_{t=1}^{T}\left(6\|\gamma_t\|^3_\infty NL^2\sigma_A^2 + \|\gamma_t\|^2_\infty(4N+1)L\sigma_A^2 + \eta^i_t\sigma_A^2\right)\right].$$

To conclude, we notice that under that our learning rate requirements it holds that $3\|\gamma_t\|^2_\infty L^2(1+\sigma_M^2) + 5\|\gamma_t\|_\infty L\sigma_M^2/2 \leq 1/2$. $\qquad\square$

Our main regret guarantees of non-adaptive OptDA+ follows from the combination of Lemma 12 and Proposition 6.

**Theorem 11.** *Let Assumptions 1–3 hold and all players run* (OptDA+) *with non-increasing learning rate sequences* $(\gamma^i_t)_{t\in\mathbb{N}}$ *and* $(\eta^i_t)_{t\in\mathbb{N}}$ *satisfying* (5)*. For any* $i \in \mathcal{N}$ *and bounded set* $\mathcal{K}^i \subset \mathcal{X}^i$ *with* $R \geq \sup_{p^i}\|X^i_1 - p^i\|$, *we have:*

*(a) If* $\gamma^j_t = \mathcal{O}(1/t^{\frac{1}{4}})$ *and* $\eta^j_t = \Theta(1/\sqrt{t})$ *for all* $j \in \mathcal{N}$, *then*

$$\max_{p^i\in\mathcal{K}^i}\mathbb{E}\left[\sum_{t=1}^{T}\langle V^i(\mathbf{X}_{t+\frac{1}{2}}), X^i_{t+\frac{1}{2}} - p^i\rangle\right] = \mathcal{O}\left(\sqrt{T}\right).$$

*(b) If the noise is multiplicative (i.e.,* $\sigma_A = 0$*) and the learning rates are constant* $\gamma_t \equiv \gamma$, $\eta_t \equiv \eta$, *then*

$$\max_{p^i\in\mathcal{K}^i}\mathbb{E}\left[\sum_{t=1}^{T}\langle V^i(\mathbf{X}_{t+\frac{1}{2}}), X^i_{t+\frac{1}{2}} - p^i\rangle\right] \leq \frac{R^2}{2\eta^i} + \frac{5}{4}\left(\mathrm{dist}_{1/\eta}(\mathbf{X}_1, \mathcal{X}_\star)^2 + \|\mathbf{V}(\mathbf{X}_1)\|^2_\gamma\right).$$

*In particular, if the equalities hold in (5), the above is in $\mathcal{O}(N^2L(1+\sigma_M^2)^2)$.*

*Proof.* Let $p^i \in \mathcal{K}^i$ and $\mathbf{x}_\star = \arg\min_{\mathbf{x}\in\mathcal{X}_\star}\|\mathbf{X}_1 - \mathbf{x}\|_{1/\boldsymbol{\eta}}$. We define $a_t = 6\|\boldsymbol{\gamma}_t\|_\infty^3 NL^2 + \|\boldsymbol{\gamma}_t\|_\infty^2(4N+1)L+2\|\boldsymbol{\eta}_t\|_\infty$ as in the proof of Theorem 10. Combining Proposition 6 and Lemma 12, we know that

$$\mathbb{E}\left[\sum_{t=1}^{T}\langle V^i(\mathbf{X}_{t+\frac{1}{2}}), X_{t+\frac{1}{2}}^i - p^i\rangle\right]$$
$$\leq \mathbb{E}\left[\frac{R^2}{2\eta_{T+1}^i} + \frac{1}{2}\sum_{t=1}^{T}a_t\sigma_A^2 + \frac{5}{4}\left(\|\mathbf{X}_1-\mathbf{x}_\star\|_{1/\boldsymbol{\eta}_{T+1}}^2 + \|\mathbf{V}(\mathbf{X}_1)\|_{\boldsymbol{\gamma}_1}^2 + \sum_{t=1}^{T}a_t N\sigma_A^2\right)\right].$$

The claims of the theorem follow immediately. $\qquad\square$

To close this section, we bound the regret of non-adaptive OptDA+ when played against arbitrary opponents.

**Proposition 7.** *Let Assumption 3 hold and player $i$ run (OptDA+) with non-increasing learning rates $\gamma_t^i = \Theta(1/t^{\frac{1}{2}-q})$ and $\eta_t^i = \Theta(1/\sqrt{t})$ for some $q \in [0, 1/4]$. Then, if there exists $G \in \mathbb{R}_+$ such that $\sup_{x^i\in\mathcal{X}^i}\|V^i(x^i)\| \leq G$, it holds for any bounded set $\mathcal{K}^i$ with $R \geq \sup_{p^i\in\mathcal{K}^i}\|X_1^i - p^i\|$ that*

$$\max_{p^i\in\mathcal{K}^i}\mathbb{E}\left[\sum_{t=1}^{T}\langle V^i(\mathbf{X}_{t+\frac{1}{2}}), X_{t+\frac{1}{2}}^i - p^i\rangle\right] = \mathcal{O}\left(R^2\sqrt{T} + ((1+\sigma_M^2)G^2 + \sigma_A^2)T^{\frac{1}{2}+q}\right).$$

*Proof.* Let $p^i \in \mathcal{K}^i$. From Corollary 2 and Young's inequality we get

$$\langle \hat{V}_{t+\frac{1}{2}}^i, X_{t+\frac{1}{2}}^i - p^i\rangle \leq \frac{\|X_t^i - p^i\|^2}{2\eta_t^i} - \frac{\|X_{t+1}^i - p^i\|^2}{2\eta_{t+1}^i} - \frac{\|X_t^i - X_{t+1}^i\|^2}{2\eta_t^i}$$
$$+ \left(\frac{1}{2\eta_{t+1}^i} - \frac{1}{2\eta_t^i}\right)\|X_1^i - p^i\|^2 - \gamma_t^i\langle\hat{V}_{t+\frac{1}{2}}^i, \hat{V}_{t-\frac{1}{2}}^i\rangle + \eta_t^i\|\hat{V}_{t+\frac{1}{2}}^i\|^2$$
$$\leq \frac{R^2}{2\eta_t^i} - \frac{\|X_{t+1}^i - p^i\|^2}{2\eta_{t+1}^i} - \frac{\|X_t^i - X_{t+1}^i\|^2}{2\eta_t^i}$$
$$+ \left(\frac{1}{2\eta_{t+1}^i} - \frac{1}{2\eta_t^i}\right)\|X_1^i - p^i\|^2 + \frac{\gamma_t^i}{2}(\|\hat{V}_{t+\frac{1}{2}}^i\|^2 + \|\hat{V}_{t-\frac{1}{2}}^i\|^2) + \eta_t^i\|\hat{V}_{t+\frac{1}{2}}^i\|^2$$

As $\mathbf{V}_{1/2}^i = 0$ and $\eta_1^i = \eta_2^i$, summing the above from $t=1$ to $T$ gives

$$\sum_{t=1}^{T}\langle \hat{V}_{t+\frac{1}{2}}^i, X_{t+\frac{1}{2}}^i - p^i\rangle \leq \frac{\|X_1^i - p^i\|^2}{2\eta_{T+1}^i} - \sum_{t=1}^{T}\frac{\|X_t^i - X_{t+1}^i\|^2}{2\eta_t^i} + \sum_{t=1}^{T}(\gamma_t^i + \eta_t^i)\|\hat{V}_{t+\frac{1}{2}}^i\|^2. \qquad (37)$$

Dropping the negative term and taking expectation leads to

$$\mathbb{E}\left[\sum_{t=1}^{T}\langle V^i(\mathbf{X}_{t+\frac{1}{2}}), X_{t+\frac{1}{2}}^i - p^i\rangle\right] \leq \mathbb{E}\left[\frac{R^2}{2\eta_{T+1}^i} + \sum_{t=1}^{T}(\gamma_t^i + \eta_t^i)((1+\sigma_M^2)\|V^i(\mathbf{X}_{t+\frac{1}{2}})\|^2 + \sigma_A^2)\right]$$
$$\leq \frac{R^2}{2\eta_{T+1}^i} + \sum_{t=1}^{T}(\gamma_t^i + \eta_t^i)((1+\sigma_M^2)G^2 + \sigma_A^2)$$

The claim then follows immediately from the choice of the learning rates. $\qquad\square$

# G   Regret Analysis with Adaptive Learning Rates

In this section, we tackle the regret analysis of OptDA+ run with adaptive learning rates. For ease of notation, we introduce the following quantities[5]

$$\lambda_t^i = \sum_{s=1}^{t}\|\hat{V}_{s+\frac{1}{2}}^i\|^2, \quad \mu_t^i = \sum_{s=1}^{t}\|X_s^i - X_{s+1}^i\|^2.$$

---

[5]For $t \leq 0$, $\lambda_t^i = \mu_t^i = 0$.

Clearly, our adaptive learning rates (Adapt) correspond to $\gamma_t^i = 1/(1 + \lambda_{t-2}^i)^{\frac{1}{2}-q}$ and $\eta_t^i = 1/\sqrt{1 + \lambda_{t-2}^i + \mu_{t-2}^i}$. As Assumption 4 assumes the noise to be bounded *almost surely*, whenever this assumption is used, the stated inequalities only hold almost surely. To avoid repetition, we will not mention this explicitly in the following. Finally, Assumption 5 is obviously satisfied by the learning rates given by (Adapt); therefore, Lemmas 10 and 11 can be effectively applied.

### G.1 Preliminary Lemmas

We start by establishing several basic lemmas that will be used repeatedly in the analysis. We first state the apparent fact that $\lambda_t^i$ grows at most linearly under Assumption 4.

**Lemma 13.** *Let Assumption 4 hold. Then, for all $i \in \mathcal{N}$ and $T \in \mathbb{N}$, we have*

$$\lambda_T^i \leq 2(G^2 + \bar{\sigma}^2)T.$$

*Proof.* Using Assumption 4, we deduce that

$$\|\hat{V}_{t+\frac{1}{2}}^i\|^2 \leq 2\|V^i(\mathbf{X}_{t+\frac{1}{2}})\|^2 + 2\|\xi_{t+\frac{1}{2}}^i\|^2 \leq 2G^2 + 2\bar{\sigma}^2,$$

The claimed inequality is then immediate from the definition of $\lambda_T^i$. $\qquad\square$

The next lemma is a slight generalization of the AdaGrad lemma [6, Lemma 3.5].

**Lemma 14.** *Let $T \in \mathbb{N}$, $\varepsilon > 0$, and $q \in [0,1)$. For any sequence of non-negative real numbers $a_1, \ldots, a_T$, it holds*

$$\sum_{t=1}^{T} \frac{a_t}{\left(\varepsilon + \sum_{s=1}^{t} a_s\right)^q} \leq \frac{1}{1-q}\left(\sum_{t=1}^{T} a_t\right)^{1-q}. \tag{38}$$

*Proof.* The function $y \in \mathbb{R}_+ \mapsto y^{1-q}$ is concave and has derivative $y \mapsto (1-q)/y^q$. Therefore, it holds for every $y, z > 0$ that

$$z^{1-q} \leq y^{1-q} + \frac{1-q}{y^q}(z-y).$$

For $\varepsilon' \in (0, \varepsilon)$, we apply the above inequality to $y = \varepsilon' + \sum_{s=1}^{t} a_s$ and $z = \varepsilon' + \sum_{s=1}^{t-1} a_s$. This gives

$$\frac{1}{1-q}\left(\varepsilon' + \sum_{s=1}^{t-1} a_s\right)^{1-q} \leq \frac{1}{1-q}\left(\varepsilon' + \sum_{s=1}^{t} a_s\right)^{1-q} - \frac{a_t}{\left(\varepsilon' + \sum_{s=1}^{t} a_s\right)^q}$$

$$\leq \frac{1}{1-q}\left(\varepsilon' + \sum_{s=1}^{t} a_s\right)^{1-q} - \frac{a_t}{\left(\varepsilon + \sum_{s=1}^{t} a_s\right)^q}. \tag{39}$$

Moreover, at $t = 1$ we have

$$\frac{a_1}{(\varepsilon + a_1)^q} \leq (\varepsilon' + a_1)^{1-q} \leq \frac{1}{1-q}(\varepsilon' + a_1)^{1-q}. \tag{40}$$

Summing (39) from $t = 2$ to $T$, adding (40), and rearranging leads to

$$\sum_{t=1}^{T} \frac{a_t}{\left(\varepsilon + \sum_{s=1}^{t} a_s\right)^q} \leq \frac{1}{1-q}\left(\epsilon' + \sum_{t=1}^{T} a_t\right)^{1-q}.$$

Provided that the above inequality holds for any $\varepsilon' \in (0, \varepsilon)$, we obtain (38) by taking $\varepsilon' \to 0$. $\quad\square$

The above two lemmas together provide us with the following bound on the sum of the weighted squared norms of feedback.

**Lemma 15.** *Let Assumption 4 hold, $s \in \mathbb{N}_0$, and $r \in [0,1)$. Then, for all $i \in \mathcal{N}$ and $T \in \mathbb{N}$, we have*

$$\sum_{t=1}^{T} \frac{\|\hat{V}_{t+\frac{1}{2}}^i\|^2}{(1 + \lambda_{t-s}^i)^r} \leq \frac{(\lambda_T^i)^{1-r}}{1-r} + 2s(G^2 + \bar{\sigma}^2).$$

*Proof.* Since $1/(1 + \lambda_t^i)^r \leq 1/(1 + \lambda_{t-s}^i)^r$ and $\|\hat{V}_{t+\frac{1}{2}}^i\|^2 \leq 2G^2 + 2\bar{\sigma}^2$, we have

$$\left(\frac{1}{(1 + \lambda_{t-s}^i)^r} - \frac{1}{(1 + \lambda_t^i)^r}\right)\|\hat{V}_{t+\frac{1}{2}}^i\|^2 \leq \left(\frac{1}{(1 + \lambda_{t-s}^i)^r} - \frac{1}{(1 + \lambda_t^i)^r}\right)2(G^2 + \bar{\sigma}^2).$$

Subsequently, it follows from Lemma 14 that

$$\begin{aligned}
\sum_{t=1}^{T} \frac{\|\hat{V}_{t+\frac{1}{2}}^i\|^2}{(1 + \lambda_{t-s}^i)^r} &= \sum_{t=1}^{T} \left(\frac{\|\hat{V}_{t+\frac{1}{2}}^i\|^2}{(1 + \lambda_t^i)^r} + \left(\frac{1}{(1 + \lambda_{t-s}^i)^r} - \frac{1}{(1 + \lambda_t^i)^r}\right)\|\hat{V}_{t+\frac{1}{2}}^i\|^2\right) \\
&\leq \sum_{t=1}^{T} \frac{\|\hat{V}_{t+\frac{1}{2}}^i\|^2}{(1 + \lambda_t^i)^r} + \sum_{t=1}^{T} \left(\frac{1}{(1 + \lambda_{t-s}^i)^r} - \frac{1}{(\lambda_t^i)^r}\right)2(G^2 + \bar{\sigma}^2) \\
&\leq \frac{(\lambda_T^i)^{1-r}}{1 - r} + \sum_{t=-s+1}^{0} \frac{2(G^2 + \bar{\sigma}^2)}{(1 + \lambda_t^i)^r} \\
&= \frac{(\lambda_T^i)^{1-r}}{1 - r} + 2s(G^2 + \bar{\sigma}^2). \qquad \square
\end{aligned}$$

We also state a variant of the above result that takes into account the feedback of all players.

**Lemma 16.** *Let Assumption 4 hold, $s \in \mathbb{N}_0$, $r \in [0, 1)$, and $(\boldsymbol{\alpha}_t)_{t \in \mathbb{N}}$ be a sequence of non-negative $N$-dimensional vectors such that $\alpha_t^i \leq 1/(1 + \lambda_{t-s}^i)^r$. Then, for all $T \in \mathbb{N}$, we have*

$$\sum_{t=1}^{T} \|\hat{\mathbf{V}}_{t+\frac{1}{2}}\|_{\boldsymbol{\alpha}_t}^2 \leq 2Ns(G^2 + \bar{\sigma}^2) + \sum_{i=1}^{N} \frac{(\lambda_T^i)^{1-r}}{1 - r}.$$

*Proof.* This is immediate from Lemma 15. $\qquad \square$

Both Lemma 15 and Lemma 16 are essential for our analysis as they allow us to express the sums appearing in our analysis as a power of $\lambda_t^i$ plus a constant. We end up with a technical lemma for bounding the inverse of $\eta_t^i$.

**Lemma 17.** *Let the learning rates be defined as in (Adapt). For any $i \in \mathcal{N}$, $T \in \mathbb{N}$, and $a, b \in \mathbb{R}_+$, we have*

$$\frac{a}{\eta_{T+1}^i} - b\sum_{t=1}^{T} \frac{\|X_t^i - X_{t+1}^i\|^2}{\eta_t^i} \leq a\sqrt{1 + \lambda_{T-1}^i} + \frac{a^2}{4b}.$$

*Proof.* On one hand, we have

$$\frac{a}{\eta_{T+1}^i} = a\sqrt{1 + \lambda_{T-1}^i + \mu_{T-1}^i} \leq a\sqrt{1 + \lambda_{T-1}^i} + a\sqrt{\mu_{T-1}^i}.$$

On the other hand, with $\eta_t^i \leq 1$, it holds

$$b\sum_{t=1}^{T} \frac{\|X_t^i - X_{t+1}^i\|^2}{\eta_t^i} \geq b\sum_{t=1}^{T} \|X_t^i - X_{t+1}^i\|^2 \geq b\mu_{T-1}^i.$$

Let us define the function $f : y \in \mathbb{R} \mapsto -by^2 + ay$. Then

$$a\sqrt{\mu_{T-1}^i} - b\mu_{T-1}^i \leq \max_{y \in \mathbb{R}} f(y) = \frac{a^2}{4b}.$$

Combining the above inequalities gives the desired result. $\qquad \square$

## G.2 Robustness Against Adversarial Opponents

In this part, we derive regret bounds for adaptive OptDA+ when played against adversarial opponents.

**Proposition 8.** *Let Assumption 4 hold and player $i$ run (OptDA+) with learning rates (Adapt). Then, for any bounded set $\mathcal{K}^i$ with $R \geq \sup_{p^i \in \mathcal{K}^i} \|X_1^i - p^i\|$, it holds*

$$\max_{p^i \in \mathcal{K}^i} \mathbb{E}\left[\sum_{t=1}^T \langle V^i(\mathbf{X}_{t+\frac{1}{2}}), X_{t+\frac{1}{2}}^i - p^i \rangle\right]$$
$$= \mathcal{O}\left(((G^2 + \bar{\sigma}^2)T)^{\frac{1}{2}+q} + R^2(G + \bar{\sigma})\sqrt{T} + R^4 + G^2 + \bar{\sigma}^2\right).$$

*Proof.* To begin, we notice that inequality (37) that we established in the proof of Proposition 7 still holds here for any $p^i \in \mathcal{K}^i$. Furthermore, applying Lemma 17 with $a \leftarrow R^2/2$, $b \leftarrow 1/2$ leads to

$$\frac{R^2}{2\eta_{T+1}^i} - \sum_{t=1}^T \frac{\|X_t^i - X_{t+1}^i\|^2}{2\eta_t^i} \leq \frac{R^2\sqrt{1 + \lambda_{T-1}^i}}{2} + \frac{R^4}{8}.$$

On the other hand, invoking Lemma 15 with either $r \leftarrow 1/4 + q$ or $r \leftarrow 1/2$ guarantees that

$$\sum_{t=1}^T (\gamma_t^i + \eta_t^i)\|\hat{V}_{t+\frac{1}{2}}^i\|^2 \leq \frac{4(\lambda_T^i)^{3/4-q}}{3 - 4q} + 2\sqrt{\lambda_T^i} + 8(G^2 + \bar{\sigma}^2).$$

Putting the above inequalities together, we obtain

$$\max_{p^i \in \mathcal{K}^i} \mathbb{E}\left[\sum_{t=1}^T \langle V^i(\mathbf{X}_{t+\frac{1}{2}}), X_{t+\frac{1}{2}}^i - p^i \rangle\right] \leq \mathbb{E}\left[\frac{R^2\sqrt{1 + \lambda_{T-1}^i}}{2} + \frac{4(\lambda_T^i)^{3/4-q}}{3 - 4q} + 2\sqrt{\lambda_T^i}\right]$$
$$+ \frac{R^4}{8} + 8(G^2 + \bar{\sigma}^2).$$

We conclude with the help of Lemma 13. $\qquad\square$

## G.3 Smaller Regret Against Opponents with Same Learning Algorithm

We now address the more challenging part of the analysis: fast regret minimization when all players adopt adaptive OptDA+. For this, we need to control the different terms appearing in Lemmas 10 and 11. For the latter we build the following lemma to control the sum of some differences. As argued in Section 5, this is the reason that we include $\|X_s^i - X_{s+1}^i\|^2$ in the definition of $\eta_t^i$.

**Lemma 18.** *Let Assumptions 1 and 4 hold and the learning rates be defined as in (Adapt), then for all $T \in \mathbb{N}$, we have*

$$\sum_{t=1}^T \left(3\|\mathbf{V}(\mathbf{X}_t) - \mathbf{V}(\mathbf{X}_{t+1})\|_{\boldsymbol{\gamma}_t}^2 - \|\mathbf{X}_t - \mathbf{X}_{t+1}\|_{1/(4\boldsymbol{\eta}_t)}^2\right) \leq 432N^3L^6 + 24N^2G^2.$$

*Proof.* For all $i \in \mathcal{N}$, let us define

$$\bar{t}^i := \max\left\{s \in \{0, ..., T\} : \eta_t^i \geq \frac{1}{12NL^2}\right\},$$

where we set $\eta_0^i = 1/(12NL^2)$ to ensure that $\bar{t}^i$ is always well-defined. By the definition of $\eta_t^i$, the inequality $\eta_{\bar{t}^i}^i \geq 1/(12NL^2)$ implies $\mu_{\bar{t}^i-2}^i \leq 144N^2L^2$. We next define the sets

$$\mathcal{T} := \bigcup_{i \in \mathcal{N}} \{\bar{t}^i - 1, \bar{t}^i\} \cap \{1, ..., T\}$$

Clearly, $\mathrm{card}(\mathcal{T}) \leq 2N$. With $\boldsymbol{\gamma}_t \leq 1$, Assumptions 1 and 4, we obtain

$$
\sum_{t=1}^{T} 3\|\mathbf{V}(\mathbf{X}_t) - \mathbf{V}(\mathbf{X}_{t+1})\|_{\boldsymbol{\gamma}_t}^2
$$

$$
\leq \sum_{t=1}^{T} 3\|\mathbf{V}(\mathbf{X}_t) - \mathbf{V}(\mathbf{X}_{t+1})\|^2
$$

$$
= \sum_{t\in[T]\setminus\mathcal{T}} 3\|\mathbf{V}(\mathbf{X}_t) - \mathbf{V}(\mathbf{X}_{t+1})\|^2 + \sum_{t\in\mathcal{T}} 3\|\mathbf{V}(\mathbf{X}_t) - \mathbf{V}(\mathbf{X}_{t+1})\|^2
$$

$$
\leq \sum_{t\in[T]\setminus\mathcal{T}} 3NL^2\|\mathbf{X}_t - \mathbf{X}_{t+1}\|^2 + \sum_{t\in\mathcal{T}} 6\left(\|\mathbf{V}(\mathbf{X}_t)\|^2 + \|\mathbf{V}(\mathbf{X}_{t+1})\|^2\right)
$$

$$
\leq \sum_{i=1}^{N} \sum_{t\in[T]\setminus\mathcal{T}} 3NL^2\|X_t^i - X_{t+1}^i\|^2 + \sum_{t\in\mathcal{T}} 12NG^2
$$

$$
\leq \sum_{i=1}^{N} 3NL^2 \bigg( \underbrace{\sum_{t=1}^{\bar{t}^i-2} \|X_t^i - X_{t+1}^i\|^2}_{\mu_{\bar{t}^i-2}^i \leq 144N^2L^2} + \sum_{t=\bar{t}^i+1}^{T} \|X_t^i - X_{t+1}^i\|^2 \bigg) + 24N^2G^2 \tag{41}
$$

On the other hand, by the choice of $\bar{t}^i$ we know that $1/\eta_t^i \geq 12NL^2$ for all $t \geq \bar{t}^i + 1$; hence

$$
\sum_{t=1}^{T} \|\mathbf{X}_t - \mathbf{X}_{t+1}\|_{1/(4\boldsymbol{\eta}_t)}^2 = \sum_{i=1}^{N} \sum_{t=1}^{T} \frac{\|X_t^i - X_{t+1}^i\|^2}{4\eta_t^i}
$$

$$
\geq \sum_{i=1}^{N} \sum_{t=\bar{t}^i+1}^{T} \frac{\|X_t^i - X_{t+1}^i\|^2}{4\eta_t^i}
$$

$$
\geq \sum_{i=1}^{N} \sum_{t=\bar{t}^i+1}^{T} 3NL^2\|X_t^i - X_{t+1}^i\|^2. \tag{42}
$$

Combining (41) and (42) gives the desired result. $\qquad\square$

With Lemma 18 and the lemmas introduced in Appendix G.1, we are in a position to provide a bound on the expectation of the sum of the weighted squared operators norms plus the second-order path length. The next lemma is a fundamental building block for showing faster rates of adaptive OptDA+.

**Lemma 19** (Bound on sum of squared norms). *Let Assumptions 1, 2 and 4 hold and all players run OptDA+ with adaptive learning rates (Adapt). Then, for all $T \in \mathbb{N}$ we have*

$$
\sum_{t=1}^{T} \mathbb{E}[\|\mathbf{V}(\mathbf{X}_{t+\frac{1}{2}})\|_{\boldsymbol{\gamma}_t}^2] + \frac{1}{8}\sum_{t=1}^{T} \mathbb{E}[\|\mathbf{X}_t - \mathbf{X}_{t+1}\|^2] \leq c_1 \sum_{i=1}^{N} \mathbb{E}\left[\sqrt{\lambda_T^i}\right] + c_2,
$$

*where*

$$
\rho = \min_{\mathbf{x}_\star \in \mathcal{X}_\star} \max_{i\in\mathcal{N}} \|X_1^i - x_\star^i\|,
$$

$$
c_1 = 12NL^2 + 8NL + 2L + \rho^2 + 4,
$$

$$
c_2 = 432N^3L^6 + 24N^2G^2 + (12NL^2 + 8NL + 2L + 8)(NG^2 + N\bar{\sigma}^2) + N\rho^2 + 2N\rho^4.
$$

*Proof.* As in the proof of Proposition 6, and proceed to bound in expectation the sum of the following quantities

$$
A_t = 3\|\mathbf{V}(\mathbf{X}_t) - \mathbf{V}(\mathbf{X}_{t+1})\|_{\boldsymbol{\gamma}_t}^2 - \|\mathbf{X}_t - \mathbf{X}_{t+1}\|_{1/(2\boldsymbol{\eta}_t)}^2,
$$

$$
B_t = 6\|\boldsymbol{\gamma}_t\|_1 L^2\|\hat{\mathbf{V}}_{t-\frac{1}{2}}\|_{\boldsymbol{\gamma}_t^2}^2 + (4N+1)L\|\boldsymbol{\xi}_{t-\frac{1}{2}}\|_{\boldsymbol{\gamma}_t^2}^2, \quad C_t = 2\|\hat{\mathbf{V}}_{t+\frac{1}{2}}\|_{\boldsymbol{\eta}_t}^2. \tag{43}
$$

Thanks to Lemma 18, we know that the sum of $A_t$ can be bounded directly without taking expectation by

$$\sum_{t=1}^{T} A_t = \sum_{t=1}^{T} \left( 3\|\mathbf{V}(\mathbf{X}_t) - \mathbf{V}(\mathbf{X}_{t+1})\|_{\boldsymbol{\gamma}_t}^2 - \|\mathbf{X}_t - \mathbf{X}_{t+1}\|_{1/(4\boldsymbol{\eta}_t)}^2 - \|\mathbf{X}_t - \mathbf{X}_{t+1}\|_{1/(4\boldsymbol{\eta}_t)}^2 \right)$$

$$\leq 432N^3L^6 + 24N^2G^2 - \sum_{t=1}^{T} \|\mathbf{X}_t - \mathbf{X}_{t+1}\|_{1/(4\boldsymbol{\eta}_t)}^2.$$

To obtain the above inequality we have also used $\boldsymbol{\eta}_t \leq 1$. To bound $\mathbb{E}[B_t]$, we use $\mathbb{E}[\|\boldsymbol{\xi}_{t-\frac{1}{2}}\|_{\boldsymbol{\gamma}_t^2}^2] \leq \mathbb{E}[\|\hat{\mathbf{V}}_{t-\frac{1}{2}}\|_{\boldsymbol{\gamma}_t^2}^2]$ shown in (49), $\|\boldsymbol{\gamma}_t\|_1 \leq N$, and Lemma 16 (as $(\gamma_{t+1}^i)^2 \leq 1/\sqrt{1 + \lambda_{t-1}^i}$) to obtain

$$\sum_{t=2}^{T} \mathbb{E}[B_t] \leq \mathbb{E}\left[ \sum_{t=2}^{T} (6NL^2 + (4N+1)L)\|\hat{\mathbf{V}}_{t-\frac{1}{2}}\|_{\boldsymbol{\gamma}_t^2}^2 \right]$$

$$= \mathbb{E}\left[ \sum_{t=1}^{T-1} (6NL^2 + (4N+1)L)\|\hat{\mathbf{V}}_{t+\frac{1}{2}}\|_{(\boldsymbol{\gamma}_{t+1})^2}^2 \right]$$

$$\leq (6NL^2 + (4N+1)L)\left( 2N(G^2 + \bar{\sigma}^2) + \sum_{i=1}^{N} 2\,\mathbb{E}\left[ \sqrt{\lambda_{T-1}^i} \right] \right). \tag{44}$$

Similarly, the sum of $C_t$ can be bounded in expectation by

$$\sum_{t=1}^{T} \mathbb{E}[C_t] \leq 8N(G^2 + \bar{\sigma}^2) + \sum_{i=1}^{N} 4\,\mathbb{E}\left[ \sqrt{\lambda_T^i} \right]. \tag{45}$$

Let us choose $\mathbf{x}_\star = \arg\min_{\mathbf{x} \in \mathcal{X}_\star} \max_{i \in \mathcal{N}} \|X_1^i - x^i\|$ so that $\rho = \max_{i \in \mathcal{N}} \|X_1^i - x_\star^i\|$. Plugging (43), (44), and (45) into (28) of Lemma 11, we get readily

$$\sum_{t=2}^{T} \mathbb{E}[\|\mathbf{V}(\mathbf{X}_{t+\frac{1}{2}})\|_{\boldsymbol{\gamma}_t}^2 + \|\mathbf{V}(\mathbf{X}_{t-\frac{1}{2}})\|_{\boldsymbol{\gamma}_t}^2] + \sum_{t=1}^{T} \mathbb{E}[\|\mathbf{X}_t - \mathbf{X}_{t+1}\|_{1/(8\boldsymbol{\eta}_t)}^2]$$

$$\leq \mathbb{E}[\|\mathbf{X}_1 - \mathbf{x}_\star\|_{1/\boldsymbol{\eta}_{T+1}}^2] - \sum_{t=1}^{T} \|\mathbf{X}_t - \mathbf{X}_{t+1}\|_{1/(8\boldsymbol{\eta}_t)}^2 + (12NL^2 + 8NL + 2L + 4)\sum_{i=1}^{N} \mathbb{E}\left[ \sqrt{\lambda_T^i} \right]$$

$$+ 432N^3L^6 + 24N^2G^2 + (12NL^2 + 8NL + 2L + 8)(NG^2 + N\bar{\sigma}^2) \tag{46}$$

Using Lemma 17, we can then further bound the RHS of (46) with

$$\|\mathbf{X}_1 - \mathbf{x}_\star\|_{1/\boldsymbol{\eta}_{T+1}}^2 - \sum_{t=1}^{T} \|\mathbf{X}_t - \mathbf{X}_{t+1}\|_{1/(8\boldsymbol{\eta}_t)}^2 = \sum_{i=1}^{N} \left( \frac{\|X_1^i - x_\star^i\|^2}{\eta_{T+1}^i} - \sum_{t=1}^{T} \frac{\|X_t^i - X_{t+1}^i\|^2}{8\eta_t^i} \right)$$

$$\leq \sum_{i=1}^{N} \left( \|X_1^i - x_\star^i\|^2 \sqrt{1 + \lambda_{T-1}^i} + 2\|x_\star^i\|^4 \right)$$

$$\leq N\rho^2 + 2N\rho^4 + \sum_{i=1}^{N} \rho^2 \sqrt{\lambda_{T-1}^i}. \tag{47}$$

Finally, using $\boldsymbol{\eta}_t \leq 1$, $\mathbf{X}_{3/2} = \mathbf{X}_1$, and $\boldsymbol{\gamma}_2 = \boldsymbol{\gamma}_1$, the left-hand side (LHS) of (46) can be bounded from below by

$$\sum_{t=2}^{T} \mathbb{E}[\|\mathbf{V}(\mathbf{X}_{t+\frac{1}{2}})\|_{\boldsymbol{\gamma}_t}^2 + \|\mathbf{V}(\mathbf{X}_{t-\frac{1}{2}})\|_{\boldsymbol{\gamma}_t}^2] + \sum_{t=1}^{T} \mathbb{E}[\|\mathbf{X}_t - \mathbf{X}_{t+1}\|_{1/(8\boldsymbol{\eta}_t)}^2]$$

$$\geq \sum_{t=1}^{T} \mathbb{E}[\|\mathbf{V}(\mathbf{X}_{t+\frac{1}{2}})\|_{\boldsymbol{\gamma}_t}^2] + \frac{1}{8} \sum_{t=1}^{T} \mathbb{E}[\|\mathbf{X}_t - \mathbf{X}_{t+1}\|^2]. \tag{48}$$

Combining (46), (47), and (48) gives the desired result. $\qquad\square$

We also refine Lemma 10 for the case of adaptive learning rates. The next lemma suggests the terms that need to be bounded in expectation in order to control the regret.

**Lemma 20** (Bound on linearized regret). *Let Assumptions 1, 2 and 4 hold and all players run OptDA+ with adaptive learning rates (Adapt). Then, for all $i \in \mathcal{N}$, $T \in \mathbb{N}$, and bounded set $\mathcal{K}^i \subset \mathcal{X}^i$ with $R \geq \sup_{p^i \in \mathcal{K}^i} \|X_1^i - p^i\|$, it holds that*

$$
\max_{p^i \in \mathcal{K}^i} \mathbb{E}\left[\sum_{t=1}^{T} \langle V^i(\mathbf{X}_{t+\frac{1}{2}}), X_{t+\frac{1}{2}}^i - p^i \rangle\right] \leq \mathbb{E}\Bigg[ \left(\frac{R^2}{2} + \frac{L+1}{2}\right) \sqrt{\lambda_T^i} + (6L^2 + 4L)\sum_{j=1}^{N} \sqrt{\lambda_{T-1}^j}
$$
$$
+ \frac{R^2 \sqrt{\mu_{T-1}^i}}{2} + \frac{3L^2}{2}\sum_{t=1}^{T-1} \|\mathbf{X}_t - \mathbf{X}_{t+1}\|^2
$$
$$
+ \frac{R^2}{2} + (6NL^2 + 4NL + L + 2)(G^2 + \bar{\sigma}^2)\Bigg].
$$

*Proof.* We will derive inequality (49) from Lemma 10. To begin, by Assumption 3(a) the noises are conditionally unbiased and we can thus write

$$
\mathbb{E}_{t-1}[\|\hat{V}_{t-\frac{1}{2}}^i\|^2] = \|V^i(X_{t-\frac{1}{2}}^i)\|^2 + \mathbb{E}_{t-1}[\|\xi_{t-\frac{1}{2}}^i\|^2] \geq \mathbb{E}_{t-1}[\|\xi_{t-\frac{1}{2}}^i\|^2].
$$

Subsequently, $\gamma_t$ being $\mathcal{F}_{t-1}$-measurable, applying the law of total expectation gives

$$
\mathbb{E}[\|\boldsymbol{\xi}_{t-\frac{1}{2}}\|_{\gamma_t^2}^2] = \mathbb{E}\left[\sum_{i=1}^{N} (\gamma_t^i)^2 \, \mathbb{E}_{t-1}[\|\xi_{t-\frac{1}{2}}^i\|^2]\right]
$$
$$
\leq \mathbb{E}\left[\sum_{i=1}^{N} (\gamma_t^i)^2 \, \mathbb{E}_{t-1}[\|\hat{V}_{t-\frac{1}{2}}^i\|^2]\right] = \mathbb{E}[\|\hat{\mathbf{V}}_{t-\frac{1}{2}}\|_{\gamma_t^2}^2].
$$

(49)

Plugging the above two inequalities into the inequality of Lemma 10 and using $\gamma_t^i \leq 1$ results in

$$
\max_{p^i \in \mathcal{K}^i} \mathbb{E}\left[\sum_{t=1}^{T} \langle V^i(\mathbf{X}_{t+\frac{1}{2}}), X_{t+\frac{1}{2}}^i - p^i \rangle\right]
$$
$$
\leq \mathbb{E}\Bigg[ \frac{R^2}{2\eta_{T+1}^i} + \sum_{t=2}^{T} \gamma_t^i L^2 \left(3\|\hat{\mathbf{V}}_{t-\frac{1}{2}}\|_{\gamma_t^2}^2 + \frac{3}{2}\|\mathbf{X}_t - \mathbf{X}_{t-1}\|^2\right)
$$
$$
+ \frac{1}{2}\sum_{t=2}^{T}((\gamma_t^i)^2 L\|\hat{V}_{t-\frac{1}{2}}^i\|^2 + 4L\|\hat{\mathbf{V}}_{t-\frac{1}{2}}\|_{\gamma_t^2}^2) + \frac{1}{2}\sum_{t=1}^{T} \eta_t^i \|\hat{V}_{t+\frac{1}{2}}^i\|^2 \Bigg]
$$
$$
\leq \mathbb{E}\Bigg[ \frac{R^2 \sqrt{1 + \lambda_{T-1}^i + \mu_{T-1}^i}}{2} + \sum_{t=1}^{T-1}(3L^2 + 2L)\|\hat{\mathbf{V}}_{t+\frac{1}{2}}\|_{(\gamma_{t+1})^2}^2 + \frac{3L^2}{2}\sum_{t=1}^{T-1}\|\mathbf{X}_t - \mathbf{X}_{t+1}\|^2
$$
$$
+ \frac{1}{2}\sum_{t=1}^{T-1}(\gamma_{t+1}^i)^2 L\|\hat{V}_{t+\frac{1}{2}}^i\|^2 + \frac{1}{2}\sum_{t=1}^{T} \eta_t^i \|\hat{V}_{t+\frac{1}{2}}^i\|^2 \Bigg].
$$

Since we have both $(\gamma_{t+1}^i)^2 \leq 1/\sqrt{1 + \lambda_{t-1}^i}$ and $\eta_t^i \leq 1/\sqrt{1 + \lambda_{t-2}^i}$, applying Lemma 15 leads to

$$
\sum_{t=1}^{T-1}(\gamma_{t+1}^i)^2 L\|\hat{V}_{t+\frac{1}{2}}^i\|^2 + \sum_{t=1}^{T} \eta_t^i \|\hat{V}_{t+\frac{1}{2}}^i\|^2 \leq L\left(\sqrt{\lambda_{T-1}^i} + 2(G^2 + \bar{\sigma}^2)\right) + \sqrt{\lambda_T^i} + 4(G^2 + \bar{\sigma}^2).
$$

Similarly, using Lemma 16 we deduce

$$
\sum_{t=1}^{T-1}(3L^2 + 2L)\|\hat{\mathbf{V}}_{t+\frac{1}{2}}\|_{(\gamma_{t+1})^2}^2 \leq (3L^2 + 2L)\left(2N(G^2 + \bar{\sigma}^2) + \sum_{j=1}^{N} 2\sqrt{\lambda_{T-1}^j}\right)
$$

Putting the above inequalities together and using $\sqrt{1 + \lambda_{T-1}^i + \mu_{T-1}^i} \leq 1 + \sqrt{\lambda_{T-1}^i} + \sqrt{\mu_{T-1}^i}$ gives the desired result. $\qquad\square$

### G.3.1 The Case of Additive Noise

From Lemma 19 and Lemma 20 we can readily derive our main results for the case of additive noise.

**Theorem 12.** *Let Assumptions 1, 2 and 4 hold and all players run OptDA+ with adaptive learning rates* (Adapt). *Then,*

$$\sum_{t=1}^{T} \mathbb{E}[\|\mathbf{V}(\mathbf{X}_{t+\frac{1}{2}})\|]^2 = \mathcal{O}\left(T^{1-q}\right).$$

*Proof.* With Lemma 13, for $t \in \{1, ..., T\}$, we can lower bound the learning rate $\gamma_t^i$ by

$$\gamma_t^i = \frac{1}{(1 + \lambda_{t-2}^i)^{\frac{1}{2}-q}} \geq \frac{1}{(1 + 2\max(t-2, 0)(G^2 + \bar{\sigma}^2))^{\frac{1}{2}-q}} \geq \frac{1}{(1 + 2T(G^2 + \bar{\sigma}^2))^{\frac{1}{2}-q}}.$$

Lemma 19 thus guarantees

$$\frac{\sum_{t=1}^{T} \mathbb{E}[\|\mathbf{V}(\mathbf{X}_{t+\frac{1}{2}})\|^2]}{(1 + 2T(G^2 + \bar{\sigma}^2))^{\frac{1}{2}-q}} \leq c_1 \sum_{i=1}^{N} \mathbb{E}\left[\sqrt{\lambda_t^i}\right] + c_2.$$

We conclude by using again Lemma 13. $\qquad\square$

**Theorem 13.** *Let Assumptions 1, 2 and 4 hold and all players run OptDA+ with adaptive learning rates* (Adapt). *Then, for any $i \in \mathcal{N}$ and bounded set $\mathcal{K}^i \subset \mathcal{X}^i$, we have*

$$\max_{p^i \in \mathcal{K}^i} \mathbb{E}\left[\sum_{t=1}^{T} \langle V^i(\mathbf{X}_{t+\frac{1}{2}}), X_{t+\frac{1}{2}}^i - p^i \rangle\right] = \mathcal{O}\left(\sqrt{T}\right).$$

*Proof.* This follows from Lemma 20. To begin, with Lemma 13, we have clearly

$$\mathbb{E}\left[\left(\frac{R^2}{2} + \frac{L+1}{2}\right)\sqrt{\lambda_t^i} + (6L^2 + 4L)\sum_{j=1}^{N}\sqrt{\lambda_{T-1}^j}\right] = \mathcal{O}\left(\sqrt{T}\right).$$

Next, thanks to Lemma 19 we can bound

$$\mathbb{E}\left[\frac{R^2\sqrt{\mu_{T-1}^i}}{2} + \frac{3L^2}{2}\sum_{t=1}^{T-1}\|\mathbf{X}_t - \mathbf{X}_{t+1}\|^2\right] \leq \mathbb{E}\left[\left(\frac{R^2}{2} + \frac{3L^2}{2}\right)\sum_{t=1}^{T-1}\|\mathbf{X}_t - \mathbf{X}_{t+1}\|^2\right]$$

$$\leq (4R^2 + 12L^2)\left(c_1\sum_{i=1}^{N}\mathbb{E}\left[\sqrt{\lambda_t^i}\right] + c_2\right).$$

This is again in $\mathcal{O}(\sqrt{T})$. Plugging the above into Lemma 20 concludes the proof. $\qquad\square$

### G.3.2 The Case of Multiplicative Noise

The case of multiplicative noise is more delicate. As explained in Section 5, the main step is to establish an inequality in the form of (6). This is achieved in Lemma 22 by using Lemma 19. Before that, we derive a lemma to show how inequality (6) implies boundedness of the relevant quantities.

**Lemma 21.** *Let $p, r, c \in \mathbb{R}_+$ such that $p > r$, $c \in \mathbb{R}_+$, and $(a^1, \ldots, a^N)$ be a collection of $N$ non-negative real-valued random variables. If*

$$\sum_{i=1}^{N} \mathbb{E}[(a^i)^p] \leq c\sum_{i=1}^{N} \mathbb{E}[(a^i)^r], \tag{50}$$

*Then $\sum_{i=1}^{N} \mathbb{E}[(a^i)^p] \leq Nc^{\frac{p}{p-r}}$ and $\sum_{i=1}^{N} \mathbb{E}[(a^i)^r] \leq Nc^{\frac{r}{p-r}}$ .*

*Proof.* Since $p > r$, the function $y \in \mathbb{R}_+ \cup \{0\} \mapsto y^{\frac{r}{p}}$ is concave. Applying Jensen's inequality for the expectation gives $\mathbb{E}[(a^i)^r] \leq \mathbb{E}[(a^i)^p]^{\frac{r}{p}}$. Next, we apply Jensen's inequality for the average to obtain

$$\frac{1}{N} \sum_{i=1}^N \mathbb{E}[(a^i)^p]^{\frac{r}{p}} \leq \left( \frac{1}{N} \sum_{i=1}^N \mathbb{E}[(a^i)^p] \right)^{\frac{r}{p}}. \tag{51}$$

Along with inequality (50) we then get

$$\sum_{i=1}^N \mathbb{E}[(a^i)^p] \leq c \sum_{i=1}^N \mathbb{E}[(a^i)^r] \leq cN^{1-\frac{r}{p}} \left( \sum_{i=1}^N \mathbb{E}[(a^i)^p] \right)^{\frac{r}{p}}. \tag{52}$$

In other words

$$\left( \sum_{i=1}^N \mathbb{E}[(a^i)^p] \right)^{1-\frac{r}{p}} \leq cN^{1-\frac{r}{p}}.$$

Taking both sides of the inequality to the power of $p/(p-r)$, we obtain effectively

$$\sum_{i=1}^N \mathbb{E}[(a^i)^p] \leq Nc^{\frac{p}{p-r}}$$

The second inequality combines the above with second part of (52). $\qquad\square$

In the next lemma we build inequality (6), and combined with Lemma 21 we obtain the boundedness of various quantities. This is also where the factor $1/q$ shows up.

**Lemma 22.** *Let Assumptions 1, 2 and 4 hold and all players run OptDA+ with adaptive learning rates (Adapt). Assume additionally Assumption 3 with $\sigma_A = 0$. Then, for any $T \in \mathbb{N}$, we have*

$$\sum_{i=1}^N \mathbb{E}\left[ (1 + \lambda_T^i)^{\frac{1}{2}+q} \right] \leq N \left( (1 + \sigma_M^2)c_1 + \frac{(1+\sigma_M^2)c_2 + 1}{N} \right)^{1+\frac{1}{2q}}, \tag{53}$$

$$\sum_{i=1}^N \mathbb{E}\left[ \sqrt{1 + \lambda_T^i} \right] \leq N \left( (1 + \sigma_M^2)c_1 + \frac{(1+\sigma_M^2)c_2 + 1}{N} \right)^{\frac{1}{2q}}, \tag{54}$$

$$\sum_{i=1}^N \mathbb{E}[\mu_T^i] \leq 8Nc_1 \left( (1 + \sigma_M^2)c_1 + \frac{(1+\sigma_M^2)c_2 + 1}{N} \right)^{\frac{1}{2q}} + 8c_2. \tag{55}$$

*Proof.* From Lemma 19 we know that

$$\sum_{t=1}^T \mathbb{E}[\|\mathbf{V}(\mathbf{X}_{t+\frac{1}{2}})\|_{\gamma_t}^2] \leq c_1 \sum_{i=1}^N \mathbb{E}\left[ \sqrt{\lambda_T^i} \right] + c_2,$$

Since $\gamma_t$ is $\mathcal{F}_t$-measurable (it is even $\mathcal{F}_{t-1}$-measurable), using the relative noise assumption and the law of total expectation we get

$$\mathbb{E}[\|\mathbf{V}(\mathbf{X}_{t+\frac{1}{2}})\|_{\gamma_t}^2] = \sum_{i=1}^N \mathbb{E}[\gamma_t^i \mathbb{E}_t[\|V^i(\mathbf{X}_{t+\frac{1}{2}})\|^2] \geq \sum_{i=1}^N \mathbb{E}\left[ \gamma_t^i \mathbb{E}_t\left[ \frac{\|\hat{V}_{t+\frac{1}{2}}^i\|^2}{1+\sigma_M^2} \right] \right] = \frac{\|\hat{\mathbf{V}}_{t+\frac{1}{2}}\|_{\gamma_t}^2}{1+\sigma_M^2}.$$

The learning rates $\gamma_t$ being non-increasing, we can then bound from below the sum of $\mathbb{E}[\|\mathbf{V}(\mathbf{X}_{t+\frac{1}{2}})\|^2_{\gamma_t}]$ by

$$
\begin{aligned}
\sum_{t=1}^{T} \mathbb{E}[\|\mathbf{V}(\mathbf{X}_{t+\frac{1}{2}})\|^2_{\gamma_t}] &\geq \frac{1}{1+\sigma_M^2} \sum_{t=1}^{T} \mathbb{E}[\|\hat{\mathbf{V}}_{t+\frac{1}{2}}\|^2_{\gamma_t}] \\
&\geq \frac{1}{1+\sigma_M^2} \sum_{t=1}^{T} \mathbb{E}[\|\hat{\mathbf{V}}_{t+\frac{1}{2}}\|^2_{\gamma_{T+2}}] \\
&= \frac{1}{1+\sigma_M^2} \sum_{i=1}^{N} \mathbb{E}\left[\frac{\sum_{t=1}^{T}\|\hat{V}^i_{t+\frac{1}{2}}\|^2}{(1+\lambda_T^i)^{\frac{1}{2}-q}}\right] \\
&= \frac{1}{1+\sigma_M^2} \sum_{i=1}^{N} \mathbb{E}\left[\frac{\lambda_T^i + 1 - 1}{(1+\lambda_T^i)^{\frac{1}{2}-q}}\right] \\
&\geq -\frac{1}{1+\sigma_M^2} + \frac{1}{1+\sigma_M^2} \sum_{i=1}^{N} \mathbb{E}\left[(1+\lambda_T^i)^{\frac{1}{2}+q}\right].
\end{aligned}
$$

As a consequence, we have shown that

$$
\sum_{i=1}^{N} \mathbb{E}\left[(1+\lambda_T^i)^{\frac{1}{2}+q}\right] \leq (1+\sigma_M^2)c_1 \sum_{i=1}^{N} \mathbb{E}\left[\sqrt{\lambda_T^i}\right] + (1+\sigma_M^2)c_2 + 1,
$$

Subsequently,

$$
\sum_{i=1}^{N} \mathbb{E}\left[(1+\lambda_T^i)^{\frac{1}{2}+q}\right] \leq \left((1+\sigma_M^2)c_1 + \frac{(1+\sigma_M^2)c_2 + 1}{N}\right) \sum_{i=1}^{N} \mathbb{E}\left[\sqrt{1+\lambda_T^i}\right].
$$

We deduce (53) and (54) with the help of Lemma 21 taking $p \leftarrow 3/4 - q$, $r \leftarrow 1/2$, $c \leftarrow (1+\sigma_M^2)c_1 + ((1+\sigma_M^2)c_2 + 1)/N$, and $a^i \leftarrow 1 + \lambda_T^i$. Plugging (54) into Lemma 19 gives (55). $\qquad\square$

Now, as an immediate consequence of all our previous results, we obtain the constant regret bound of adaptive OptDA+ under multiplicative noise.

**Theorem 14.** *Let Assumptions 1, 2 and 4 hold and all players run OptDA+ with adaptive learning rates (Adapt). Assume additionally Assumption 3 with $\sigma_A = 0$. Then, for any $i \in \mathcal{N}$ and bounded set $\mathcal{K}^i$, we have*

$$
\mathbb{E}\left[\max_{p^i \in \mathcal{K}^i} \sum_{t=1}^{T}\langle V^i(\mathbf{X}_{t+\frac{1}{2}}), X^i_{t+\frac{1}{2}} - p^i\rangle\right] = \mathcal{O}\left(\exp\left(\frac{1}{2q}\right)\right).
$$

*Proof.* Combining Lemma 20 and Lemma 22 gives the desired result. $\qquad\square$

# H   Last-iterate Convergence

We close our appendix with proofs on almost-sure last-iterate convergence of the trajectories. The global proof schema was sketched in Section 6 for the particular case of Theorem 4 (which corresponds to the upcoming Theorem 17). To prove last-iterate convergence we make heavy use of the different results that we derived in previous sections.

## H.1   Lemmas on Stochastic Sequences

To begin, we state several basic lemmas concerning stochastic sequences. The first one translates a bound of expectation into almost sure boundedness and convergence. It is a special case of Doob's martingale convergence theorem [25], but we also provide another elementary proof below. For simplicity, throughout the sequel, we use the term finite random variable to refer to those random variables which are finite almost surely.

**Lemma 23.** *Let $(U_t)_{t \in \mathbb{N}}$ be a sequence of non-decreasing and non-negative real-valued random variables. If there exists constant $C \in \mathbb{R}$ such that*

$$\forall\, t \in \mathbb{N}, \quad \mathbb{E}[U_t] \leq C.$$

*Then $(U_t)_{t \in \mathbb{N}}$ converges almost surely to a finite random variable. In particular, for any sequence of non-negative real-valued random variables $(\chi_t)_{t \in \mathbb{N}}$, the fact that $\sum_{t=1}^{+\infty} \mathbb{E}[\chi_t] < +\infty$ implies $\sum_{t=1}^{+\infty} \chi_t < +\infty$ almost surely, and accordingly $\lim_{t \to +\infty} \chi_t = 0$ almost surely.*

*Proof.* Let $U_\infty$ be the pointwise limit of $(U_t)_{t \in \mathbb{N}}$. Applying Beppo Levi's lemma we deduce that $U_\infty$ is also measurable and $\lim_{t \to +\infty} \mathbb{E}[U_t] = \mathbb{E}[U_\infty]$. Accordingly, $\mathbb{E}[U_\infty] \leq C$. The random variable $U_\infty$ being non-negative, $\mathbb{E}[U_\infty] \leq C < +\infty$ implies that $U_\infty$ is finite almost surely, which concludes the first statement of the lemma. The second statement is derived from the first statement by setting $U_t = \sum_{s=1}^{t} \chi_s$. $\qquad \square$

The next lemma is essential for building almost sure last-iterate converge in the case of vanishing learning rates, as it allows to extract a convergent subsequence.

**Lemma 24.** *Let $(U_t)_{t \in \mathbb{N}}$ be a sequence of non-negative real-valued random variables such that*

$$\liminf_{t \to +\infty} \mathbb{E}[U_t] = 0.$$

*Then, i) there exists a subsequence $(U_{\omega(t)})_{t \in \mathbb{N}}$ of $(U_t)_{t \in \mathbb{N}}$ that converges to $0$ almost surely;[6] and accordingly ii) it holds almost surely that $\liminf_{t \to +\infty} U_t = 0$.*

*Proof.* Since $\liminf_{t \to +\infty} \mathbb{E}[U_t] = 0$, we can extract a subsequence $(U_{\omega(t)})_{t \in \mathbb{N}}$ such that for all $t \in \mathbb{N}$, $\mathbb{E}[U_{\omega(t)}] \leq 2^{-t}$. This gives $\sum_{t=1}^{+\infty} \mathbb{E}[U_{\omega(t)}] < +\infty$ and invoking Lemma 23 we then know that $\sum_{t=1}^{+\infty} U_{\omega(t)} < +\infty$ almost surely, which in turn implies that $U_{\omega(t)}$ converges to $0$ almost surely. To prove (*ii*), we just notice that for any realization such that $\lim_{t \to +\infty} U_{\omega(t)} = 0$, we have $0 = \lim_{t \to +\infty} U_{\omega(t)} \geq \liminf_{t \to +\infty} U_t \geq 0$ and thus the equalities must hold, i.e., $\liminf_{t \to +\infty} U_t = 0$. $\qquad \square$

Another important building block is Robbins-Sigmunds's theorem that allows us to show almost sure convergence of the Lyapunov function to a finite random variable.

**Lemma 25** (Robbins and Sigmund [54]). *Consider a filtration $(\mathcal{G}_t)_{t \in \mathbb{N}}$ and four non-negative real-valued $(\mathcal{G}_t)_{t \in \mathbb{N}}$-adapted processes $(U_t)_{t \in \mathbb{N}}$, $(\alpha_t)_{t \in \mathbb{N}}$, $(\chi_t)_{t \in \mathbb{N}}$, $(\zeta_t)_{t \in \mathbb{N}}$ such that $\mathbb{E}[U_1] < +\infty$, $\sum_{t=1}^{+\infty} \mathbb{E}[\alpha_t] < \infty$, $\sum_{t=1}^{+\infty} \mathbb{E}[\chi_t] < \infty$, and for all $t \in \mathbb{N}$,*

$$\mathbb{E}[U_{t+1} \mid \mathcal{G}_t] \leq (1 + \alpha_t) U_t + \chi_t - \zeta_t.$$

*Then $(U_t)_{t \in \mathbb{N}}$ converges almost surely to a finite random variable and $\sum_{t=1}^{+\infty} \zeta_t < \infty$ almost surely.*

Finally, since the solution may not be unique, we need a to translate the result with respect to a single point to the one that applies to the entire set. This is achieved through the following lemma.

**Lemma 26.** *Let $\mathcal{K} \subseteq \mathbb{R}^d$ be a closed set, $(\mathbf{u}_t)_{t \in \mathbb{N}}$ be a sequence of $\mathbb{R}^d$-valued random variable, and $(\boldsymbol{\alpha}_t)_{t \in \mathbb{N}}$ be a sequence of $\mathbb{R}^N$-valued random variable such that*

*(a) For all $i \in \mathcal{N}$, $\alpha_1^i \geq 1$, $(\alpha_t^i)_{t \in \mathbb{N}}$ is non-decreasing and converges to a finite constant almost surely.*

*(b) For all $\mathbf{x} \in \mathcal{K}$, $\|\mathbf{u}_t - \mathbf{x}\|_{\boldsymbol{\alpha}_t}$ converges almost surely.*

*Then, with probability $1$, the vector $\boldsymbol{\alpha}_\infty = \lim_{t \to +\infty} \boldsymbol{\alpha}_t$ is well-defined, finite, and $\|\mathbf{u}_t - \mathbf{x}\|_{\boldsymbol{\alpha}_\infty}$ converges for all $\mathbf{x} \in \mathcal{K}$.*

*Proof.* As $\mathbb{R}^d$ is a separable metric space, $\mathcal{K}$ is also separable and we can find a countable set $\mathcal{Z}$ such that $\mathcal{K} = \mathrm{cl}(\mathcal{Z})$. Let us define the event

$$\mathcal{E} := \{ \boldsymbol{\alpha}_\infty = \lim_{t \to +\infty} \boldsymbol{\alpha}_t \text{ is well-defined and finite}; \ \|\mathbf{u}_t - \mathbf{z}\|_{\boldsymbol{\alpha}_t} \text{ converges for all } \mathbf{z} \in \mathcal{Z}. \} \quad (56)$$

---

[6]We remark that the choice of the subsequence does not depend on the realization but only the distribution of the random variables.

The set $\mathcal{Z}$ being countable, from (a) and (b) we then know that $\mathbb{P}(\mathcal{E}) = 1$. In the following, we show that $\|\mathbf{u}_t - \mathbf{x}\|_{\boldsymbol{\alpha}_\infty}$ converges for all $\mathbf{x} \in \mathcal{K}$ whenever $\mathcal{E}$ happens, which concludes our proof.

Let us now consider a realization of $\mathcal{E}$. We first establish the convergence of $\|\mathbf{u}_t - \mathbf{z}\|_{\boldsymbol{\alpha}_\infty}$ for any $\mathbf{z} \in \mathcal{Z}$. To begin, the convergence of $\|\mathbf{u}_t - \mathbf{z}\|_{\boldsymbol{\alpha}_t}$ implies the boundedness of this sequence, from which we deduce immediately the boundedness of $\|\mathbf{u}_t - \mathbf{z}\|$ as $\|\mathbf{u}_t - \mathbf{z}\| \leq \|\mathbf{u}_t - \mathbf{z}\|_{\boldsymbol{\alpha}_t}$ by $\boldsymbol{\alpha}_t \geq \boldsymbol{\alpha}_1 \geq 1$. In other words, $C = \sup_{t \in \mathbb{N}} \|\mathbf{u}_t - \mathbf{z}\|$ is finite. Furthermore, we have

$$0 \leq \|\mathbf{u}_t - \mathbf{z}\|_{\boldsymbol{\alpha}_\infty}^2 - \|\mathbf{u}_t - \mathbf{z}\|_{\boldsymbol{\alpha}_t}^2 = \sum_{i=1}^N (\alpha_\infty^i - \alpha_t^i) \|u_t^i - z^i\|^2 \leq \sum_{i=1}^N (\alpha_\infty^i - \alpha_t^i) C^2. \qquad (57)$$

Since $\alpha_\infty^i - \alpha_t^i$ converges to $0$ when $t$ goes to infinity, from (57) we get immediately $\lim_{t \to +\infty}(\|\mathbf{u}_t - \mathbf{z}\|_{\boldsymbol{\alpha}_\infty}^2 - \|\mathbf{u}_t - \mathbf{z}\|_{\boldsymbol{\alpha}_t}^2) = 0$. This shows that $\|\mathbf{u}_t - \mathbf{z}\|_{\boldsymbol{\alpha}_\infty}^2$ converges to $\lim_{t \to +\infty} \|\mathbf{u}_t - \mathbf{z}\|_{\boldsymbol{\alpha}_t}^2$, which exists by definition of $\mathcal{E}$. We have thus shown the convergence of $\|\mathbf{u}_t - \mathbf{z}\|_{\boldsymbol{\alpha}_\infty}$.

To conclude, we need to show that $\|\mathbf{u}_t - \mathbf{x}\|_{\boldsymbol{\alpha}_\infty}$ in fact converges for all $\mathbf{x} \in \mathcal{K}$. Let $\mathbf{x} \in \mathcal{K}$. As $\mathcal{Z}$ is dense in $\mathcal{K}$, there exists a sequence of points $(\mathbf{z}_k)_{k \in \mathbb{N}}$ with $\mathbf{z}_k \in \mathcal{Z}$ for all $k \in \mathbb{N}$ such that $\lim_{k \to +\infty} \mathbf{z}_k = \mathbf{x}$. For any $t, k \in \mathbb{N}$, the triangular inequality implies

$$-\|\mathbf{z}_k - \mathbf{x}\|_{\boldsymbol{\alpha}_\infty} \leq \|\mathbf{u}_t - \mathbf{x}\|_{\boldsymbol{\alpha}_\infty} - \|\mathbf{u}_t - \mathbf{z}_k\|_{\boldsymbol{\alpha}_\infty} \leq \|\mathbf{z}_k - \mathbf{x}\|_{\boldsymbol{\alpha}_\infty}.$$

Since $\mathbf{z}_k \in \mathcal{Z}$, we have shown that $\lim_{t \to +\infty} \|\mathbf{u}_t - \mathbf{z}_k\|$ exists. Subsequently, we get

$$\begin{aligned}
-\|\mathbf{z}_k - \mathbf{x}\|_{\boldsymbol{\alpha}_\infty} &\leq \liminf_{t \to +\infty} \|\mathbf{u}_t - \mathbf{x}\|_{\boldsymbol{\alpha}_\infty} - \lim_{t \to +\infty} \|\mathbf{u}_t - \mathbf{z}_k\|_{\boldsymbol{\alpha}_\infty} \\
&\leq \limsup_{t \to +\infty} \|\mathbf{u}_t - \mathbf{x}\|_{\boldsymbol{\alpha}_\infty} - \lim_{t \to +\infty} \|\mathbf{u}_t - \mathbf{z}_k\|_{\boldsymbol{\alpha}_\infty} \\
&\leq \|\mathbf{z}_k - \mathbf{x}\|_{\boldsymbol{\alpha}_\infty}.
\end{aligned}$$

Taking the limit as $k \to +\infty$, we deduce that $\lim_{k \to +\infty} \lim_{t \to +\infty} \|\mathbf{u}_t - \mathbf{z}_k\|_{\boldsymbol{\alpha}_\infty}$ exists and

$$\liminf_{t \to +\infty} \|\mathbf{u}_t - \mathbf{x}\|_{\boldsymbol{\alpha}_\infty} = \lim_{k \to +\infty} \lim_{t \to +\infty} \|\mathbf{u}_t - \mathbf{z}_k\|_{\boldsymbol{\alpha}_\infty} = \limsup_{t \to +\infty} \|\mathbf{u}_t - \mathbf{x}\|_{\boldsymbol{\alpha}_\infty}.$$

This shows the convergence of $\|\mathbf{u}_t - \mathbf{x}\|_{\boldsymbol{\alpha}_\infty}$. $\qquad \square$

**Corollary 3.** *Let $\mathcal{K} \subseteq \mathbb{R}^d$ be a closed set, $(\mathbf{u}_t)_{t \in \mathbb{N}}$ be a sequence of $\mathbb{R}^d$-valued random variable, and $\boldsymbol{\alpha} \in \mathbb{R}^N$ such that $\alpha^i \geq 1$ for all $i \in \mathcal{N}$, and for all $\mathbf{x} \in \mathcal{K}$, $\|\mathbf{u}_t - \mathbf{x}\|_{\boldsymbol{\alpha}}$ converges almost surely. Then, with probability 1, $\|\mathbf{u}_t - \mathbf{x}\|_{\boldsymbol{\alpha}}$ converges for all $\mathbf{x} \in \mathcal{K}$.*

### H.2 Trajectory Convergence of OG+ under Additive Noise

We start by proving the almost sure last-iterate convergence of OG+ under additive noise. This proof is, in a sense, the most technical once the results of the previous sections are established. This is because with vanishing learning rates, we cannot show that every cluster point of $(\mathbf{X}_{t+\frac{1}{2}})_{t \in \mathbb{N}}$ is a Nash equilibrium with probability 1. Instead we need to work with subsequences.

We will prove the convergence of $\mathbf{X}_t$ and $\mathbf{X}_{t+\frac{1}{2}}$ separately, and under relaxed learning rate requirements. For the convergence of $\mathbf{X}_t$, a learning rate condition introduced in [29] for double step-size EG is considered.

**Theorem 15.** *Let Assumptions 1–3 hold and all players run (OG+) with non-increasing learning rate sequences $(\gamma_t)_{t \in \mathbb{N}}$ and $(\eta_t)_{t \in \mathbb{N}}$ satisfying (4) and*

$$\sum_{t=1}^{+\infty} \gamma_t \eta_{t+1} = \infty, \quad \sum_{t=1}^{+\infty} \gamma_t^2 \eta_{t+1} < \infty, \quad \sum_{t=1}^{+\infty} \eta_t^2 < \infty. \qquad (58)$$

*Then, $\mathbf{X}_t$ converges almost surely to a Nash equilibrium.*

*Proof.* Our proof is divided into four steps. To begin, let us define $\tilde{\mathbf{X}}_1 = \mathbf{X}_1$ and for all $t \geq 2$,

$$\tilde{\mathbf{X}}_t = \mathbf{X}_t + \eta_t \boldsymbol{\xi}_{t-\frac{1}{2}} = \mathbf{X}_{t-1} - \eta_t \mathbf{V}(\mathbf{X}_{t-\frac{1}{2}})$$

Notice that $\tilde{\mathbf{X}}_t$ is $\mathcal{F}_{t-1}$-measurable. This surrogate of $\mathbf{X}_t$ plays an important role in the subsequent analysis.

(1) *With probability 1, $\|\tilde{\mathbf{X}}_t - \mathbf{x}_\star\|$ converges for all $\mathbf{x}_\star \in \mathcal{X}_\star$.* Let $x_\star \in \mathcal{X}_\star$. We would like to apply Robbins-Siegmund's theorem (Lemma 25) to the inequality of Lemma 5 with

$$
\begin{aligned}
\mathcal{G}_t &\leftarrow \mathcal{F}_{t-1}, \quad U_t \leftarrow \mathbb{E}_{t-1}[\|\mathbf{X}_t - \mathbf{x}_\star\|^2], \quad \alpha_t \leftarrow 0, \\
\zeta_t &\leftarrow \gamma_t \eta_{t+1}(\mathbb{E}_{t-1}[\|\mathbf{V}(\mathbf{X}_{t+\frac{1}{2}})\|^2] + \|\mathbf{V}(\mathbf{X}_{t-\frac{1}{2}})\|^2), \\
\chi_t &\leftarrow \mathbb{E}_{t-1}[3\gamma_t \eta_{t+1} N L^2((\eta_t^2 + \gamma_t^2)\|\hat{\mathbf{V}}_{t-\frac{1}{2}}\|^2 + (\gamma_{t-1})^2\|\hat{\mathbf{V}}_{t-\frac{3}{2}}\|^2) \\
&\qquad + (\gamma_t^2 \eta_{t+1} + N\eta_{t+1}(\eta_t + \gamma_t)^2)L\|\boldsymbol{\xi}_{t-\frac{1}{2}}\|^2 + (\eta_{t+1})^2\|\hat{\mathbf{V}}_{t+\frac{1}{2}}\|^2].
\end{aligned}
$$

As Lemma 5 only applies to $t \geq 2$, for $t = 1$ we use inequality (15). We thus choose $\zeta_1 = 0$ and $\chi_1 = \eta_2^2\|\hat{\mathbf{V}}_{3/2}\|^2$.

We claim that $\sum_{t=1}^{+\infty} \mathbb{E}[\chi_t] < +\infty$. In fact, following the proof of Proposition 5, we can deduce

$$
\sum_{t=1}^{+\infty} \mathbb{E}[\chi_t] \leq \underbrace{\sum_{t=1}^{+\infty} \gamma_t \eta_{t+1}(a_t(1 + \sigma_M^2) + b_t\sigma_M^2)\,\mathbb{E}[\|\mathbf{V}(\mathbf{X}_{t+\frac{1}{2}})\|^2]}_{(A)} + \underbrace{\sum_{t=1}^{+\infty} \gamma_t \eta_{t+1}(a_t + b_t)N\sigma_A^2}_{(B)},
$$

for $a_t = \eta_{t+1}/\gamma_t + 9\gamma_t^2 N L^2$ and $b_t = \gamma_t(4N + 1)L$. With our learning rate requirements it is true that $a_t(1 + \sigma_M^2) + b_t\sigma_M^2 \leq 3/2$, so Proposition 5 implies (A) is finite. On the other hand, from $\sum_{t=1}^{+\infty} \gamma_t^2\eta_{t+1} < +\infty$ and $\sum_{t=1}^{+\infty} \eta_t^2 < +\infty$ we deduce that (B) is also finite. We then conclude that it is effectively true that $\sum_{t=1}^{+\infty} \mathbb{E}[\chi_t] < +\infty$.

As a consequence, applying Robbins-Siegmund theorem gives the almost sure convergence of $\mathbb{E}_{t-1}[\|\mathbf{X}_t - \mathbf{x}_\star\|^2]$ to a finite random variable $U_\infty$. To proceed, we use the equality

$$
\mathbb{E}_{t-1}[\|\mathbf{X}_t - \mathbf{x}_\star\|^2] = \mathbb{E}_{t-1}[\|\tilde{\mathbf{X}}_t - \eta_t\boldsymbol{\xi}_{t-\frac{1}{2}} - \mathbf{x}_\star\|^2] = \|\tilde{\mathbf{X}}_t - \mathbf{x}_\star\|^2 + \eta_t^2\,\mathbb{E}_{t-1}[\|\boldsymbol{\xi}_{t-\frac{1}{2}}\|^2].
$$

Accordingly,

$$
\begin{aligned}
\sum_{t=2}^{+\infty} \mathbb{E}[\mathbb{E}_{t-1}[\|\mathbf{X}_t - \mathbf{x}_\star\|^2] - \|\tilde{\mathbf{X}}_t - \mathbf{x}_\star\|^2] &= \sum_{t=2}^{+\infty} \mathbb{E}[\eta_t^2\,\mathbb{E}_{t-1}[\|\boldsymbol{\xi}_{t-\frac{1}{2}}\|^2]] \\
&\leq \sum_{t=2}^{+\infty} \eta_t^2\,\mathbb{E}[\sigma_M^2\|\mathbf{V}(\mathbf{X}_{t-\frac{1}{2}})\|^2 + N\sigma_A^2] \\
&\leq \sum_{t=1}^{+\infty} (\gamma_t\eta_{t+1}\sigma_M^2\,\mathbb{E}[\|\mathbf{V}(\mathbf{X}_{t+\frac{1}{2}})\|^2] + (\eta_{t+1})^2 N\sigma_A^2) \\
&< +\infty.
\end{aligned}
$$
(59)

To obtain the last inequality we have applied *i)* Proposition 5; and *ii)* the summability of $(\eta_t^2)_{t\in\mathbb{N}}$. Invoking Lemma 23, we deduce that $\mathbb{E}_{t-1}[\|\mathbf{X}_t - \mathbf{x}_\star\|^2] - \|\tilde{\mathbf{X}}_t - \mathbf{x}_\star\|^2$ converges to 0 almost surely. This together with the almost sure convergence of $\mathbb{E}_{t-1}[\|\mathbf{X}_t - \mathbf{x}_\star\|^2]$ to $U_\infty$ we obtain the almost sure convergence of $\|\tilde{\mathbf{X}}_t - \mathbf{x}_\star\|^2$ to $U_\infty$.

To summarize, we have shown that for all $\mathbf{x}_\star \in \mathcal{X}_\star$, the distance $\|\tilde{\mathbf{X}}_t - \mathbf{x}_\star\|$ almost surely converges. Applying Corollary 3, we conclude that the event $\{\|\tilde{\mathbf{X}}_t - \mathbf{x}_\star\| \text{ converges for all } \mathbf{x}_\star \in \mathcal{X}_\star\}$ happens with probability 1.

(2) *There exists and increasing function $\omega\colon \mathbb{N} \to \mathbb{N}$ such that $\|\mathbf{V}(\mathbf{X}_{\omega(t)+\frac{1}{2}})\|^2 + \|\mathbf{X}_{\omega(t)+\frac{1}{2}} - \tilde{\mathbf{X}}_{\omega(t)}\|^2$ converges to 0 almost surely.* From Lemma 24, we know it is sufficient to show that

$$
\liminf_{t\to+\infty} \mathbb{E}[\|\mathbf{V}(\mathbf{X}_{t+\frac{1}{2}})\|^2 + \|\mathbf{X}_{t+\frac{1}{2}} - \tilde{\mathbf{X}}_t\|^2] = 0.
$$

Since $\sum_{t=1}^{+\infty} \gamma_t\eta_{t+1} = +\infty$ in all the cases, the above is implied by

$$
\sum_{t=2}^{+\infty} \gamma_t\eta_{t+1}\,\mathbb{E}[\|\mathbf{V}(\mathbf{X}_{t+\frac{1}{2}})\|^2 + \|\mathbf{X}_{t+\frac{1}{2}} - \tilde{\mathbf{X}}_t\|^2] < +\infty. \tag{60}
$$

Using Assumption 3 and $\eta_t < \gamma_t$, we have

$$\mathbb{E}[\|\mathbf{X}_{t+\frac{1}{2}} - \tilde{\mathbf{X}}_t\|^2] = \mathbb{E}[\|\gamma_t \mathbf{V}(\mathbf{X}_{t-\frac{1}{2}}) + (\eta_t + \gamma_t)\boldsymbol{\xi}_{t-\frac{1}{2}}\|^2]$$
$$= \gamma_t^2 \mathbb{E}[\|\mathbf{V}(\mathbf{X}_{t-\frac{1}{2}})\|^2] + (\eta_t + \gamma_t)^2 \mathbb{E}[\|\boldsymbol{\xi}_{t-\frac{1}{2}}\|^2]$$
$$\leq \gamma_t^2 (1 + 4\sigma_M^2) \mathbb{E}[\|\mathbf{V}(\mathbf{X}_{t-\frac{1}{2}})\|^2] + 4\gamma_t^2 N\sigma_A^2.$$

Subsequently, with Proposition 5, the summability of $(\gamma_t^2 \eta_{t+1})_{t \in \mathbb{N}}$ and the fact that the learning rates are non-increasing, we obtain

$$\sum_{t=2}^{+\infty} \gamma_t \eta_{t+1} \|\mathbf{X}_{t+\frac{1}{2}} - \tilde{\mathbf{X}}_t\|^2 \leq \sum_{t=2}^{+\infty} \gamma_t \eta_{t+1} \mathbb{E}[\gamma_t^2 (1 + 4\sigma_M^2)\|\mathbf{V}(\mathbf{X}_{t-\frac{1}{2}})\|^2 + 4\gamma_t^2 N\sigma_A^2]$$
$$\leq \sum_{t=1}^{+\infty} \gamma_1^2 \gamma_t \eta_{t+1}(1 + 4\sigma_M^2)\|\mathbf{V}(\mathbf{X}_{t+\frac{1}{2}})\|^2 + \sum_{t=1}^{+\infty} 4\gamma_1 \gamma_t^2 \eta_{t+1} N\sigma_A^2$$
$$< +\infty.$$

Invoking Proposition 5 again gives $\sum_{t=2}^{+\infty} \gamma_t \eta_{t+1} \mathbb{E}[\|\mathbf{V}(\mathbf{X}_{t+\frac{1}{2}})\|^2] < +\infty$ and thus we have effectively (60). This concludes the proof of this step.

(3) $(\tilde{\mathbf{X}}_t)_{t \in \mathbb{N}}$ *converges to a point in $\mathcal{X}_\star$ almost surely.*  Let us define the event

$$\mathcal{E} = \{\|\tilde{\mathbf{X}}_t - \mathbf{x}_\star\| \text{ converges for all } \mathbf{x}_\star \in \mathcal{X}_\star; \ \|\mathbf{V}(\mathbf{X}_{\omega(t)+\frac{1}{2}})\|^2 + \|\mathbf{X}_{\omega(t)+\frac{1}{2}} - \tilde{\mathbf{X}}_{\omega(t)}\|^2 \text{ converges to } 0\}$$

Combining the aforementioned two points we know that $\mathbb{P}(\mathcal{E}) = 1$. It is thus sufficient to show that $(\tilde{\mathbf{X}}_t)_{t \in \mathbb{N}}$ converges to a point in $\mathcal{X}_\star$ for any realization $\mathcal{E}$.

Let us consider a realization of $\mathcal{E}$. The set $\mathcal{X}_\star$ being non-empty, the convergence of $\|\tilde{\mathbf{X}}_t - \mathbf{x}_\star\|$ for a $\mathbf{x}_\star \in \mathcal{X}_\star$ implies the boundedness of $(\tilde{\mathbf{X}}_t)_{t \in \mathbb{N}}$. Therefore, we can extract a subsequence of $(\tilde{\mathbf{X}}_{\omega(t)})_t$, which we denote by $(\tilde{\mathbf{X}}_{\omega(\psi(t))})_t$ that converges to a point $\mathbf{x}_\infty \in \mathcal{X}$. As $\lim_{t \to +\infty} \|\mathbf{X}_{\omega(\psi(t))+\frac{1}{2}} - \tilde{\mathbf{X}}_{\omega(\psi(t))}\|^2 = 0$, we deduce that $(\mathbf{X}_{\omega(\psi(t))+\frac{1}{2}})_t$ also converges to $\mathbf{x}_\infty \in \mathcal{X}$. Moreover, we also have $\lim_{t \to +\infty} \|\mathbf{V}(\mathbf{X}_{\omega(\psi(t))+\frac{1}{2}})\|^2 = 0$. By continuity of $\mathbf{V}$ we then know that $\mathbf{V}(\mathbf{x}_\infty) = 0$, i.e., $\mathbf{x}_\infty \in \mathcal{X}_\star$. By definition of $\mathcal{E}$, this implies the convergence of $\|\tilde{\mathbf{X}}_t - \mathbf{x}_\infty\|$. The limit $\lim_{t \to +\infty} \|\tilde{\mathbf{X}}_t - \mathbf{x}_\infty\|$ is thus well defined and $\lim_{t \to +\infty} \|\tilde{\mathbf{X}}_t - \mathbf{x}_\infty\| = \lim_{t \to +\infty} \|\tilde{\mathbf{X}}_{\omega(\psi(t))} - \mathbf{x}_\infty\|$. However, $\lim_{t \to +\infty} \|\tilde{\mathbf{X}}_{\omega(\psi(t))} - \mathbf{x}_\infty\| = 0$ by the choice of $\mathbf{x}_\infty$. We have therefore $\lim_{t \to +\infty} \|\tilde{\mathbf{X}}_t - \mathbf{x}_\infty\| = 0$. Recalling that $\mathbf{x}_\infty \in \mathcal{X}_\star$, we have indeed shown that $(\tilde{X}_t)_{t \in \mathbb{N}}$ converges to a point in $\mathcal{X}_\star$.

(4) *Conclude: $(\mathbf{X}_t)_{t \in \mathbb{N}}$ converges to a point in $\mathcal{X}_\star$ almost surely .*   We claim that $\|\mathbf{X}_t - \tilde{\mathbf{X}}_t\|$ converges to 0. In fact, similar to (59), it holds that

$$\sum_{t=1}^{+\infty} \mathbb{E}[\|\mathbf{X}_t - \tilde{\mathbf{X}}_t\|^2] = \sum_{t=2}^{+\infty} \eta_t^2 \mathbb{E}[\|\boldsymbol{\xi}_{t-\frac{1}{2}}\|^2] < +\infty.$$

Invoking Lemma 23 we get almost sure convergence of $\|\mathbf{X}_t - \tilde{\mathbf{X}}_t\|$ to 0. Moreover, we have shown in the previous point that $(\tilde{\mathbf{X}}_t)_{t \in \mathbb{N}}$ converges to a point in $\mathcal{X}_\star$ almost surely. Combining the above two arguments we obtain the almost sure convergence of $(\mathbf{X}_t)_{t \in \mathbb{N}}$ to a point in $\mathcal{X}_\star$. $\qquad \square$

Provided that the players use larger extrapolation steps, the convergence of $\mathbf{X}_t$ does not necessarily imply the convergence of $\mathbf{X}_{t+\frac{1}{2}}$. The next theorem derives sufficient condition for the latter to hold.

**Theorem 16.** *Let Assumptions 1–3 hold and all players run (OG+) with non-increasing learning rate sequences $(\gamma_t)_{t \in \mathbb{N}}$ and $(\eta_t)_{t \in \mathbb{N}}$ satisfying (4) and (58). Assume further that $\gamma_t^3 = \mathcal{O}(\eta_t)$ and there exists $r \in (2, 4]$ and $\sigma > 0$ such that $\mathbb{E}[\|\boldsymbol{\xi}_t\|^r] \leq \sigma^r$ for all $t$ and $\sum_{t=1}^{+\infty} \gamma^r < \infty$. Then, the actual point of play $\mathbf{X}_{t+\frac{1}{2}}$ converges almost surely to a Nash equilibrium.*

*Proof.* Since we already know that $(\mathbf{X}_t)_{t \in \mathbb{N}}$ converges to a point in $\mathcal{X}_\star$ almost surely, it is sufficient to show that $\lim_{t \to +\infty} \|\mathbf{X}_t - \mathbf{X}_{t+\frac{1}{2}}\| = 0$ almost surely. By the update rule of OG+, we have, for

$t \geq 2$, $\mathbf{X}_t - \mathbf{X}_{t+\frac{1}{2}} = \gamma_t \mathbf{V}(\mathbf{X}_{t-\frac{1}{2}}) + \gamma_t \boldsymbol{\xi}_{t-\frac{1}{2}}$. We will deal with the two terms separately. For the noise term, we notice that under the additional assumptions we have

$$\sum_{t=2}^{+\infty} \mathbb{E}[\|\gamma_t \boldsymbol{\xi}_{t-\frac{1}{2}}\|^r] \leq \sum_{t=2}^{+\infty} \gamma_t^r \sigma^r < +\infty.$$

Therefore, applying Lemma 23 gives the almost sure convergence of $\|\gamma_t \boldsymbol{\xi}_{t-\frac{1}{2}}\|$ to 0. As for the operator term, for $t \geq 3$ we bound

$$\|\gamma_t \mathbf{V}(\mathbf{X}_{t-\frac{1}{2}})\| \leq \gamma_t \|\mathbf{V}(\mathbf{X}_{t-\frac{1}{2}}) - \mathbf{V}(\mathbf{X}_{t-1})\| + \gamma_t \|\mathbf{V}(\mathbf{X}_{t-1})\|.$$

On one hand, as $(\mathbf{X}_t)_{t \in \mathbb{N}}$ converges to a point in $\mathcal{X}_\star$ almost surely, the term $\gamma_t \|\mathbf{V}(\mathbf{X}_{t-1})\|$ converges to 0 almost surely by continuity of $\mathbf{V}$. On the other hand, by Lipschitz continuity of $\mathbf{V}$ we have

$$\sum_{t=2}^{+\infty} \mathbb{E}[(\gamma_{t+1})^2 \|\mathbf{V}(\mathbf{X}_{t+\frac{1}{2}}) - \mathbf{V}(\mathbf{X}_t)\|^2] \leq \sum_{t=2}^{+\infty} (\gamma_{t+1})^2 \gamma_t^2 N L^2 \, \mathbb{E}[\|\hat{\mathbf{V}}_{t-\frac{1}{2}}\|^2]$$

$$\leq \sum_{t=2}^{+\infty} \gamma_t^4 N L^2 \, \mathbb{E}[\|\mathbf{V}(\mathbf{X}_{t-\frac{1}{2}})\|^2] + \sum_{t=2}^{+\infty} \gamma_t^4 N L^2 \, \mathbb{E}[\|\boldsymbol{\xi}_{t-\frac{1}{2}}\|^2].$$
(61)

Since $\gamma_t^3 = \mathcal{O}(\eta_t)$, there exists $C \in \mathbb{R}_+$ such that $\gamma_t^3 \leq C \eta_t$ for all $t \in \mathbb{N}$. Along with Proposition 5 we get

$$\sum_{t=2}^{+\infty} \gamma_t^4 N L^2 \, \mathbb{E}[\|\mathbf{V}(\mathbf{X}_{t-\frac{1}{2}})\|^2] \leq \sum_{t=2}^{+\infty} \gamma_{t-1} \eta_t C N L^2 \, \mathbb{E}[\|\mathbf{V}(\mathbf{X}_{t-\frac{1}{2}})\|^2] < +\infty. \tag{62}$$

Since $r > 2$, by Jensen's inequality $\mathbb{E}[\|\boldsymbol{\xi}_t\|^r] \leq \sigma^r$ implies $\mathbb{E}[\|\boldsymbol{\xi}_t\|^2] \leq \sigma^2$. Along with $r \leq 4$ and $\sum_{t=1}^{+\infty} \gamma_t^r < +\infty$ we deduce

$$\sum_{t=2}^{+\infty} \gamma_t^4 N L^2 \, \mathbb{E}[\|\boldsymbol{\xi}_{t-\frac{1}{2}}\|^2] \leq \sum_{t=2}^{+\infty} \gamma_t^r \gamma_1^{4-r} N L^2 \sigma^2 < +\infty. \tag{63}$$

Combining (61), (62), and (63) we obtain $\sum_{t=2}^{+\infty} \mathbb{E}[(\gamma_{t+1})^2 \|\mathbf{V}(\mathbf{X}_{t+\frac{1}{2}}) - \mathbf{V}(\mathbf{X}_t)\|^2] < +\infty$, which implies $\lim_{t \to +\infty} \gamma_{t+1} \|\mathbf{V}(\mathbf{X}_{t+\frac{1}{2}}) - \mathbf{V}(\mathbf{X}_t)\| = 0$ using Lemma 23. In summary, we have shown the three sequences $(\gamma_t \|\boldsymbol{\xi}_{t-\frac{1}{2}}\|)_{t \in \mathbb{N}}$, $(\gamma_t \|\mathbf{V}(\mathbf{X}_{t-1})\|)_{t \in \mathbb{N}}$, and $(\gamma_t \|\mathbf{V}(\mathbf{X}_{t-\frac{1}{2}}) - \mathbf{V}(\mathbf{X}_{t-1})\|)_{t \in \mathbb{N}}$ converge almost surely to 0. As we have

$$\|\mathbf{X}_t - \mathbf{X}_{t+\frac{1}{2}}\| = \|\gamma_t \mathbf{V}(\mathbf{X}_{t-\frac{1}{2}}) + \gamma_t \boldsymbol{\xi}_{t-\frac{1}{2}}\| \leq \gamma_t \|\mathbf{V}(\mathbf{X}_{t-\frac{1}{2}}) - \mathbf{V}(\mathbf{X}_{t-1})\| + \gamma_t \|\mathbf{V}(\mathbf{X}_{t-1})\| + \gamma_t \|\boldsymbol{\xi}_{t-\frac{1}{2}}\|,$$

we can indeed conclude that $\lim_{t \to +\infty} \|\mathbf{X}_t - \mathbf{X}_{t+\frac{1}{2}}\| = 0$ almost surely. $\square$

### H.3 Trajectory Convergence of Non-Adaptive OptDA+ under Multiplicative Noise

We now turn to the case of multiplicative noise and prove almost sure last-iterate convergence with constant learning rates.

**Theorem 17.** *Let Assumptions 1–3 hold with $\sigma_A = 0$ and all players run (OG+) / (OptDA+) with learning rates given in Theorem 2(b). Then, both $\mathbf{X}_t$ and $\mathbf{X}_{t+\frac{1}{2}}$ converge almost surely to a Nash equilibrium.*

*Proof.* As in the proof Theorem 15, we define $\tilde{\mathbf{X}}_1 = \mathbf{X}_1$ and for all $i \in \mathcal{N}$, $t \geq 2$,

$$\tilde{X}_t^i = X_t^i + \eta^i \xi_{t-\frac{1}{2}}^i = -\eta^i \sum_{s=1}^{t-2} \hat{V}_{t+\frac{1}{2}}^i - \eta^i V^i(\mathbf{X}_{t-\frac{1}{2}}).$$

$\tilde{\mathbf{X}}_t$ serves a surrogate for $\mathbf{X}_t$ and is $\mathcal{F}_{t-1}$-measurable. Our first step is to show that

*With probability* 1, $\|\tilde{\mathbf{X}}_t - \mathbf{x}_\star\|_{1/\boldsymbol{\eta}}$ *converges for all* $\mathbf{x}_\star \in \mathcal{X}_\star$.

For this, we fix $\mathbf{x}_\star \in \mathcal{X}_\star$ and apply Robbins-Siegmund's theorem (Lemma 25) to inequality (13) of Lemma 7 with

$$\mathcal{G}_t \leftarrow \mathcal{F}_{t-1}, \quad U_t \leftarrow \mathbb{E}_{t-1}[\|\mathbf{X}_t - \mathbf{x}_\star\|_{1/\boldsymbol{\eta}}^2], \quad \alpha_t \leftarrow 0, \quad \zeta_t \leftarrow \mathbb{E}_{t-1}[\|\mathbf{V}(\mathbf{X}_{t+\frac{1}{2}})\|_{\boldsymbol{\gamma}}^2] + \|\mathbf{V}(\mathbf{X}_{t-\frac{1}{2}})\|_{\boldsymbol{\gamma}}^2,$$

$$\chi_t \leftarrow \mathbb{E}_{t-1}[3\|\mathbf{V}(\mathbf{X}_t) - \mathbf{V}(\mathbf{X}_{t-1})\|_{\boldsymbol{\gamma}}^2 + (4N+1)L\|\boldsymbol{\xi}_{t-\frac{1}{2}}\|_{\boldsymbol{\gamma}^2}^2$$

$$+ 3L^2(\|\boldsymbol{\gamma}\|_1\|\hat{\mathbf{V}}_{t-\frac{1}{2}}\|_{\boldsymbol{\gamma}^2}^2 + \|\boldsymbol{\gamma}\|_1\|\hat{\mathbf{V}}_{t-\frac{3}{2}}\|_{\boldsymbol{\gamma}^2}^2) + 2\|\hat{\mathbf{V}}_{t+\frac{1}{2}}\|_{\boldsymbol{\eta}}^2].$$

For $t = 1$ we use (31); thus $\zeta_t = 0$ and $\chi_t = \|\hat{\mathbf{V}}_{3/2}\|_{\boldsymbol{\eta}}^2$. To see that Robbins-Siegmund's theorem is effectively applicable, we use Assumptions 1 and 3 with $\sigma_A = 0$ to establish[7]

$$\mathbb{E}[\chi_t] \leq \mathbb{E}[3\|\boldsymbol{\gamma}\|_\infty L^2 \|\mathbf{X}_t - \mathbf{X}_{t-1}\|^2$$

$$+ (\|\boldsymbol{\gamma}\|_\infty (4N+1)L\sigma_M^2 + 3\|\boldsymbol{\gamma}\|_\infty^2 NL^2(1+\sigma_M^2))\|\mathbf{V}(\mathbf{X}_{t-\frac{1}{2}})\|_{\boldsymbol{\gamma}}^2]$$

$$+ 3\|\boldsymbol{\gamma}\|_\infty^2 NL^2(1+\sigma_M^2)\|\mathbf{V}(\mathbf{X}_{t-\frac{3}{2}})\|_{\boldsymbol{\gamma}}^2 + 2(1+\sigma_M^2)\|\mathbf{V}(\mathbf{X}_{t+\frac{1}{2}})\|_{\boldsymbol{\eta}}^2].$$

With $2(1+\sigma_M^2)\boldsymbol{\eta} \leq \boldsymbol{\gamma}$, it follows immediately from Proposition 6 that $\sum_{t=1}^{+\infty} \mathbb{E}[\chi_t] < +\infty$. Robbins-Siegmund's theorem thus ensures the almost sure convergence of $\mathbb{E}_{t-1}[\|\mathbf{X}_t - \mathbf{x}_\star\|^2]$ to a finite random variable. By definition of $\tilde{X}_t^i$, we have

$$\mathbb{E}_{t-1}[\|X_t^i - x_\star^i\|^2] = \mathbb{E}_{t-1}[\|\tilde{X}_t^i - \eta^i \xi_{t-\frac{1}{2}}^i - x_\star^i\|^2] = \|\tilde{X}_t^i - x_\star^i\|^2 + (\eta^i)^2 \mathbb{E}_{t-1}[\|\xi_{t-\frac{1}{2}}^i\|^2].$$

Subsequently

$$\mathbb{E}_{t-1}[\|\mathbf{X}_t - \mathbf{x}_\star\|_{1/\boldsymbol{\eta}}^2] = \|\tilde{\mathbf{X}}_t - \mathbf{x}_\star\|_{1/\boldsymbol{\eta}}^2 + \mathbb{E}_{t-1}[\|\boldsymbol{\xi}_{t-\frac{1}{2}}\|_{\boldsymbol{\eta}}^2].$$

Therefore, by Assumption 3 with $\sigma_A = 0$ and Proposition 6 we get

$$\sum_{t=2}^{+\infty} \mathbb{E}[\mathbb{E}_{t-1}[\|\mathbf{X}_t - \mathbf{x}_\star\|_{1/\boldsymbol{\eta}}^2] - \|\tilde{\mathbf{X}}_t - \mathbf{x}_\star\|_{1/\boldsymbol{\eta}}^2] = \sum_{t=2}^{+\infty} \mathbb{E}[\|\boldsymbol{\xi}_{t-\frac{1}{2}}\|_{\boldsymbol{\eta}}^2] \leq \sum_{t=2}^{+\infty} \sigma_M^2 \, \mathbb{E}[\|\mathbf{V}(\mathbf{X}_{t-\frac{1}{2}})\|_{\boldsymbol{\eta}}^2] < +\infty.$$

Following the proof of Theorem 15, we deduce with the help of Lemma 23 and Corollary 3 that the claimed argument is effectively true, i.e., with probability 1, $\|\tilde{\mathbf{X}}_t - \mathbf{x}_\star\|_{1/\boldsymbol{\eta}}$ converges for all $\mathbf{x}_\star \in \mathcal{X}_\star$.

Since $\|\mathbf{X}_t - \tilde{\mathbf{X}}_t\|^2 = \|\boldsymbol{\xi}_{t-\frac{1}{2}}\|_{\boldsymbol{\eta}^2}^2$ and $\|\mathbf{X}_{t+\frac{1}{2}} - \tilde{\mathbf{X}}_t\|^2 = \sum_{i=1}^N \|\gamma^i \hat{V}_{t-\frac{1}{2}}^i + \eta^i \xi_{t-\frac{1}{2}}^i\|^2$ (for $t \geq 2$), applying the multiplicative noise assumption, Proposition 6, and Lemma 23 we deduce that both $\|\mathbf{X}_t - \tilde{\mathbf{X}}_t\|$ and $\|\mathbf{X}_{t+\frac{1}{2}} - \tilde{\mathbf{X}}_t\|$ converge to 0 almost surely. Moreover, Proposition 6 along with Lemma 23 also implies the almost sure convergence of $\|\mathbf{V}(X_{t+\frac{1}{2}})\|$ to 0. In summary, we have shown that the event

$$\mathcal{E} := \left\{ \begin{array}{c} \|\tilde{\mathbf{X}}_t - \mathbf{x}_\star\|_{1/\boldsymbol{\eta}} \text{ converges for all } \mathbf{x}_\star \in \mathcal{X}_\star, \\ \lim_{t \to +\infty} \|\mathbf{X}_t - \tilde{\mathbf{X}}_t\| = 0, \quad \lim_{t \to +\infty} \|\mathbf{X}_{t+\frac{1}{2}} - \tilde{\mathbf{X}}_t\| = 0, \quad \lim_{t \to +\infty} \|\mathbf{V}(\mathbf{X}_{t+\frac{1}{2}})\| = 0 \end{array} \right\}$$

happens almost surely. To conclude, we just need to show that $\mathbf{X}_t$ and $\mathbf{X}_{t+\frac{1}{2}}$ converge to a point in $\mathcal{X}_\star$ whenever $\mathcal{E}$ happens. The convergence of $\|\tilde{\mathbf{X}}_t - \mathbf{x}_\star\|_{1/\boldsymbol{\eta}}$ for a point $\mathbf{x}_\star$ in particular implies the boundedness of $(\tilde{\mathbf{X}}_t)_{t \in \mathbb{N}}$. Therefore, $(\tilde{\mathbf{X}}_t)_{t \in \mathbb{N}}$ has at least a cluster point, which we denote by $\mathbf{x}_\infty$. Provided that $\lim_{t \to +\infty} \|\mathbf{X}_{t+\frac{1}{2}} - \tilde{\mathbf{X}}_t\| = 0$, the point $\mathbf{x}_\infty$ is clearly also a cluster point of $(\mathbf{X}_{t+\frac{1}{2}})_{t \in \mathbb{N}}$. By $\lim_{t \to +\infty} \|\mathbf{V}(\mathbf{X}_{t+\frac{1}{2}})\| = 0$ and the continuity of $\mathbf{V}$ we then have $\mathbf{V}(\mathbf{x}_\infty) = 0$, i.e., $\mathbf{x}_\infty \in \mathcal{X}_\star$. This in turn implies that $\|\tilde{\mathbf{X}}_t - \mathbf{x}_\infty\|_{1/\boldsymbol{\eta}}$ converges, so this limit can only be 0. In other words, $(\tilde{\mathbf{X}}_t)_{t \in \mathbb{N}}$ converges to $\mathbf{x}_\infty$; we conclude by $\lim_{t \to +\infty} \|\mathbf{X}_t - \tilde{\mathbf{X}}_t\| = 0$ and $\lim_{t \to +\infty} \|\mathbf{X}_{t+\frac{1}{2}} - \tilde{\mathbf{X}}_t\| = 0$. □

## H.4 Trajectory Convergence of Adaptive OptDA+ under Multiplicative Noise

In closing, we prove the almost sure last-iterate convergence of adaptive OptDA+ under multiplicative noise. As claimed in Section 6, we first show that the learning rates almost surely converge to positive constant. This intuitively means that the analysis of the last section should apply as well.

---

[7]For $t = 1$ and $t = 2$, we remove the terms that involve either $\mathbf{X}_{1/2}$, $\mathbf{X}_0$, or $\mathbf{X}_{-1/2}$.

**Lemma 27.** *Let Assumptions 1–4 hold with $\sigma_A = 0$ and all players run OptDA+ with adaptive learning rates (Adapt). Then,*

(a) *With probability $1$, for all $i \in \mathcal{N}$, $(\lambda_t^i)_{t\in\mathbb{N}}$ and $(\mu_t^i)_{t\in\mathbb{N}}$ converge to finite constant.*

(b) *With probability $1$, for all $i \in \mathcal{N}$, the learning rates $(\gamma_t^i)_{t\in\mathbb{N}}$ and $(\eta_t^i)_{t\in\mathbb{N}}$ converge to positive constants.*

*Proof.* We notice that (b) is a direct consequence of (a) so we will only show (a) below. For this, we make use of Lemma 22 and Lemma 23. In fact, $(\sqrt{\lambda_t^i})_{t\in\mathbb{N}}$ is clearly non-decreasing and by Lemma 22, $\sup_{t\in\mathbb{N}}\mathbb{E}[\sqrt{\lambda_t^i}] < +\infty$. Therefore, Lemma 23 ensures the almost sure convergence of $(\sqrt{\lambda_t^i})_{t\in\mathbb{N}}$ to a finite random variable, which in turn implies that $(\lambda_t^i)_{t\in\mathbb{N}}$ converges to a finite constant almost surely. Similarly, $(\mu_t^i)_{t\in\mathbb{N}}$ is non-decreasing and $\sup_{t\in\mathbb{N}}\mathbb{E}[\mu_t^i] < +\infty$ by Lemma 22. We thus deduce by Lemma 23 that $(\mu_t^i)_{t\in\mathbb{N}}$ converges to finite constant almost surely. $\qquad\square$

We now adapt the proof of Theorem 17 to the case of adaptive learning rates. Note that the fact that the learning rates are not constant also causes some additional challenges.

**Theorem 18.** *Let Assumptions 1–4 hold with $\sigma_A = 0$ and all players run (OptDA+) with adaptive learning rates (Adapt). Then,*

(a) *It holds almost surely that $\sum_{t=1}^{+\infty}\|\mathbf{V}(\mathbf{X}_{t+\frac{1}{2}})\|^2 < +\infty$.*

(b) *Both $(\mathbf{X}_t)_{t\in\mathbb{N}}$ and $(\mathbf{X}_{t+\frac{1}{2}})_{t\in\mathbb{N}}$ converge to a Nash equilibrium almost surely.*

*Proof.* In the following, we define $\boldsymbol{\gamma}_\infty = \lim_{t\to+\infty}\boldsymbol{\gamma}_t$ and $\boldsymbol{\eta}_\infty = \lim_{t\to+\infty}\boldsymbol{\eta}_t$ as the limits of the learning rate sequences. Since for each $i \in \mathcal{N}$, $(\gamma_t^i)_{t\in\mathbb{N}}$ and $(\eta_t^i)_{t\in\mathbb{N}}$ are non-negative non-increasing sequences, both $\boldsymbol{\gamma}_\infty$ and $\boldsymbol{\eta}_\infty$ are well-defined. Moreover, by Lemma 27 we know that $\boldsymbol{\gamma}_\infty$ and $\boldsymbol{\eta}_\infty$ are positive almost surely.

(a) Combining Lemma 19 and Lemma 22 we get immediately $\sum_{t=1}^{+\infty}\mathbb{E}[\|\mathbf{V}(\mathbf{X}_{t+\frac{1}{2}})\|_{\boldsymbol{\gamma}_t}^2] < +\infty$. Therefore, using Lemma 23 we deduce that $\sum_{t=1}^{+\infty}\|\mathbf{V}(\mathbf{X}_{t+\frac{1}{2}})\|_{\boldsymbol{\gamma}_t}^2 < +\infty$ almost surely. By definition of $\boldsymbol{\gamma}_\infty$ we have

$$\sum_{t=1}^{+\infty}\|\mathbf{V}(\mathbf{X}_{t+\frac{1}{2}})\|_{\boldsymbol{\gamma}_t}^2 \geq \sum_{t=1}^{+\infty}\|\mathbf{V}(\mathbf{X}_{t+\frac{1}{2}})\|_{\boldsymbol{\gamma}_\infty}^2 \geq \min_{i\in\mathcal{N}}\gamma_\infty^i \sum_{t=1}^{+\infty}\|\mathbf{V}(\mathbf{X}_{t+\frac{1}{2}})\|^2$$

As a consequence, whenever *i)* $C := \sum_{t=1}^{+\infty}\|\mathbf{V}(\mathbf{X}_{t+\frac{1}{2}})\|_{\boldsymbol{\gamma}_t}^2$ is finite; and *ii)* $\min_{i\in\mathcal{N}}\gamma_\infty^i > 0$, we have

$$\sum_{t=1}^{+\infty}\|\mathbf{V}(\mathbf{X}_{t+\frac{1}{2}})\|^2 \leq \frac{C}{\min_{i\in\mathcal{N}}\gamma_\infty^i} < +\infty.$$

As both *i)* and *ii)* hold almost surely, we have indeed shown that $\sum_{t=1}^{+\infty}\|\mathbf{V}(\mathbf{X}_{t+\frac{1}{2}})\|^2 < +\infty$ almost surely.

(b) To prove this point, we follow closely the proof of Theorem 17. To begin, we fix $\mathbf{x}_\star \in \mathcal{X}_\star$ and show that we can always apply Robbins-Siegmund's theorem (Lemma 25) to inequality (13) of Lemma 7 (or inequality (31) for $t = 1$). This gives, for $t \geq 2$,

$$\mathcal{G}_t = \mathcal{F}_{t-1}, \quad U_t = \mathbb{E}_{t-1}[\|\mathbf{X}_t - \mathbf{x}_\star\|_{1/\boldsymbol{\eta}_t}^2], \quad \alpha_t = 0, \quad \zeta_t = \mathbb{E}_{t-1}[\|\mathbf{V}(\mathbf{X}_{t+\frac{1}{2}})\|_{\boldsymbol{\gamma}_t}^2] + \|\mathbf{V}(\mathbf{X}_{t-\frac{1}{2}})\|_{\boldsymbol{\gamma}_t}^2,$$

$$\chi_t = \mathbb{E}_{t-1}[3\|\mathbf{V}(\mathbf{X}_t) - \mathbf{V}(\mathbf{X}_{t-1})\|_{\boldsymbol{\gamma}_t}^2 + \|\mathbf{X}_1 - \mathbf{x}_\star\|_{1/\boldsymbol{\eta}_{t+1}-1/\boldsymbol{\eta}_t}^2 + (4N+1)L\|\boldsymbol{\xi}_{t-\frac{1}{2}}\|_{\boldsymbol{\gamma}_t^2}^2 + 3L^2$$

$$(\|\boldsymbol{\gamma}_t\|_1\|\hat{\mathbf{V}}_{t-\frac{1}{2}}\|_{\boldsymbol{\gamma}_t^2}^2 + \|\boldsymbol{\gamma}_{t-1}\|_1\|\hat{\mathbf{V}}_{t-\frac{3}{2}}\|_{(\boldsymbol{\gamma}_{t-1})^2}^2) + 2\|\hat{\mathbf{V}}_{t+\frac{1}{2}}\|_{\boldsymbol{\eta}_t}^2].$$

As for $t = 1$, we replace the above with $\zeta_t = 0$ and $\chi_t = \|\hat{\mathbf{V}}_{3/2}\|_{\boldsymbol{\eta}_1}^2$. Using Assumption 1, (44), and (45), we can bound the sum of the expectation of $\chi_t$ by

$$\sum_{t=1}^{T}\mathbb{E}[\chi_t] \leq \sum_{t=1}^{T-1}3L^2\,\mathbb{E}[\|\mathbf{X}_t - \mathbf{X}_{t+1}\|^2] + \sum_{i=1}^{N}\left(\|X_1^i - x_\star^i\|^2\,\mathbb{E}\left[\sqrt{1 + \lambda_{T-1}^i + \mu_{T-1}^i}\right]\right)$$

$$+ (6NL^2 + (4N+1)L) \left( 2N(G^2 + \bar{\sigma}^2) + \sum_{i=1}^{N} 2\,\mathbb{E}\left[ \sqrt{\lambda_{T-1}^i} \right] \right) +$$

$$+ 8N(G^2 + \bar{\sigma}^2) + \sum_{i=1}^{N} 4\,\mathbb{E}\left[ \sqrt{\lambda_T^i} \right]$$

It then follows immediately from Lemma 22 that $\sum_{t=1}^{+\infty} \mathbb{E}[\chi_t] < +\infty$. With Robbins-Siegmund's theorem we deduce that $\mathbb{E}_{t-1}[\|\mathbf{X}_t - \mathbf{x}_\star\|_{1/\boldsymbol{\eta}_t}^2]$ converges almost surely to a finite random variable.

As in the proof of Theorems 15 and 17, we next define $\tilde{\mathbf{X}}_1 = \mathbf{X}_1$ and for all $i \in \mathcal{N}$, $t \geq 2$,

$$\tilde{X}_t^i = X_t^i + \eta_t^i \xi_{t-\frac{1}{2}}^i = -\eta_t^i \sum_{s=1}^{t-2} \hat{V}_{t+\frac{1}{2}}^i - \eta_t^i V^i(\mathbf{X}_{t-\frac{1}{2}}).$$

Then,

$$\mathbb{E}_{t-1}[\|\mathbf{X}_t - \mathbf{x}_\star\|_{1/\boldsymbol{\eta}_t}^2] = \|\tilde{\mathbf{X}}_t - \mathbf{x}_\star\|_{1/\boldsymbol{\eta}_t}^2 + \mathbb{E}_{t-1}[\|\boldsymbol{\xi}_{t-\frac{1}{2}}\|_{\boldsymbol{\eta}_t}^2].$$

Using $\mathbb{E}_{t-1}[\|\xi_{t-\frac{1}{2}}^i\|^2] \leq \mathbb{E}_{t-1}[\|\hat{V}_{t-\frac{1}{2}}^i\|^2]$, the law of total expectation, the fact that $\boldsymbol{\eta}_t$ is $\mathcal{F}_{t-1}$-measurable, Lemma 16, and Lemma 22, we then get

$$\sum_{t=2}^{+\infty} \mathbb{E}[\mathbb{E}_{t-1}[\|\mathbf{X}_t - \mathbf{x}_\star\|_{1/\boldsymbol{\eta}_t}^2] - \|\tilde{\mathbf{X}}_t - \mathbf{x}_\star\|_{1/\boldsymbol{\eta}_t}^2] = \sum_{t=2}^{+\infty} \mathbb{E}[\|\boldsymbol{\xi}_{t-\frac{1}{2}}\|_{\boldsymbol{\eta}_t}^2]$$

$$\leq \sum_{t=2}^{+\infty} \mathbb{E}[\|\hat{\mathbf{V}}_{t-\frac{1}{2}}\|_{\boldsymbol{\eta}_t}^2]$$

$$\leq 2N(G^2 + \bar{\sigma}^2) + \sup_{t \in \mathbb{N}} \sum_{i=1}^{N} 2\,\mathbb{E}\left[ \sqrt{\lambda_t^i} \right]$$

$$< +\infty. \tag{64}$$

Invoking Lemma 23 we deduce that $\mathbb{E}_{t-1}[\|\mathbf{X}_t - \mathbf{x}_\star\|_{1/\boldsymbol{\eta}_t}^2] - \|\tilde{\mathbf{X}}_t - \mathbf{x}_\star\|_{1/\boldsymbol{\eta}_t}^2$ almost surely converges to 0. Since we have shown $\mathbb{E}_{t-1}[\|\mathbf{X}_t - \mathbf{x}_\star\|_{1/\boldsymbol{\eta}_t}^2]$ almost surely converges to a finite random variable, we now know that $\|\tilde{\mathbf{X}}_t - \mathbf{x}_\star\|_{1/\boldsymbol{\eta}_t}^2$ almost surely converges to this finite random variable as well. To summarize, we have shown that for any $\mathbf{x}_\star \in \mathcal{X}_\star$, $\|\tilde{\mathbf{X}}_t - \mathbf{x}_\star\|_{1/\boldsymbol{\eta}_t}$ converges almost surely. Moreover, we also know that $(1/\boldsymbol{\eta}_\infty)$, the limit of $(1/\boldsymbol{\eta}_t)_{t \in \mathbb{N}}$ is finite almost surely. Therefore, applying Lemma 26 with $\mathcal{K} \leftarrow \mathcal{X}_\star$, $\mathbf{u}_t \leftarrow \tilde{\mathbf{X}}_t$, and $\boldsymbol{\alpha}_t \leftarrow 1/\boldsymbol{\eta}_t$, we deduce that with probability 1, the vector $1/\boldsymbol{\eta}_\infty$ is finite and $\|\tilde{\mathbf{X}}_t - \mathbf{x}_\star\|_{1/\boldsymbol{\eta}_\infty}$ converges for all $\mathbf{x}_\star \in \mathcal{X}_\star$.

Next, with $\|\mathbf{X}_t - \tilde{\mathbf{X}}_t\|^2 = \|\boldsymbol{\xi}_{t-\frac{1}{2}}\|_{\boldsymbol{\eta}_t^2}^2$ and $\|\mathbf{X}_{t+\frac{1}{2}} - \tilde{\mathbf{X}}_t\|^2 = \sum_{i=1}^{N} \|\gamma_t^i \hat{V}_{t-\frac{1}{2}}^i + \eta_t^i \xi_{t-\frac{1}{2}}^i\|^2$ (for $t \geq 2$), following the reasoning of (64), we get both $\sum_{t=1}^{+\infty} \mathbb{E}[\|\mathbf{X}_t - \tilde{\mathbf{X}}_t\|^2] < +\infty$ and $\sum_{t=1}^{+\infty} \mathbb{E}[\|\mathbf{X}_{t+\frac{1}{2}} - \tilde{\mathbf{X}}_t\|^2] < +\infty$. By Lemma 23 we then know that $\|\mathbf{X}_t - \tilde{\mathbf{X}}_t\|$ and $\|\mathbf{X}_{t+\frac{1}{2}} - \tilde{\mathbf{X}}_t\|$ converge to 0 almost surely. Finally, from point (a) we know that $\|\mathbf{V}(\mathbf{X}_{t+\frac{1}{2}})\|$ converges to 0 almost surely. To conclude, let us define the event

$$\mathcal{E} := \left\{ \begin{array}{c} 1/\boldsymbol{\eta}_\infty \text{ is finite and } \|\tilde{\mathbf{X}}_t - \mathbf{x}_\star\|_{1/\boldsymbol{\eta}_\infty} \text{ converges for all } \mathbf{x}_\star \in \mathcal{X}_\star, \\ \lim_{t \to +\infty} \|\mathbf{X}_t - \tilde{\mathbf{X}}_t\| = 0, \quad \lim_{t \to +\infty} \|\mathbf{X}_{t+\frac{1}{2}} - \tilde{\mathbf{X}}_t\| = 0, \quad \lim_{t \to +\infty} \|\mathbf{V}(\mathbf{X}_{t+\frac{1}{2}})\| = 0 \end{array} \right\}$$

We have shown that $\mathbb{P}(\mathcal{E}) = 1$. Moreover, with the arguments of Theorem 17 we deduce that whenever $\mathcal{E}$ happens both $(\mathbf{X}_t)_{t \in \mathbb{N}}$ and $(\mathbf{X}_{t+\frac{1}{2}})_{t \in \mathbb{N}}$ converge to a point in $\mathcal{X}_\star$, and this ends the proof. $\qquad \square$