# OpenReview forum: "No-regret learning in games with noisy feedback: Faster rates and adaptivity via learning rate separation"
_NeurIPS.cc/2022/Conference — NeurIPS 2022 Accept_

### Official Review · Reviewer_sxiN · 2022-06-28

**Rating:** 7
**Confidence:** 4
**Soundness:** 4 excellent
**Presentation:** 4 excellent
**Contribution:** 4 excellent

**Summary:**

The paper considers learning in variationally stable games, with noisy gradient feedback. The cost function of each player is assumed to be convex and smooth. It is shown that optimistic gradients methods can achieve constant regret with multiplicative noise while still achieving O(sqrt(T)) for additive noise. The key novelty in the methods is to separate the step-size sequence that multiplies the current gradient with the sequence that multiplies the "optimistic" term. Two methods are proposed, where the second one is shown to also maintain O(sqrt(T)) regret against an adversary. An adaptive version of both is shown that doesn't need to be tuned. Finally, convergence to Nash equilibrium is shown if all players use one of these algorithms (even if with different sequences).

**Questions:**

1) Do you see any meaning to the regret against other players beyond its connection with convergence to equilibrium? (average or last iterate)?

2) Do you have a sense of whether the O(sqrt(T)) result for additive noise is tight? in other words, what is the answer to the main question in line 52 for noise that is more general than multiplicative? if unknown, I think it would be interesting to state that in the concluding remakes.

**Limitations:**

This is a theory paper with no societal impact. The paper clearly states all technical limitations.

**Strengths And Weaknesses:**

The paper is very well-written, and the technical contribution is solid and significant. The paper advances the understanding of gradient learning methods in variationally stable games, which to the best of my knowledge are the largest class of games where gradients methods are known to converge.  It was definitely interesting and enjoyable to read. I believe the paper should be accepted.

My main issue is with the presentation and position of the results. I'll admit this is highly subjective, but perhaps can be useful when revising the paper:

The paper can easily be made more interesting to a broader audience. As written, the paper is mainly interesting for readers who already agree that the technical questions tackled here are interesting. I think that the main issue is that the notion of regret loses its meaning against other players since the benchmark is arbitrary - it is how much the best action in hindsight would achieve if for some reason the other players were forced to play the same thing they played against the algorithm, without responding to this best action. Indeed, the technical reason this regret is still interesting (at least to me) is that gives convergence to equilibrium, which in this case is the Nash equilibrium, which is even the last iterate convergence rather than the immediate average iterate convergence (given the regret). While the statement in Line 25 about the learner's objective makes sense against an oblivious adversary (or a stochastic environment), I'm not sure if it still does for multiplayer environments. Why would a learner concern itself with optimizing the regret against other players, if the benchmark has no meaning? Low regret doesn't show that "the sequence is efficient" in the multi-player case. Without the convergence rate to Nash as motivating, going beyond the O(sqrt(T)) lower bound therefore seems purely mathematical. Of course, I might have missed something here, so please see the question below.

The way the introduction is written, I got the impression that the main goal of the paper is to prove lower regret guarantees for a multiplayer environment with noisy gradient feedback. However, this is only done for multiplicative noise. Where certainly impressive on its own, it gives an (unjustified) sense of disappointment as a partial answer to the question in line 52. I'm therefore not sure if this question properly motivates the great results in the paper.  At the same time, the paper acheives something very impressive with no qualifiers - it provides the first gradient methods that converge to Nash in variationally stable games with noisy gradient feedback, even if they're not strict (to the best of my knowledge, so please verify). It also provides convergence rates. Along with the fact that regret is an arbitrary performance measure against other players, I find this to be the main contribution of the paper. Then, discussing the comparison with [40] is necessary to highlight this contribution. I would consider highlighting this aspect of the results.

Minor comments:

1) What do you mean by superlinear regret in line 51? worse than O(T)?

2) Please provide a reference for "Nash equilibria coincide precisely with the zeros..." in line 106.

3) The V notation is easy to miss since it's in the text above Assumption 2, which doesn't state it's a definition.

4) A table mapping the results in this paper can help. I don't mean the conversion table in the appendix, but one that gives a brief meaning for each result.

5)  The notation X compared to x is confusing since it's usually reserved for random variables, whereas here both are random.

6)  Line 146 can emphasize that the 1/2 shifted sequence is the actual sequence of actions.

7) Lines 150-151: here the utilities don't change, or do you mean from a single-player point of view? it's better to explain why you're interested in these methods. Then, "in certain classes of games" is vague, please provide a reference.

8) Line 160-161 should mention that the regret is constant only for multiplicative noise.

9)  In Theorem 3, provide the variance parameters to supplement the statement  "if the noise is multiplicative".

10) Line 340 - probably a typo, this is Theorem 5.

11) Line 345 - "relative" - relatively

12) Line 768 - "The proof is proved" - the result is proved.

13) Lemma 8 - "holds" - hold.

14) Unify "infinity" and "+infinity".

15) (24) - I think this should be \eta_{t+1}

---

> ### Author Response · Authors · 2022-08-02
> **Replies to Reviewer sxiN**
>
> Dear Reviewer,
>
> Thank you very much for your thoughtful input, detailed remarks, and positive evaluation.  We reply to your main questions point-by-point below (and we will of course include your remarks in our revision).
>
> 1. **On the use of regret in game-theoretic setting**
>
>     We agree that the notion of regret is a fairly weak performance criterion for a game-theoretic setting: given that players are not facing an arbitrary environment but each other, convergence to a Nash equilibrium (or, at the very least, some approximation / slight relaxation thereof) is a much more meaningful target. At the same time, the minimization of regret remains a minimal worst-case requirement for any learning algorithm, as players who do not know if they will be facing other rational players or a dispassionate nature would like to be able to do well against both (and without prior knowledge of which environment they will be called to operate in). [The classical example of a routing game is particularly apt in this context, as congestion could be caused by a confluence of both agent-driven and environment-driven factors (choice of routes and weather conditions for example).]
>
>     In this regard, we believe that developing efficient adaptive algorithms is a valuable endeavor. We acknowledge that our results can be tightened further in the adversarial setup, but we believe that they nevertheless shed light on several trade-offs that arise in this setting and can provide a stepping stone for further advances on the topic.
>
> 1. **On the tightness of the $O(\sqrt{T})$ regret bound**
>
>     To the best of our knowldge, the tightness of the $\mathcal{O}(\sqrt{T})$ bound for online convex optimization with gradient feedback is due to Abernethy et al. (COLT 2008, "Optimal strategies and minimax lower bounds for online convex games"). In the multi-armed bandit case, similar lower bounds seem to be folk, cf. the classical textbook of Cesa-Bianchi and Lugosi (2006). We are not aware of a lower bound for the specific learning model that we consider, but we conjecture that $O(\sqrt{T})$ is still tight here in the genuine stochastic case.
>
>     We will of course include a remark about the above -- thanks again for bringing it up.
>
> 1. **On superlinear regret**
>
>     Yes, by "superlinear" we mean that the regret grows faster than $\Theta(T)$.
>
> 1. **On trajectory convergence**
>
>     Thank you very much for highlighting this point. In Appendix B, we have a paragraph discussing the works that prove trajectory convergence for learning in games with noisy feedback. As far as we are aware, there are no previous works proving the result of *no-regret* algorithms that converge in all *variationally stable* games when the feedback is *noisy*. However, due to the discrepancy of the results that we manage to obtain for the three algorithms, we have made the decision to focus more on the regret bounds.

---

> > ### Comment · Reviewer_sxiN · 2022-08-04
> > **Thank you for the response**
> >
> > I'm already positive about the paper and believe it should be accepted.
> >
> > I still think that the regret result is the least exciting part of the paper, for the reasons detailed above (and below). On the other hand, the trajectory convergence to Nash is novel and exciting. This way or another, a paper where different readers can find different points of interest is certainly a good one. Highlighting some over others is a matter of presentation which is inherently subjective after all.
> >
> > For the sake of completeness:
> >
> > 1. I think that your response mixes two unrelated things. Worst case regret guarantees against an adversary is one thing, which I agree can be useful in practice. Improved regret for the case that the environment consists of other players is a separate thing, that makes sense only if the regret means anything as a benchmark against other players. Since this is not the case,  your algorithm actually doesn't have improved performance guarantees in a (noisy) game setting. Your work doesn't improve the worst-case guarantees against an adversary either, so I'm not sure how this information is relevant to this argument. Convergence to Nash, however, is highly relevant to defend the impressive technical results of this paper. These are my two cents.
> >
> > 2. One does not encounter superlinear regret on a daily basis. I guess this is possible in this setting since the action set is unbounded. Clarifying that in the text can avoid some confusion.

---

> > > ### Author Response · Authors · 2022-08-04
> > > **Further reply**
> > >
> > > Dear Reviewer,
> > >
> > > We would like to thank you again for your valuable feedback and positive evaluation. We truly appreciate it. Below we briefly reply to the two points mentioned in your response.
> > >
> > > 1. We agree that whether the notion of (external) regret is relevant to the multi-player setting beyond its connection to convergence to equilibrium is an important and fundamental question. This probably deserves more attention from the community. For future research, a promising direction is then to investigate whether our techniques can be adapted to prove bounds for other notions of regret such as policy regret ([Arora et al. NIPS 2018](https://arxiv.org/pdf/1811.04127.pdf)).
> > >
> > > 2. Yes it is possible to have superlinear regret here because the action set is unbounded. We will clarify this point in our revision. Thank you for bring this up.

---

### Official Review · Reviewer_NjYe · 2022-07-07

**Rating:** 7
**Confidence:** 3
**Soundness:** 4 excellent
**Presentation:** 3 good
**Contribution:** 2 fair

**Summary:**

The paper studies the expected regret achievable by individual players
in the general class of variationally stable games when the gradient
observations are perturbed by noise that can be either additive or
multiplicative. An important technique is learning rate separation for
optimistic methods, which refers to scaling the optimistic extrapolation
steps by a larger step size than the update steps. Learning rate
separation was previously introduced by Hsieh et al for optimistic
online gradient descent in the context of additive noise, but their
algorithm is vulnerable to adversarial attacks in early iterations, so
the present paper adapts it for optimistic dual
averaging/follow-the-regularized-leader. The paper considers two cases:

1. All players use (more or less) the same algorithm, which is described
in the paper;
2. Some of the players use an arbitrary strategy.

For case 1, Thms 1&2 show that all players can achieve constant regret
if the noise is multiplicative. (It is also shown that, with a different
step size tuning, they can get tilde O(sqrt{T}) regret, but that would
already be achievable using standard results, so this seems of interest
only as a lead-up to the adaptive results in the next section.)

In section 5, adaptive step sizes are introduced, which achieve a type of
best-of-both-worlds result:
* If some of the other players do not use the same algorithm, then they
  get O(T^{1/2 + q}) regret, where q is a hyperparameter of the
  algorithm. This is optimal for q=0, and slightly suboptimal otherwise.
* If all players use the same algorithm with these adaptive step sizes,
  then they get:
  - constant regret for q>0 if the noise is multiplicative, where the
    constant depends on q.
  - O(sqrt{T}) regret otherwise.

Finally, section 6 contains a trajectory analysis for the proposed
methods with non-adaptive step sizes.

Minor remarks:

* In Theorems 1 and 2, "Reg_{p^i}(T)" should be "Reg_T(p^i)" and the
  statement should contain a "for all p^i".
* In Theorem 3: there should be parentheses around "2q" to show that q
  is in the denominator of the fraction and not in the numerator.
* Line 102: "widely solution" is missing a word.
* Assumption 1: add "for all x^{-i}"
* Assumption 2: define cal{X}_*


**Questions:**

* Could the authors comment on whether the following simple crude
  approach would also work to get an adaptive method?
  - Run the non-adaptive method from Theorems 1&2 that guarantees
    constant regret if the noise is multiplicative.
  - If the regret ever exceeds the constant from the regret bound,
    restart with any standard method that gets O(sqrt{T}) regret.
  Clearly this is not as elegant as the approach in the paper, but it
  seems it would solve the suboptimal dependence on q.


**Limitations:**

* The fact that the adaptive method does not recover the optimal rate,
  but O(T^{1/2 + q}) for case 2 should be mentioned much earlier. It
  should at least be mentioned in the introduction, and possibly also in
  the abstract.

* The rates depend on the optimal comparator points p^i, but this
  dependence is not made explicit in the big-Oh notation, and no attempt
  is made to achieve an optimal dependence in the tuning of the step
  sizes. This limitation is clear from the results and not uncommon in
  the related work, so it is perfectly fine and requires no extra
  discussion.


**Strengths And Weaknesses:**

This is a well-written paper, with a nice new adaptive result. It
contains a novel approach of using learning rate separation to deal with
multiplicative noise, which is shown to stabilize the last iterate of
the dual averaging algorithm, and therefore constant regret becomes
possible. The importance of studying multiplicative noise is
sufficiently motivated.

---

> ### Author Response · Authors · 2022-08-02
> **Replies to Reviewer NjYe**
>
> Dear Reviewer,
>
> Thank you very much for your detailed feedback, encouraging remarks, and positive evaluation. We reply to your main questions point-by-point below.
>
>
> 1. **On the proposed interpolation meta-scheme**
>
>     Concerning the "meta-algorithm" that you propose for achieving adaptivity, we mainly foresee two obstacles in implementing it in our problem. First, due to the presence of noise, the regret incurred by any given player is stochastic, and our bound only holds for the expected regret. This is a well-known obstacle and limitation encountered by meta-algorithms of this type, see e.g., the work of Bubeck and Slivkins "The best of both worlds: Stochastic and adversarial bandits" (COLT 2012). In this case, to ensure that an excess of the regret really implies a failure of the algorithm (and is not otherwise due to random fluctuations), we would first need to derive a high-probability version of our results using concentration inequalities -- and this could be highly challenging in the case of multiplicative noise.
>
>     Second, the non-adaptive algorithms that we considered are only really effective when we have full knowledge of the various constants and parameters involved; in particular, the players must know beforehand that the noise is multiplicative and know the associated constant beforehand, a limitation which we feel would somewhat limit the desired interpolation result.
>
>     That being said, *if* the constants are effectively known by the learner(s), we believe it is indeed possible to work out a method that retains the optimal guarantee in the adversarial case, possibly by adapting the more recent techniques of Zimmert et al. ("Beating stochastic and adversarial semi- bandits optimally and simultaneously", ICML 2019). We find this to be a very fruitful research direction for future work, and we will include it as such in our revision.
>
> 1. **Minor remarks**
>
>     Thanks a lot for this highly detailed input. We will fix the typos you spotted and we will make it clear from  the introduction that our adaptive method does not recover the optimal rate in the adversarial setup.
>
> Thanks again for the detailed input and positive evaluation!

---

> > ### Comment · Reviewer_NjYe · 2022-08-08
> > **Reply to the authors' response**
> >
> > I see your point about why the crude approach I suggested would indeed not work. Thanks for clarifying.

---

### Official Review · Reviewer_yRqj · 2022-07-11

**Rating:** 6
**Confidence:** 3
**Soundness:** 3 good
**Presentation:** 3 good
**Contribution:** 3 good

**Summary:**

This paper studies no-regret learning in multi-player smooth games in the presence of noise. The paper analyzes variants of optimistic gradient descent, and shows that when the noise is purely multiplicative, i.e. when the strength of noise is proportional to the gradient norm, all players can achieve regret independent of $T$. An adaptive learning rate scheme is then proposed to eliminate the need to know the learning rates of other players.

**Questions:**

- Can Assumption 4 be avoided, since the final regret guarantees have some sort of boundedness assumptions?

- Can be it be shown that extrapolation is algorithmically necessary? In other words, is it possible for online gradient descent to achieve a similar $O(1)$ regret bound?

- The abstract mentions a smooth extrapolation between worst- and best- case guarantees. However, based on the statement of Theorem 3, it seems that the deviation of any player from the Adapt scheduling would lead to the worse fallback regret bound. Is there an actual "extrapolation" between the two cases? In other words, can regret bounds between $O(exp(1/2q))$ and $O(T^{0.5+q})$ be provided, if only a small number of players deviate slightly from the Adapt scheme?

**Limitations:**

Limitations of this work are appropriately discussed.

**Strengths And Weaknesses:**

Strengths:
- The algorithms are simple, tuning only the learning rates and extrapolation parameter in vanilla OGD. They are also very practical, since they enjoy guarantees in both cooperative (Theorem 3) and adversarial (Proposition 2) scenarios, and can be made parameter-free.
- The results would be a nice addition to online learning literature, where few results beyond the $O(\sqrt{T})$ bound exist when the feedback is noisy.
- The analysis of optimistic gradient descent with noise is novel.

Weaknesses:
- The setting is a bit restricted, since only the unconstrained case is handled, and the interesting results are for the multiplicative noise setting. Dealing with unconstrained space precludes application to matrix games (strategies constrained on simplices) and creates issues in the definition of regret (an artificial bounded set $\mathcal{K}$ is introduced for that). While multiplicative noise is a common assumption in optimization literature, it might not be the most suitable one for game settings, since arguably one of the most common sources of noise is the sampling noise from mixed strategies, which is additive rather than multiplicative. This work would greatly benefit if it can be applied to learning in multi-player matrix games with bandit feedback.
- The authors claim that the $O(1)$ regret bound in the presence of noise to be the first of its kind. It seems to me, however, that Theorem 4.4 in Lin et al. [24] also implies an $O(1)$ regret bound, albeit under a stronger setting. If this is the case, I would suggest rephrasing this claim.

---

> ### Author Response · Authors · 2022-08-02
> **Replies to Reviewer yRqj**
>
> Dear Reviewer,
>
> Thank you very much for your thoughtful comments and positive evaluation! We address below the individual points that you raised in your review.
>
> 1. **On learning with noisy feedback in finite games.**
>
>     We fully agree that our paper is not the end of the story as far as learning with noisy feedback is concerned. As you remark, learning with sampling- or bandit-based information in finite games is a very important topic, and it is definitely an area where one would like to apply the analysis and results of our paper. However, the current state of the art in game-theoretic learning is not yet there, and, in this regard, we believe that our paper is opening the door to a range of tools and techniques that have not yet been considered.
>
> 1. **On the work of Lin et al.**
>
>     To the best of our understanding, Lin et al. [24] do not provide any regret guarantees, but after inspecting their proof and setup, we concur that it is possible to derive constant regret from Theorem 4.4 of [24], under the additional assumption of cocoercivity. This assumption rules out our running example and several of our intended applications, but this is otherwise a very valuable observation -- thanks for bringing this point to our attention, we will add a remark along these lines in our revision.
>
> 1. **On Assumption 4**
>
>     Assumption 4 is necessary for proving our results in the adaptive case. This is a technical assumption that cannot be readily bypassed when dealing with adaptive algorithms and filtration-dependent learning rates.
>
> 1. **On the necessity of the extrapolation step**
>
>     Online gradient descent can indeed achieve $O(1)$ regret under multiplicative noise **in cocoercive games**: as discussed above, this can already be inferred from Theorem 4.4 of Lin et al. [24]. However, in the more general case of merely monotone (or variationally stable) games, even **deterministic** gradient descent fails to achieve low regret, as can be seen in the classic example of $\min_{x}\max_{y} xy$. In this regard, our work serves to highlight the algorithmic tweaks that need to be made in order to achieve constant regret in noisy non-cocoercive settings: extrapolation is required to overcome the lack of cocoercivity (just as in the deterministic case), and the separation of learning rates is required to supply an indirect variance reduction mechanism.
>
> 1. **On interpolating between best- and worst-case guarantees**
>
>     The situation where only a fraction of players deviate from the prescribed "self-play" policy is a very interesting one, but not one that can be handled with any of the techniques that we are aware of. We believe that this is a very fruitful direction for future research, and we will clearly identify it as such in our revision; we will also drop the term "interpolate" from our paper's abstract to avoid any ambiguity or confusion.

---

### Official Review · Reviewer_vptX · 2022-07-11

**Rating:** 8
**Confidence:** 3
**Soundness:** 3 good
**Presentation:** 3 good
**Contribution:** 3 good

**Summary:**

The paper studies simultaneous no-regret learning in a subset of convex games (satisfying a variational stability condition) when players observe noisy estimates of their gradients. An optimistic gradient algorithm is proposed which uses learning rate separation. This allows to prove O(sqrt(T)) regret in the presence of additive noise, and O(1) regret in case of multiplicative noise. This was not possible with existing optimistic gradient schemes. Moreover, the authors propose a primal-dual variant of their method which retains the same guarantees but ensures also O(sqrt(T)) regret in fully adversarial environments. Authors consider also adaptive learning rate selection methods that do not require a-priori global knowledge of the game and finally study the last-iterate convergence of the aforementioned approaches under the considered noise models.

**Questions:**

I have the following questions/comments:
- The authors make the blankett assumptions of convexity and variational stability. Perhaps a discussion about these, and why are they needed/helpful would be beneficial.
- Besides the lack of global optimality and regret guarantees, would the proposed time-scale separation be beneficial for non-convex games too? Why?


**Limitations:**

I don't foresee potential negative societal impact.

**Strengths And Weaknesses:**

Simultaneous learning in games is a relevant problem, which has received significant attention in the past years. Moreover, the study of such dynamics in the presence of noise is definitely important. The results presented are very impressive, original, and sound. Moreover, the authors did a very good job in putting them into the context of existing works and providing intuitions along the way.
Hence, I definitely recommend acceptance.
I don't think the paper has major weaknesses. However, I feel additional experimental results would have made a stronger impact and could serve as ablation for the different convergence bounds and noise models considered.

---

> ### Author Response · Authors · 2022-08-02
> **Replies to Reviewer vptX**
>
> Dear Reviewer,
>
> Thank you for your encouraging comments and positive evaluation! We are delighted that you apprecicate our work, and we reply to your main questions point-by-point below.
>
> 1. **On the need for convexity and variational stability.**
>
>     Convexity (and its variants) is a vital requirement in the literature on online learning; otherwise, it is not possible to transform iterative gradient bounds to bona fide regret guarantees. [By comparison, there are very few works on online *non-convex* optimization, and these works either drastically relax the definition of the regret, or they exploit ad hoc characteristics of the problem to work with a convex reformulation thereof] In a similar vein, variational stability can be seen as a variant convexity assumption for multi-agent environments, where unilateral convexity assumptions do not suffice to give rise to a learnable game. [For example, finite games are unilaterally linear, but finding a Nash equilibrium of a finite game is a PPAD-complete problem]
>
>     In the specific context of our paper, variational stability allows us to establish tighter control on the agents' learning trajectory and, in a sense, to "stabilize" it. More precisely, variational stability provides us with the means to bound some weighted expected second-order path length, which in turn allows us to bound the regret of each player. Some recent works manage to bypass variational stability in deterministic settings and achieve low regret in "merely convex" games (i.e., games that are convex but not variationally stable), but these techniques are inextricably tied to the deterministic structure of the players' feedback. At this point, it is not clear if variational stability can be circumvented in a stochastic setting, but it seems crucial for the last-iterate convergence that we prove in Section 6. We will add a discussion on this point in a subsequent revision of our paper.
>
> 1. **On learning in non-convex games.**
>
>     Learning in non-convex games is a very actively researched topic, but also a very difficult one, with very few known convergence results. In particular, a series of recent results has shown that standard first-order methods can take exponential time to locate a first-order stationary point (which is a drastic relaxation of the notion of a Nash equilibrium) [Daskalakis et al., STOC 2021], or even be trapped in spurious limit cycles that don't contain any critical point of the game under study [Hsieh et al., ICML 2021].
>
>
>     In this highly complex landscape, we expect that the learning rate separation techniques proposed in our paper could resolve convergence failures due to recurrence (e.g., as in the case of bilinear min-max games whose trajectories comprise a foliation of degenerate periodic orbits), but we do not believe it would be possible to overcome the convergence obstructions mentioned above.
>
> Thank you again for your highly encouraging remarks and your positive evaluation. Needless to say, we remain at your disposal if you have any further questions.

---

### Meta-Review · Area_Chair_CxVn · 2022-08-25

**Recommendation:** Accept
**Confidence:** Certain

**Metareview:**

Reviewers are all positive and appreciate the theoretical contributions of the paper. Great work! Please make sure you address all the reviewers' comments and incorporate them (and any new experimental results, if applicable) in your camera-ready.

**Award:**

Yes

---

### Decision · Program_Chairs · 2022-09-14

Accept